# On the Sample Complexity of Lipschitz Constant Estimation

**Julien Walden Huang**  *julien.huang@sjc.ox.ac.uk*
*University of Oxford*

**Stephen Roberts**  *sjrob@robots.ox.ac.uk*
*University of Oxford*

**Jan-Peter Calliess**  *jan-peter.calliess@oxford-man.ox.ac.uk*
*University of Oxford*

**Reviewed on OpenReview:** *https: // openreview. net/ forum? id= UIalYAHdBH*

## Abstract

Estimating the Lipschitz constant of a function, also known as Lipschitz learning, is a fundamental problem with broad applications in fields such as control and global optimization. In this paper, we study the Lipschitz learning problem with minimal parametric assumptions on the target function. As a first theoretical contribution, we derive novel lower bounds on the sample complexity of this problem for both noise-free and noisy settings under mild assumptions. Moreover, we propose a simple Lipschitz learning algorithm called *Lipschitz Constant Estimation by Least Squares Regression* (referred to as LCLS). We show that LCLS is asymptotically consistent for general noise assumptions and offers finite sample guarantees that can be translated to new upper bounds on the sample complexity of the Lipschitz learning problem. Our analysis shows that the sample complexity rates derived in this paper are optimal in both the noise-free setting and in the noisy setting when the noise is assumed to follow a Gaussian distribution and that LCLS is a sample-optimal algorithm in both cases. Finally, we show that by design, the LCLS algorithm is computationally faster than existing theoretically consistent methods, and can be readily adapted to various noise assumptions with little to no prior knowledge of the target function properties or noise distribution.

## 1 Introduction

Many applications are based on the Lipschitz continuity of an objective or target function and depend explicitly on the value of a Lipschitz constant. Examples include: robustness analysis which utilises Lipschitz constants to characterise worst-case behaviour in control settings (Limon et al. (2017), Canale et al. (2014)), system identification which leverages the Lipschitz continuity property to obtain worst-case prediction error bounds for interpolation methods (Milanese and Novara (2004),Beliakov (2006), Calliess et al. (2020)), global optimization algorithms which rely on precise Lipschitz constant estimates to ensure speedy convergence (Jones et al. (1993), González et al. (2016), Malherbe and Vayatis (2017)), multi-armed bandit problems which utilise the Lipschitz constants to obtain asymptotic lower bounds and design algorithms (Magureanu et al. (2014)) or reinforcement learning which utilises the Lipschitz constant to construct safe initial policies (Chakrabarty et al. (2020)). For these applications it is critical that the Lipschitz constant estimate used is sufficiently precise in order to ensure satisfactory performance in their specified goal. A main practical drawback is therefore the dependency on the assumption of prior knowledge of the Lipschitz constant or, in the case that this assumption is not made, the necessity of having to learn a precise Lipschitz constant estimate from data.

Consequently, a number of Lipschitz constant estimation methods (also called Lipschitz learning algorithms) have been developed. For target functions belonging to families of parametric models, Lipschitz learning

approaches generally utilise the structure of the model class to obtain precise estimates (e.g. see extensive literature on the estimation of Lipschitz constants of Neural networks (Virmaux and Scaman (2018)) (Fazlyab et al. (2019))). For frameworks that don't consider a particular parametric family, a majority of the existing approaches are black-box methods that utilise and extend Strongin's classical estimator (Strongin (1973)): $\hat{L} := r \max_{i \neq j} \frac{|f_i - f_j|}{\|x_i - x_j\|}$ where $r \in \mathbb{R}$ is a multiplicative factor and $(x_i, f_i)$ is a data sample with $f_i = f(x_i)$. In particular, we highlight: (Wood and Zhang (1996)) builds on Strongin's estimator by fitting an approximate reverse Weibull distribution to the Lipschitz estimate in the one-dimensional case and using the location parameter as a Lipschitz estimate, (Sergeyev (1995)) utilises Strongin's approach to compute local Lipschitz constant estimates and extends the approach to the multidimensional case by using space filling curves in order to solve a global optimization problem and a more recent approach (Strongin et al. (2019)) proposes dual Strongin Lipschitz estimates: with two differing "local" and "global" multiplicative factors. We remark that the class of Lipschitz learning algorithms described so far does not consider the possibility of observational noise and can explode in value if it exists. In this case, we can consider the Lispchitz constant estimator proposed by both (Novara et al. (2013)) and (Calliess et al. (2020)) which specifically extends Strongin's estimate to deal with bounded observational noise.

Alternative Lipschitz learning approaches that do not directly utilise Strongin's estimate have also been developed. These generally can consider the case of observational noise and include: (Beliakov (2005)) which utilises a short optimisation problem and cross validation/sample splitting to obtain Lipschitz constant estimates, (Bubeck et al. (2011)) which employs a similar idea to Strongin's estimate in order to propose a Lipschitz constant estimator in the context of the Lipschitz multi-armed bandit problem, (González et al. (2016)) which generates Lipschitz constant estimates using the mean function of the gradient function estimate of a fitted Gaussian Process (GP) and is directly computable using the GP-associated covariance function, and (Calliess (2017)) which obtains Lipschitz constant estimates by optimising a Lipschitz interpolation method. Unfortunately, while this class of Lipschitz learning algorithms tends to work well in practice, they generally do not guarantee asymptotic convergence and are of limited theoretical interest.

Despite the wide range of proposed Lipschitz learning algorithms, there has been little theoretical investigation into the Lipschitz constant estimation problem other than various consistency proofs of Strongin-based estimators. It is generally understood that this learning problem, without making further restrictive assumptions on the underlying space of target functions, is inevitably subject to the curse of dimensionality. However, to the best of our knowledge, this intuition has not been explored formally. A first goal of this paper is therefore to provide a theoretical investigation into the Lipschitz learning problem by deriving lower bounds on the sample complexity in the case of both a noiseless and noisy sampling settings. We confirm the general intuition on the difficulty of the Lipschitz learning problem by demonstrating that the problem has a sample complexity lower bound that scales at least exponentially on the function input dimension in both the noiseless case and the noisy sampling case when the noise is assumed to be Gaussian.

Our theoretical results imply that a precise estimation of the Lipschitz constant requires a significant number of samples. This is computationally problematic for classical Strongin-based Lipschitz learning algorithms due to the fact that the computational complexity of these methods can be shown to be quadratic in the number of samples: $O(n_{samples}^2)$. While the non-Strongin based estimators discussed above could potentially be less computationally expensive, they only provide heuristic or experimental convergence guarantees and are generally difficult to study from a theoretical perspective. Therefore, in light of our lower bounds on sample complexity, we propose a novel algorithm for Lipschitz learning called LCLS (short for *Lipschitz Constant estimation by Least Squares regression*) that has a linear computational complexity in the number of sample points and for which we can derive theoretical guarantees on asymptotic convergence and finite sample behaviour in a general noisy sampling setting. The optimality of the lower bounds on the sample complexity of the Lipschitz learning problem in both the noiseless sampling setting and in the noisy sampling setting under Gaussian noise assumptions derived in the first part of the paper follow from these theoretical results.

In practice, the proposed LCLS algorithm provides a theoretically motivated and computationally quick way of estimating the Lipschitz constant. With minimal fine-tuning, LCLS can be plugged into any computational method that utilises a Lipschitz constant estimate – see above discussion – under any sampling noise assumptions. We provide an example of such a procedure in the context of nonparametric regres-

sion for system identification in control by combining the LCLS algorithm with a classical nonlinear set-membership/Lipschitz interpolation framework and illustrating the empirical performance of the combined regression method through a series of short experiments.

A more comprehensive list of the contributions of this paper is given below.

### 1.1 Contributions and Outline of Paper

In this paper, we provide a rigorous treatment of the Lipschitz constant estimation problem discussed above. In particular, we make the following contributions:

1. In Section 2, we provide novel theoretical lower bounds on the sample complexity of the general Lipschitz learning problem (for $p \in \{1, 2\}$) in the noiseless sampling setting (see Theorem 2.2) and of a slightly modified version of the problem[1] in the noisy sampling setting (see Theorem 2.5) when the target function $f : \mathcal{X} \subset \mathbb{R}^d \to \mathbb{R}$ satisfies a regularity condition $C^2(\mathcal{X}, K)$ defined in Assumption 2. We note that these bounds can be equivalently stated as lower bounds on the convergence rate of the general Lipschitz learning problem (see Corollaries 2.3 and 2.6). As far as the authors are aware, the sample complexity and convergence bounds derived in this paper are the first theoretical results pertaining to the convergence of the general Lipschitz learning problem. We show in Section 3 that our proposed lower bound on the sample complexity rate is optimal in both the noiseless sampling setting and the noisy sampling setting under a Gaussian assumption.

2. In Section 3.1, we propose a least squares-based Lipschitz learning (LCLS) approach that utilises a partition of the input space $\mathcal{X}$ and local least squares estimates in order to generate a Lipschitz constant estimate. As discussed in the introduction, our motivation for developing the LCLS algorithm rests on the following two desiderata for an estimator:
   - it should be both theoretically and computationally tractable, and,
   - directly applicable across all noise settings considered in the literature and implementable without any prior knowledge of target function properties or of the noise structure.

3. In Sections 3.2 and 3.3, we investigate theoretical properties of the proposed algorithm:
   - Asymptotic convergence for general partition choice in noiseless and general noisy sampling set-ups (Section 3.2, see Theorem 3.7).
   - Finite sample guarantees in the noiseless and general noisy sampling set-ups when the partition is constructed using regular hypercubes (Section 3.3, see Theorem 3.10, Corollary 3.16). These guarantees can be used to provide an upper bound on the sample complexity of the Lipschitz learning problem and show that the complexity rates derived in the first part of the paper match in both the noiseless and noisy setting under a Gaussian assumption. (Section 3.3, see Remark 3.11, 3.12, Theorem 3.15).

4. In Section 3.4, we illustrate and compare the empirical performance of the LCLS algorithm against Strongin-based Lipschitz learning algorithms on a set of test functions. We consider both the noiseless and noisy sampling settings. We find that while the benchmark Strongin-based algorithms converges slightly faster in terms of number of samples, our proposed algorithm converges faster in terms of computation time for all functions in the test set (see Figure 4). This is despite the fact that we consider noise settings for which the benchmark algorithms are specifically designed in our experiments.

5. In Section 4, we explore the application of the various theoretical results and the LCLS algorithm derived in this paper to the fields of *Global Optimisation* and *Nonparametric Regression for System Identification*. More specifically, we propose a lower bound on the sample complexity of adaptive Lipschitz optimisation algorithms that follows from one of the theoretical results stated in Section 2 and a new nonparametric regression method constructed by combining the LCLS algorithm of Section 3 with a classical nonlinear set membership framework (see Milanese and Novara (2004)).

---

[1]Which can be shown to be equivalent for the majority of existing Lipschitz learning algorithms.

## 2 Assumptions and Sample Complexity Lower Bound

In this section, we provide the standing assumptions of the paper and state the main results pertaining to theoretical lower bounds on the sample complexity of Lipschitz learning algorithms.

### 2.1 Basic Assumptions

Let $p \in \mathbb{N}$, $d \in \mathbb{N}$, a function $f : \mathcal{X} \subset \mathbb{R}^d \to \mathbb{R}$ is said to be Lipschitz continuous with respect to a norm $\|.\|_p$ if there exists a positive real value $L_p(f) \in \mathbb{R}$ such that $|f(x) - f(y)| \leq L_p(f)\|x - y\|_p$, for all $x, y \in \mathcal{X}$. The smallest constant satisfying this condition denoted $L_p^*(f)$ is called the (best) Lipschitz constant and can be interpreted as the tightest bound on the rate of change of $f$. Furthermore, for any $L \geq 0$, we define the class of Lipschitz continuous functions as

$$\mathcal{F}_p(L) := \{h : \mathcal{X} \to \mathbb{R} | h \text{ is Lipschitz } \wedge L_p^*(h) = L\}.$$

The Lipschitz learning problem therefore considers the estimation of $L_p^*(f)$ when $f$ is an unknown target function. As described in the introduction, we consider a general version of this problem where $f$ is considered black-box and can can only be accessed through queries to a, possibly noisy, oracle. As $f$ is not assumed to belong to any parametric family, other assumptions are needed[2] in order to derive theoretical bounds on the sample complexity. For our results, we make the following two assumptions on the input space $\mathcal{X}$ and the regularity of $f$.

**Assumption 1** *(Domain) The domain $\mathcal{X}$ of the target function $f$ is a convex and compact subset of $\mathbb{R}^d$.*

**Assumption 2** *(Functional) The target function $f \in C^2(\mathcal{X})$ and there exists an upper bound $K \in \mathbb{R}_+$ on the second-order partial derivatives of $f$, i.e. $|\frac{\partial f}{\partial x_i \partial x_j}| \leq K$ for all $x \in \mathcal{X}$ and $i, j \in \{1, ..., d\}$. Furthermore, $f$ can be extended on an open set $\bar{\mathcal{X}}$ such that $\mathcal{X} \subset \bar{\mathcal{X}}$.*

For a given $K \in \mathbb{R}_+$, we denote by $C^2(\mathcal{X}, K)$ the class of functions that satisfies Assumption 2 with an upper bound $K$ on the second degree partial derivatives. It is important to point out that this bound does need to be tight and that if Assumption 1 holds then any $f \in C^2(\mathcal{X})$ automatically belongs to $C^2(\mathcal{X}, \bar{K})$ for some $\bar{K} \in \mathbb{R}_+$. Finally, we assume that we have access to the target function $f$ through an oracle $\Omega : \mathcal{X} \to \mathbb{R}$ – defined formally below for each sampling setting – which can be queried in order to generate observations of $f$. In particular, this oracle can be freely used by any Lipschitz learning algorithm as described in the following definition.

**Definition 2.1** *(Lipschitz Learning Algorithms) We define $\mathcal{L}_{n,p}(\mathcal{X})$ as the set of all $\|.\|_p$-Lipschitz learning algorithms that utilise at most $n \in \mathbb{N}$ queries to the Oracle $\Omega$ with inputs in $\mathcal{X}$. The sampling procedure is considered to be a part of the Lipschitz learning algorithm and $\forall \hat{L} \in \mathcal{L}_{n,p}(\mathcal{X})$ we denote the set of generated sample points by $G_{\mathcal{X}}^{\hat{L}} = \{(x_i^{\hat{L}}, \Omega(x_i^{\hat{L}}))_{i=1,...,n}\}$.*

We note that Definition 2.1 defines a general class of Lipschitz learning algorithms without any structural specifications and that the inclusion of the sampling procedure in the algorithm is common for applications in both control and global optimisation.

### 2.2 Noiseless Sampling Setting

Assumptions (1)-(2) are sufficient to formulate a lower bound on the sample complexity rate of the Lipschitz learning problem in the case where one has access to an oracle[3] $\Omega$ that can be queried to obtain noiseless observations of the underlying target function. Formally, the noiseless Oracle is described by

$$\Omega : \ \mathcal{X} \to \mathbb{R}$$

---

[2]Otherwise, a theoretical characterisation of the Lipschitz learning problem is not feasible.

[3]Note: in the noiseless case, the oracle and the target function are equivalent.

$$x \xmapsto{\Omega} f(x).$$

The lower bound on the sample complexity of any Lipschitz learning algorithm is given in the following theorem.

**Theorem 2.2** *(Sample Complexity Bound – Noiseless) Let $M \in \mathbb{R}_+$, $d \in \mathbb{N}$, $p \in \{1,2\}$ and suppose $\mathcal{X} := [0,M]^d$. Assume that Assumption (2) holds and that a noiseless Oracle $\Omega$, (described above) is available. $\forall L^* \geq 0, \forall \epsilon \in (0, MK)$, if*

$$\inf_{\hat{L} \in \mathcal{L}_{n,p}(\mathcal{X})} \sup_{f \in C^2(\mathcal{X}, K) \cap \mathcal{F}_p(L^*)} |\hat{L}(f) - L^*| < \epsilon$$

*then*

$$n \geq \left( C(d,p) \frac{MK}{\epsilon} \right)^d$$

*In this paper, we find $C(d,p) = \frac{1}{20 d^{\frac{1}{p} - \frac{1}{2}}}$, however this value has not been optimized.*

Theorem 2.2 provides a lower bound on the minimum number of oracle queries that are needed in order for a Lipschitz learning algorithm to ensure a precise estimate of the Lipschitz constant for all underlying target functions in $C^2(\mathcal{X}, K)$. As speculated in the introduction, it shows that the Lipschitz Learning problem is a computationally expensive one that depends heavily on the input dimension. The lower bounding expression is given as a function of the size of the input space ($M$), the assumed bound on the second order partial derivatives ($K$) and the precision parameter ($\epsilon$) but is independent of the true Lipschitz constant ($L^*$) of the target function. The product $MK$ can be understood as a bound on the maximum change in the gradient values of functions in $C^2(\mathcal{X}, K)$. In Section 3, the proposed LCLS algorithm will be shown capable of estimating the Lipschitz constant of all functions $f$ in $C^2(\mathcal{X}, K)$ using $O((\frac{MK}{\epsilon})^d)$ queries to the noiseless sampling oracle $\Omega$ implying that the lower bound on the sample complexity rate stated in Theorem 2.2 is optimal.

An equivalent reformulation of Theorem 2.2 in the form of a lower bound on the convergence rate of Lipschitz learning algorithms is provided in the following corollary.

**Corollary 2.3** *(Convergence Rate Bound – Noiseless) Assume the same setting as Theorem 2.2. Then, $\forall L^* \geq 0$,*

$$\inf_{\hat{L} \in \mathcal{L}_{n,p}(\mathcal{X})} \sup_{f \in C^2(\mathcal{X}, K) \cap \mathcal{F}_p(L^*)} |\hat{L}(f) - L^*| \geq C(d,p) \frac{MK}{\sqrt[d]{n}}$$

*where $C(d,p)$ is defined in Theorem 2.2.*

Corollary 2.3 is generally more practical to use then Theorem 2.2 when considering convergence properties of applications of Lipschitz constant estimators. In Section 4, we show how Corollary 2.3 can be applied in conjunction with recent theoretical results (Bachoc et al. (2021)) to derive lower bounds on the sample complexity of adaptive Lipschitz optimisation algorithms.

## 2.3 Noisy Setting

In many practical cases, the sampling oracle cannot be assumed reliable and only approximate observations of the target function are obtainable. In this case, we model $\Omega : \mathcal{X} \to \mathbb{R}$ as being corrupted by additive noise and a new lower bound on the sample complexity of Lipschitz learning algorithms can be derived. In order to do so, additional assumptions must be made on the additive observational noise process.

**Assumption 3** *(Noisy Oracle – Gaussian Noise) Let $\sigma^2 > 0$. We define a noisy sampling oracle as*

$$\tilde{\Omega} : \ \mathcal{X} \to \mathbb{R}$$

$$x \xmapsto{\tilde{\Omega}} \tilde{f}_x := f(x) + \gamma_x$$

*where $\gamma_x$ are independent Gaussian random variables ($x \in \mathcal{X}$) with mean $0$ and variance $\sigma^2$. Note: $\gamma_x$ is an abuse of notation as the noise is not dependent on the input $x$. In other words: if $x \in \mathcal{X}$ is sampled twice, then $\gamma_x^1 \neq \gamma_x^2$.*

As the class of Lipschitz learning has been loosely defined so far, with no parametric or functional assumptions, a slight reformulation of the Lipschitz learning problem is needed in order to derive lower bounds on the sample complexity. Consider the function class $C^2(\mathcal{X}, K)$ as defined above and $p \in \mathbb{N}$. It is known (e.g. see Lemma B.2) that for all $f \in C^2(\mathcal{X}, K)$, $L_p^*(f) = \max_{\mathcal{X}}\{\|\nabla f(x)\|_q\}$ if $p = 1, 2$ and $L_p^*(f) \leq \max_{\mathcal{X}}\{\|\nabla f(x)\|_q\}$ otherwise, where $q$ is the Hölder conjugate of $p$. Then, instead of directly considering the estimation error $|\hat{L}(f) - L_p^*|$ with a Lipschitz learning algorithm $\hat{L}(f) \in \mathcal{L}_{n,p}(\mathcal{X})$ as done in Theorem 2.2, one can consider the problem of obtaining $x^{\hat{L}(f)} \in \mathcal{X}$ such that $|\|\nabla f(x^{\hat{L}(f)})\|_q - L_p^*|$ is minimised. In this case, we say that the algorithm $\hat{L}(f)$ belongs to the class $\overline{\mathcal{L}}_{n,p}(\mathcal{X})$ of *Lipschitz Learning search algorithms* which is defined formally as follows:

**Definition 2.4** *(Lipschitz Learning Search Algorithms) We define $\overline{\mathcal{L}}_{n,p}(\mathcal{X})$ as the set of all $\|.\|_p$-Lipschitz learning search algorithms that utilise at most $n \in \mathbb{N}$ queries to the Oracle $\tilde{\Omega}$ with inputs in $\mathcal{X}$ in order to produce an estimate $\hat{x}$ that aims to minimise: $Loss(\hat{x}, f) := |\|\nabla f(\hat{x})\|_q - L_p^*|$.*

This type of paradigm is similar to the one considered in the literature on minimax global optimisation where one generally aims to obtain the minimising argument $\hat{x} \in \mathcal{X}$ of a target function rather than directly estimating the minimum (e.g. Bull (2011)). We stress that if a good estimate of $x^{\hat{L}(f)}$ can be obtained, then it is relatively straightforward to obtain an accurate Lipschitz constant estimate by computing a local gradient or slope estimate of the target function near $x^{\hat{L}(f)}$. In fact, the majority of Lipschitz algorithms either directly or implicitly operate by maximising gradient or slope estimates (e.g. Strongin (1973), Calliess et al. (2020), the LCLS algorithm proposed in this paper) and could therefore be trivially modified to generate a search estimate: $x^{\hat{L}(f)}$.

Using Assumption 3, the following lower bound on the sample complexity rate can be derived in the noisy sampling setting.

**Theorem 2.5** *(Sample Complexity Bound – Noisy) Let $M \in \mathbb{R}_+$, $d \in \mathbb{N}$, $p \in \{1, 2\}$ and suppose $\mathcal{X} := [0, M]^d$. Assume that Assumptions (2)-(3) hold, that one has access to a noisy oracle $\tilde{\Omega} : \mathcal{X} \to \mathbb{R}$ as specified in Assumption (3) and that the sample inputs are uniformly and independently sampled on $\mathcal{X}$. Define $\overline{\mathcal{L}}_{n,p}(\mathcal{X})$ as in Definition 2.4. If there exists $\delta \in (0, 1)$ such that*

$$\lim_{\epsilon \to 0^+} \inf_{\hat{L} \in \overline{\mathcal{L}}_{n,p}(\mathcal{X})} \sup_{f \in C^2(\mathcal{X}, K)} \mathbb{P}(Loss(x^{\hat{L}(f)}, f) > \epsilon) \leq \delta$$

*then,*

$$n \in \Omega\left(\frac{\sigma^2 M^d K^{d+2} \log(\frac{MK}{\epsilon})}{\epsilon^{d+4}}\right).$$

In contrast to the sample complexity bounds obtained in the noiseless sampling setting, the bounds proposed in Theorem 2.5 only hold asymptotically and can therefore not be utilised in order to obtain finite sample guarantees. Furthermore, the Lipschitz learning (search) algorithms considered in Theorem 2.5 are assumed to be passive[4] (as the sampling inputs are sampled randomly) as opposed to the active algorithms considered in Theorem 2.2. Nonetheless, the obtained bounds provide insight into the necessary sampling requirements needed to ensure convergence for the Lipschitz learning (search) problem in the noisy sampling setting. In Section 3.3, we will show that the LCLS algorithm matches the convergence rate stated in Theorem 2.5 under the same assumptions implying that the rate $\Theta\left(\sigma^2 \frac{M^d K^{d+2} \log(\frac{MK}{\epsilon})}{\epsilon^{d+4}}\right)$ is optimal.

Finally, as in the noiseless sampling setting, an equivalent reformulation of Theorem 2.5 is provided in the form of a probabilistic lower bound on the convergence rate of Lipschitz learning (search) algorithms.

---

[4]Note: we are not aware of any existing active Lipschitz learning algorithms as Lipschitz constant estimation is usually computed as a secondary task to a main objective (e.g. optimisation, nonparametric regression).

**Corollary 2.6** *(Sample Complexity Bound – Noisy) Assume the same setting as Theorem 2.5. Then, for all $\delta \in (0,1)$, there exists $C > 0$ such that*

$$\lim_{n \to \infty} \inf_{\hat{L} \in \overline{\mathcal{L}}_{n,p}(\mathcal{X})} \sup_{f \in C^2(\mathcal{X}, K)} \mathbb{P}(Loss(x^{\hat{L}(f)}, f) > CMK \left( \frac{\log(MKn)}{nM^4 K^2 \sigma^2} \right)^{\frac{1}{d+4}}) > \delta.$$

Theorem 2.5 and Corollary 2.6 are particularly interesting in the context of system identification for control applications (e.g. Milanese and Novara (2004), Calliess et al. (2020)) where robustness properties depend explicitly on estimating a feasible Lipschitz constant from noisy data. These frameworks often ignore the modeling error arising from the Lipschitz constant estimation which is problematic when the goal is to provide worst-case guarantees. One source of usefulness for the two bounds stated in this subsection is therefore to provide a theoretical understanding of the worst-case estimation of Lipschitz constants in this context and therefore to make possible a more realistic robustness analysis of Lipschitz constant-based system identification methods in practice. A short illustrative comparison of the convergence rate given in Corollary 2.6 with the convergence of existing Lipschitz learning algorithms used for system identification purposes is given in Figure 3.

**Remark 2.7** *(Comparison to Nonparametric Estimation) The optimal convergence rates of nonparametric estimation in the noisy sampling setting with Gaussian noise are well-known (Tsybakov (2008)). In particular, the uniform[5] convergence rate of the nonparametric estimation of first-order partial derivatives on $C^2(\mathcal{X}, K)$ for some $K > 0$ is given by $\Theta \left( \left( \frac{log(n)}{n} \right)^{\frac{1}{d+4}} \right)_{n \in \mathbb{N}}$ which corresponds exactly to the lower bounds derived in Corollary 2.6. This implies that although Lipschitz learning seems more straightforward than partial derivative estimation, asymptotically their sample complexities are equivalent. Note that this observation is somewhat unsurprising as similar results hold in the context of global optimisation (Bull (2011), Wang et al. (2018)).*

# 3 Lipschitz Constant estimation by Least Squares regression (LCLS)

The theoretical results of Section 2 imply that a significant number of samples must be used in order to obtain a precise estimation of the best Lipschitz constant. As noted in the introduction, this is problematic computationally for classical Strongin-based Lipschitz learning algorithms due to the fact that the computational complexity of these methods can be shown to be quadratic in the number of samples. Using existing non-Strongin based methods could resolve this computational problem, however obtaining convergence guarantees would then be difficult as this class of methods is generally complicated to study from a theoretical perspective. Therefore, in the goal of obtaining a Lipschitz learning approach that can provide asymptotically consistency guarantees, has low computational complexity and for which we can derive upper bounds on the sample complexity (in the goal of comparing with the sample complexity lower bounds derived in Section 2), we define a new estimator: *Lipschitz Constant estimation by Least Squares regression* (LCLS).

## 3.1 Overview

The general intuition behind the Lipschitz learning algorithm proposed in this paper follows from the simple observation that the coefficients from a least squares regression can be interpreted as a local approximation of the gradient and that the maximum q-norm of the gradient of $f$ on $\mathcal{X}$ coincides[6] (for certain values of $p \in \mathbb{N}$) with the best Lipschitz constant associated to the p-norm, where $q$ is the Hölder-conjugate of $p$, i.e. $\frac{1}{p} + \frac{1}{q} = 1$. Therefore, by using a partition $\mathcal{H}$ of the input space $\mathcal{X}$ that is sufficiently refined to properly capture the gradient variation of $f$ and computing the maximum $q$-norm of the least squares coefficients associated to each subset of $\mathcal{H}$, a precise estimate of the Lipschitz constant is obtainable. Practically, in order to ensure that the refinement of the partition suffices[7], the proposed estimation framework is designed

---

[5]With respect to $\|.\|_\infty$.

[6]See Lemma B.2 in Appendix for a formal statement.

[7]In the case where the upper bound K given in Assumption 2 is known beforehand it is possible to directly partition at the required refinement level (See Theorem 3.10 for example).

---

**Algorithm 1** General LCLS

**Input:** $\tilde{\Omega}$ (Oracle), $(\mathcal{H}_I)_{I\in\mathbb{N}}$ (Partition Sequence)
**Output:** $\{\hat{L}_I\}$ (Lipschitz Estimates)
**procedure:** LCLS($\tilde{\Omega}$, $(\mathcal{H}_I)_{I\in\mathbb{N}}$)
  **initialise:** $I \leftarrow 1$
  **repeat**
    $\hat{L}_I \leftarrow 0$
    **for** $H \in \mathcal{H}_\mathcal{I}$ **do**
      $(X_I^H, \tilde{f}_I^H) \leftarrow D_I^H$ generated by $\tilde{\Omega}$
      $\hat{\beta}_I^H \leftarrow (X_I^{H\top} X_I^H)^{-1} X_I^{H\top} \tilde{f}_I^H$
      $\hat{L}_I \leftarrow \max(\|\hat{\beta}_I^H\|_q, \hat{L}_I)$
    **end**
    $I \leftarrow I + 1$
**return** $\{\hat{L}_I\}_{I\in\mathbb{N}}$

**Algorithm 2** Hypercube LCLS[8] on $[0, M]^d$

**Input:** $\tilde{\Omega}$ (Oracle), $K$ (Bound from (2)), $\eta$ (covering constant), $\sigma^2$ (noise variance), $(\epsilon, \delta)$ (precision)
**Output:** $\hat{L}$ (Lipschitz Constant Estimation)
**procedure:** LCLS($\tilde{\Omega}$, $K$, $\eta$, $\sigma^2$, $(\epsilon, \delta)$)
  **initialise:** $\hat{L} \leftarrow 0$ $I \leftarrow (C_1(d)\frac{MK}{\sqrt{\eta}\epsilon})$,
  $N_I \leftarrow (C_2(d,q)\frac{\sigma^2}{\eta\delta}\frac{I^{d+2}}{M^2\epsilon^2})$
  $\mathcal{H} \leftarrow$ hypercube partition of $[0, M]^d$ with side-length $\frac{M}{I}$
  **for** $H \in \mathcal{H}$ **do**
    $(X^H, \tilde{f}^H) \leftarrow D^H$ generated by $\tilde{\Omega}$
    $\hat{\beta}^H \leftarrow (X^{H\top} X^H)^{-1} X^{H\top} \tilde{f}^H$
    $\hat{L} \leftarrow \max(\|\hat{\beta}^H\|_q, \hat{L})$
**end**
**return** $\hat{L}$

---

Figure 1: *Algorithm 1 details the implementation of the LCLS algorithm for a general input space and partition choice. Algorithm 2 details the theoretical implementation given in Theorem 3.10 of the LCLS algorithm when the input space is a hypercube $[0, M]^d$ and the partitions are regular. In practice, $I \in \mathbb{N}$ and $N_I \in \mathbb{R}_+$ can be set heuristically in order to improve convergence. We note that the generated data points $X_I^H$ used by the two algorithms are selected arbitrarily in each $H \in \mathcal{H}_I$. In order to ensure convergence of the LCLS algorithm, $X_I^H$ will need to verify Assumption 4 for all $I \in \mathbb{N}$ and $H \in \mathcal{H}_I$.*

as an iterative method that utilises a sequence of increasingly fine convex partitions $(\mathcal{H}_I)_{I\in\mathbb{N}}$ that are given as input. A brief technical description of an iteration of the algorithm can be described as follows: *For a given iteration, indexed by $I \in \mathbb{N}$, a set of observations $D_I^H := \{(x_{H_i}, \tilde{f}_{H_i})\}_{i\in\{1,...,N_I^H\}}$ is generated by an oracle $\Omega : \mathcal{X} \to \mathbb{R}$ (defined in Section 2) for each subset $H$ of the partition $\mathcal{H}_I$ and used individually to compute the coefficients $\{\beta_I^H\}_{H\in\mathcal{H}}$ of an ordinary least squares regression for each subset $H \in \mathcal{H}_I$. The Lipschitz constant estimate can then be directly computed: $\hat{L}_I := \max_{H\in\mathcal{H}_I}\{\|\beta_I^H\|_q\}$ where $q$ is the Hölder-conjugate of $p$.*

**Definition 3.1** *(Notation Overview) For a partition $\mathcal{H}_I$ of $\mathcal{X}$ and a set of samples $D_I := \{(x_i, \tilde{f}_i)\}_{i\in\{1,...,N_I\}}$ as described above, we utilise the following notation.*

1. *The subset of samples that belongs to $H \in \mathcal{H}_I$ is denoted $D_I^H := \{(x_{H_i}, \tilde{f}_{H_i})\}_{i\in\{1,...,N_I^H\}}$. Note: samples can only belong to one subset $H$. If a sample point is on the border between two sets, then it can be included in either design matrix.*

2. *We denote the design matrices of the least squares regressions $X_I^H \in \mathbb{R}^{N_I^H \times (d+1)}$ and the observation vectors $\tilde{f}_I^H \in \mathbb{R}^{N_I^H}$;*

$$X_I^H = \begin{bmatrix} 1 & x_{H_1}^\top \\ 1 & x_{H_2}^\top \\ \vdots & \vdots \\ 1 & x_{H_{N_I^H}}^\top \end{bmatrix}, \tilde{f}_I^H = \begin{bmatrix} \tilde{f}_{H_1} \\ \tilde{f}_{H_2} \\ \vdots \\ \tilde{f}_{H_{N^H}} \end{bmatrix} = \underbrace{\begin{bmatrix} f_{H_1} \\ f_{H_2} \\ \vdots \\ f_{H_{N_I^H}} \end{bmatrix}}_{f_I^H :=} + \underbrace{\begin{bmatrix} \gamma_{H_1} \\ \gamma_{H_2} \\ \vdots \\ \gamma_{H_{N_I^H}} \end{bmatrix}}_{\gamma_I^H :=}, \text{ where } \forall k \in \{1, ..., N_I^H\} \text{ and where}$$

*$(x_{H_k}, \tilde{f}_{H_k})$ is a sample point contained in $D_I^H$ with $\tilde{f}_{H_k} := \tilde{\Omega}(x_{H_k})$ and by abuse of notation; $\gamma_{H_k} = \gamma_{x_{H_k}}$.*

3. *We denote by $[\hat{b}_I^H, \hat{\beta}_I^H] \in \mathbb{R}^{d+1}$ (where $\hat{b}_I^H \in \mathbb{R}$ is the intercept) the least squares coefficients associated to $H \in \mathcal{H}_I$ and computed using $X_I^H$ and $\tilde{f}_I^H$.*

The LCLS algorithm is described here in its most general form in order to allow flexibility in the choice of the input space partitions and sampling scheme. Algorithm 1 provides an algorithmic description of this approach. A more specific implementation of the LCLS algorithm which utilises a regular hypercube

---

[8]The method described in this algorithm corresponds to the specific case where the $(K, \sigma^2)$ variables are known.

partition of the input space is given in Algorithm 2 and discussed later on in this section in Theorem 3.10 and ensuing discussions. As one might expect, the structure of $(\mathcal{H}_I)_{I \in \mathbb{N}}$ is a key part of the LCLS estimator. In practice, these partitions can be defined using domain or functional knowledge in order to better estimate the gradient variation and therefore speed up the convergence of the algorithm. The distribution of the sample points given by $(N_I^H)_{n \in \mathbb{N}}$ should also be considered carefully and can be selected in a partition dependent way to take advantage of any prior knowledge of $f$ or of the underlying noise distribution. We note that the relation between the structure of $(\mathcal{H}_I)_{I \in \mathbb{N}}$ and $(N_I^H)_{n \in \mathbb{N}}$ is essential in the proofs of Theorem 3.7 and Theorem 3.10.

The following variables are used to formally describe a partition belonging to $(\mathcal{H}_I)_{I \in \mathbb{N}}$.

**Notation 3.2** *Let $\delta(A) = \sup_{x,y \in A} \|x - y\|_2$ denote the diameter function and $B_r(x)$ the d-dimensional ball centered in $x \in \mathcal{X}$ and with radius $r$ with respect to $\|.\|_2$.*

**Definition 3.3** *(Partition Variables) Let $\mathcal{H}_J \in (\mathcal{H}_I)_{I \in \mathbb{N}}$. We define the following two $\mathcal{H}_J$ related quantities: the maximum diameters of $\mathcal{H}_J$: $\{\Delta_J^H\}_{H \in H_J}$, $\Delta_J^H := \delta(H)$ and the minimum diameters of the biggest subset-inscribed balls of $\mathcal{H}_J$: $\{\delta_J^H\}_{H \in H_J}$, $\delta_J^H := 2\max\{r \in \mathbb{R}_+ | \exists x \in H \text{ such that } B_r(x) \subset H\}$.*

The quantities $\{\Delta_J^H\}_{H \in H_J}$ and $\{\delta_J^H\}_{H \in H_J}$ are used in Definition 3.5 and in Theorem 3.7 to define sufficient conditions on the structure of the $(\mathcal{H}_I)_{I \in \mathbb{N}}$ partitions in order for the general version of the LCLS algorithm to converge.

We conclude this subsection by giving a result on the computational complexity of the proposed algorithm.

**Proposition 3.4** *(Computational Complexity of LCLS) The computational complexity of the Lipschitz Constant Least Squares Estimator is $O\left(d^2 n_{samples}\right)$ where $n_{samples}$ denotes the number of observations sampled by the algorithm and the $d \in \mathbb{N}$ is the input dimension of the target function.*

The computational complexity derived in Proposition 3.4 is significantly smaller than the complexity of Strongin-based approaches which is $O\left(d n_{samples}^2\right)$. The difference in computation speed is illustrated empirically on a set of test functions in Section 3.4.

## 3.2 General Theoretical Analysis

An investigation of the theoretical behaviour and performance of the proposed LCLS algorithm is carried out in this section. This analysis provides an understanding of the design constraints required for the construction of the input space partitions and for the choice of sampling schemes in order to ensure satisfactory performance – see Remark 3.8. We begin by stating an asymptotic convergence result for the general form of the algorithm in the noiseless and general noisy sampling settings before stating and discussing finite sample results for a more concrete application of LCLS when the partition of the input space is constructed to be a set of regular hypercubes.

The following definition defines two quantities $(a_I)_{I \in \mathbb{N}}$, $(b_I)_{I \in \mathbb{N}}$ as a function of $\{\Delta_J^H\}_{H \in H_J}$, $\{\delta_J^H\}_{H \in H_J}$, $(\{N_I^H\}_{H \in \mathcal{H}_\mathcal{I}})_{I \in \mathbb{N}}$ and $(|\mathcal{H}_I|)_{I \in \mathbb{N}}$ in order to alleviate notation[9]. They will be used to describe the conditions on the structure of the input partition sequence needed in order to ensure asymptotic consistency.

**Definition 3.5** *For any sequence of convex and compact partitions, $(\mathcal{H}_I)_{I \in \mathbb{N}}$, we construct the following sequences:*

- $(a_I)_{I \in \mathbb{N}}$, $a_I = \max_{H \in \mathcal{H}_I}(\frac{(\Delta_I^H)^2}{\delta_I^H})$
- $(b_I)_{I \in \mathbb{N}}$, $b_I = \max_{H \in \mathcal{H}_I}\left(\frac{|\mathcal{H}_I|}{N_I^H (\delta_I^H)^2}\right)$.

Before stating the first main result of this section, a condition on the sampling procedure used by the LCLS algorithm and a generalisation of the Oracle noise assumption of Section 2 are given.

---

[9]Here. $|.|$ denotes the cardinality operator.

**Definition 3.6** *Let $H \subseteq \mathcal{X}$ be compact and convex and denote by $D_I^H := \{(x_{H_i}, \hat{f}_{H_i})\}_{i \in \{1, \ldots, N_I^H\}}$ the subset of generated or archived data samples in $H$. We say that $H$ is $(\epsilon, \eta)$-covered for $\epsilon > 0, \eta \in ]0, 1]$ if there exists an $\epsilon$-cover of $H$ (with respect to $\|.\|_2$) with at least $\eta N_I^H$ samples of $D_I^H$ in each of the balls associated to an element in the $\epsilon$-cover.*

**Assumption 4** *(Sampling) For a given $\eta \in (0, 1]$, the sampling scheme selected for LCLS is such that $\forall I \in \mathbb{N}$ and $\forall H \in \mathcal{H}_I$, $H$ is $(\frac{\delta_I^H}{8}, \eta)$-covered.*

The sampling condition stated in Assumption 4 is used in order to ensure the stability of the least squares coefficient as the sequence of partitions becomes increasingly refined and can be satisfied by using quasi-Monte Carlo schemes in practice. In essence, it ensures that the samples are sufficiently well distributed in each subset $H$ of the partition $\mathcal{H}_I$, avoiding extreme cases such as when a single sample input gets repeatedly sampled. A possible (conservative) value for $\eta$ in the regular hypercube implementation of the LCLS algorithm of Section 3.3 is given by $\eta = \frac{vol(B_1(0))}{2^{3d+2}}$ with $vol(B_1(0))$ denoting the volume of the d-dimensional unit ball. In the proof of Theorem 3.15, under a uniform and independent sampling (on $\mathcal{X}$) assumption, Assumption 4 is shown to hold for $\eta = \frac{vol(B_1(0))}{2^{3d+1}3^d}$ with high probability when the number of observations is sufficiently large.

The second assumption of this section generalises the Gaussian noise assumption made in Assumption 3 to include all distributions with zero mean and finite variance and weaken the assumption of independence.

**Assumption 5** *(Noisy Oracle – General) Let $\sigma^2 > 0$ and denote by $\mathcal{D}(0, \sigma^2)$ the set of all probability distributions on $\mathbb{R}$ with zero mean and finite variance $\sigma^2 > 0$. We define a general noisy sampling oracle as*

$$\tilde{\Omega} : \ \mathcal{X} \to \mathbb{R}$$

$$x \xmapsto{\tilde{\Omega}} \tilde{f}_x := f(x) + \gamma_x$$

*where $\gamma_x \sim D_{\gamma_x} \in \mathcal{D}(0, \sigma^2)$ are uncorrelated random variables $(x \in \mathcal{X})$.*

The general nature of Assumption 5 ensures that the convergence results obtained for the proposed LCLS algorithm hold for a wide range of noise distributions and therefore removes the necessity of having to verify a sub-Gaussian noise assumption when applying LCLS in practice.

Theorem 3.7 formalizes the consistency of the proposed Lipschitz learning framework for the general noisy sampling setting.

**Theorem 3.7** *(General Convergence Rate) If Assumptions (1),(2),(4),(5) (for a given $\eta \in (0, 1]$) hold and the following conditions are verified:*

> *1. $\forall I \in \mathbb{N}$, $\mathcal{H}_I$ is a convex partition of $\mathcal{X}$,*
> *2. $\lim_{I \to \infty} a_I = 0$, $\lim_{I \to \infty} b_I = 0$, $\lim_{I \to \infty} \max_{H \in \mathcal{H}_I} (\Delta_I^H) = 0$,*

*then $\forall \ D_\gamma^n \in \mathcal{D}(0, \sigma^2)$, $f \in C^2(\mathcal{X}, K)$,*

$$\hat{L}_I(f) \xrightarrow[I \to \infty]{\mathbb{P}} L_p(f)$$

*where $L_p(f) = L_p^*(f)$ for $p = 1, 2$, $L_p \geq L_p^*(f)$ for $p > 2$. $\mathbb{P}$ denotes convergence in probability and $(\hat{L}_I(f))_{I \in \mathbb{N}}$ is the sequence of Lipschitz constant estimates generated by the LCLS estimator.*

**Remark 3.8** *(Design Constraints) Condition 2 of Theorem 3.7 specifies the design constraints needed in the construction of the partition sequence $(\mathcal{H}_I)_{I \in \mathbb{N}}$ and the number of sample points $(\{N_I^H\}_{H \in \mathcal{H}_\mathcal{I}})_{I \in \mathbb{N}}$ required per hypercube in order to ensure convergence. In particular:*

> *1. $\lim_{I \to \infty} a_I = 0$ provides the limitations on the shape of the sets in each partition $\mathcal{H}_I$ as $I$ goes to infinity. In particular, as $I \to \infty$, $(\Delta_I^H)^2 << \delta_I^H < \Delta_I^H$.*
> *2. In the noisy sampling setting, $\lim_{I \to \infty} b_I = 0$ specifies a condition on the number of samples needed per hypercube. As $I \to \infty$, $N_I^H >> \frac{|\mathcal{H}_I|}{(\delta_I^H)^2}$. This is made precise in the next section in Remark 3.11.*

*3.* $\lim_{I \to \infty} \max_{H \in \mathcal{H}_I}(\Delta_I^H) = 0$ *ensures that the partitions are increasingly refined.*

In practice, applying the theoretical conditions used in Theorem 3.7 produces an overly conservative estimator in terms of required number of queries made to the oracle – see Section 3.4 for an illustration of the empirical convergence of the LCLS estimator. This is due to the fact that the LCLS estimator makes minimal functional assumptions and therefore has to explore all of $\mathcal{X}$ to generate a precise Lipschitz estimate. In order to avoid this issue, the number of samples per hypercube as measured by $(b_I)_{I \in \mathbb{N}}$ can be set heuristically in order to improve the empirical performance.

In the noiseless sampling setting, the stopping and sampling rules given in Theorem 3.7 and Remark 3.8 can be modified in order to obtain a quicker convergence. This is detailed in the following corollary.

**Corollary 3.9** *(Noiseless Sampling) If Assumptions (1),(2),(4) (for a given $\eta \in (0,1]$) hold, a noiseless oracle $\Omega : \mathcal{X} \to \mathbb{R}$ is available and the following conditions are verified:*

*1. $\forall I \in \mathbb{N}$, $\mathcal{H}_I$ is a convex partition of $\mathcal{X}$,*
*2. $\lim_{I \to \infty} a_I = 0$, $\lim_{I \to \infty} \max_{H \in \mathcal{H}_I}(\Delta_I^H) = 0$,*
*3. $\forall I \in \mathbb{N}, H \in \mathcal{H}_I$, $N_I^H \geq d + 1$,*

*then $f \in C^2(\mathcal{X}, K)$,*

$$\hat{L}_I(f) \xrightarrow[I \to \infty]{} L_p(f)$$

*where $L_p(f) = L_p^*(f)$ for $p = 1, 2$, $L_p(f) \geq L_p^*(f)$ for $p > 2$ and the right arrow denotes deterministic convergence.*

While, the conditions on the design constraints of the partition sequence needed to ensure asymptotic convergence of the LCLS algorithm remain the same as in Theorem 3.7, the sampling conditions specified in Corollary 3.9 imply that a much smaller number of samples are required per hypercube. More precisely, the only sampling condition stated in Corollary 3.9 is related to the minimum number of samples needed to ensure that the local linear regressions computed by the LCLS algorithm are well-defined.

Using the general results developed in this section, we now explore a more specific application to the $[0, M]^d$ input space. Theorem 3.10 provides finite-sample sample complexity bounds for the LCLS in the general noise sampling setting that can be utilised when limited information on the noise distribution is available. As a consequence of Theorem 3.10, sample complexity rates in the noiseless and Gaussian noise setting can also be derived and compared to the lower bounds proposed in Theorem 2.2 and Theorem 2.5. This is discussed in Remark 3.11, Remark 3.12 and Theorem 3.15.

### 3.3 LCLS with Regular Partitions and Sample Complexity Upper Bound

In the previous section, we considered a general form of the LCLS algorithm and stated the conditions on the design constraints of the input partition sequence and the sampling scheme required to ensure convergence. Here, we assume that the input space is the d-dimensional hypercube $[0, M]^d$ and consider the case where every input partition $\mathcal{H}_I$ is a regular hypercube partition of side-length $\frac{M}{I}$. The associated sampling scheme is then defined based on the sampling condition given in Assumption 4 and the desired precision of the Lipschitz constant estimate.

Under these additional constraints, the following finite sample guarantee can be obtained for the LCLS algorithm.

**Theorem 3.10** *(Finite Sample Guarantee) Let $\mathcal{X} := [0, M]^d$ and $(\mathcal{H}_I)_{I \in \mathbb{N}_{>1}}$ denote the regular partition of sub-hypercubes of $\mathcal{X}$ with side-length $\frac{M}{I}$. If Assumptions (2)-(4) (for a given $\eta \in (0,1]$) hold and if $\forall \epsilon > 0$, $\delta \in (0, \frac{1}{2}]$, the LCLS algorithm is set with a hypercube partition indexed by $I \geq \left(C_1(d)\frac{MK}{\sqrt{\eta}\epsilon}\right)$ and with $\forall H \in \mathcal{H}_I \ N_I^H \geq \left(C_2(d,q)\frac{\sigma^2}{\eta\delta}\frac{I^{d+2}}{M^2\epsilon^2}\right)$ for $C_1(d), C_2(d,q) \in \mathbb{R}_+$, then $\forall \ D_\gamma^n \in \mathcal{D}(0, \sigma^2)$:*

$$\inf_{f \in C^2(\mathcal{X},K)} \mathbb{P}(|L_p(f) - \hat{L}_I(f)| \leq \epsilon) \geq 1 - \delta. \tag{1}$$

*where $L_p(f) = L_p^*(f)$ for $p = 1, 2$ and $L_p \geq L_p^*$ for $p > 2$. Here $C_1(d) = 8d^2\sqrt{d}d^{\max\{\frac{1}{q}-\frac{1}{2},0\}}$ and $C_2(d, q) = 2^5 d^{\max\{\frac{2}{q},1\}}$ however these constants have not been optimized.*

The theoretical guarantees of Theorem 3.10 can be extended to include any $\mathcal{X} \subset \mathbb{R}^d$ that satisfies Assumption 1. Indeed, trivially there exists a hypercube $[a, b]^d \subset \mathbb{R}^d$ with $a, b \in \mathbb{R}$ such that $\mathcal{X} \subset [a, b]^d$ which can be partitioned according to the iterative regular hypercube partitioning approach. The partition sequence inputted into the LCLS algorithm then consists of the regular hypercube subsets partitions of $[a, b]^d$ that intersect with $\mathcal{X}$. In this case, under Assumptions (1)-(4), a modified version of Theorem 3.10 holds: the condition on $I$ remains the same, but the lower bound condition on $N_I^H$ can be weakened to become $N_I^H \geq \left(C_2(d, q)\frac{\sigma^2}{\eta\delta}\frac{I^{d+2}-\Gamma}{(b-a)^2\epsilon^2}\right)$, $\forall H \in \mathcal{H}_I, I \in \mathbb{N}$, where $\Gamma = |\{H \in \mathcal{H}_I | H \cap \mathcal{X} = \emptyset\}|$.

Since Theorem 3.10 holds under Assumption 5, i.e. for any $D_\gamma^n \in \mathcal{D}(0, \sigma^2)$, $n \in \mathbb{N}$ and any $f \in C^2([0, M]^d, K)$ it also holds for $\sup_{D_\gamma^n \in \mathcal{D}(0,\sigma^2)} \sup_{f \in C^2([0,M]^d,K)}$. Therefore, using Theorem 3.7, we can obtain the following general sample complexity rate for the LCLS algorithm.

**Remark 3.11** *(Sample Complexity – Noisy) For $p = 1, 2$, assuming that the lower bounds: $I \geq \left(C_1(d)\frac{MK}{\sqrt{\eta}\epsilon}\right)$ and $\forall H \in \mathcal{H}_I$ $N_I^H \geq \left(C_2(d, q)\frac{\sigma^2}{\eta\delta}\frac{I^{d+2}}{M^2\epsilon^2}\right)$ are satisfied with an equality, the total number $n_1$ of points required to ensure $\mathbb{P}(|L_p - \hat{L}_I| \leq \epsilon) \geq 1 - \delta$ is given by*

$$n_1 = |\mathcal{H}_I|N_I = C_2(d, q)\frac{\sigma^2}{\eta\delta}\left(\frac{C_1(d)}{\sqrt{\eta}}\frac{MK}{\epsilon}\right)^{2d+2}\frac{1}{M^2\epsilon^2} = O\left(\left(\frac{MK}{\epsilon}\right)^{2d+2}\frac{1}{M^2\epsilon^2}\right).$$

*This sample complexity differs significantly from the lower bound on the sample complexity derived in Theorem 2.5. This is expected given the more general noise assumptions made in Theorem 3.10.*

By slightly modifying the necessary conditions used in Theorem 3.10, we can also compare the sample complexity of the LCLS algorithm implied by Theorem 3.10 in the noiseless sampling setting to the lower bound on the sample complexity of the noiseless Lipschitz learning problem stated in Theorem 2.2. In order to do so, we define

$$N(I) := \max_{H \in \mathcal{H}_I} \min\{|D_I^H| : D_I^H \text{ contains a disjointed } \delta_I^H\text{-cover of } H\}$$

which is constant $\forall I \in \mathbb{N}$ when $(\mathcal{H}_I)_{I \in \mathbb{N}}$ is defined as a sequence of regular hypercube partitions on $[0, M]^d$. In this case, we remove the dependence on $I$ and write $N := N(I)$. We note that the following two inequalities hold: (1) $\eta \leq \frac{1}{N}$ (tight) and (2) $N < \sqrt{d}^d$ (loose).

**Remark 3.12** *(Sample Complexity – Noiseless) In the case of noiseless sampling, the lower bound on $N_I^H$ stated in Theorem 3.10 can be replaced by condition 3. of Corollary 3.9 and the definition of $N$ given above, i.e. $\forall I \in \mathbb{N}, H \in \mathcal{H}_I, N_I^H = \max(d + 1, N)$. Proceeding as in Remark 3.11, we have in this case:*

$$n_2 = |\mathcal{H}_I|N_I = \max(d + 1, N)\left(C_1(d)\frac{MK}{\sqrt{\eta}\epsilon}\right)^d = O\left(\left(\frac{MK}{\epsilon}\right)^d\right).$$

*This convergence rate corresponds exactly to the lower bound on the noiseless sample complexity rate stated in the Theorem 2.2 and therefore implies that the sample complexity rate $\left(\frac{MK}{\epsilon}\right)^d$ is optimal (up to constant factors dependent on $d$ and $p$) in the sense that it characterises the minimum number of samples that are needed to obtain an $\epsilon$-precise Lipschitz constant estimate for any $f \in C^2(\mathcal{X}, K)$.*

As in Section 2, we can reformulate the sample complexity rates of the LCLS algorithm given in Remarks 3.11 and 3.12 as convergence rates and therefore as upper bounds on the convergence rate of the general Lipschitz learning problem. This is done in the following corollary.

**Corollary 3.13** *(Convergence Rate Comparison)*

1. *(Noiseless) Assume the same setting as Remark 3.12. Then,*

$$\inf_{\hat{L} \in \mathcal{L}_{n,p}(\mathcal{X})} \sup_{f \in C^2(\mathcal{X},K)} |\hat{L}(f) - L_p^*(f)| \leq C(d,p) \frac{MK}{\sqrt[d]{n}}$$

   *where $C(d,p)$ can be determined from Remark 3.12.*

2. *(Noisy) Assume the same setting as Remark 3.11. Then, $\forall$ distribution $D_\gamma^n \in \mathcal{D}(0,\sigma^2)$:*

$$\sup_{\hat{L} \in \mathcal{L}_{n,p}(\mathcal{X})} \inf_{f \in C^2(\mathcal{X},K)} \mathbb{P}_{D_\gamma^n}(|\hat{L}(f) - L_p^*(f)| < C(\sigma^2,\delta,d) \frac{M^{\frac{d}{d+2}} K^{\frac{d+1}{d+2}}}{\sqrt[2d+4]{n}}) \geq 1 - \delta$$

   *where $C(\sigma^2,\delta,d)$ can be determined from Remark 3.11.*

An interesting consequence of Corollary 3.13 is that it provides a way of generating a sequence of feasible[10] Lipschitz constant estimates that converge to the best Lipschitz constant if a potentially loose upper bound on the second degree partial derivatives is known. More precisely, one can consider the Lipschitz constant estimates:

- $\hat{L}_{up}(f) := \hat{L}(f) + C(d,p) \frac{MK}{\sqrt[d]{n}}$ in the noiseless sampling setting
- $\hat{L}_{up}(f) := \hat{L}(f) + C(\sigma^2,\delta,d) \frac{M^{\frac{d}{d+2}} K^{\frac{d+1}{d+2}}}{\sqrt[2d+4]{n}}$ in the general noisy sampling setting

where $\hat{L}(f)$ denotes the Lipschitz constant estimate generated by the LCLS algorithm. Such an approach is useful in practice as Lipschitz constant-based computational frameworks often rely on the assumption that the estimated Lipschitz constant used is feasible. This is briefly discussed further in Section 4 where a direct application of the LCLS algorithm in the context of nonparametric regression for system identification is developed.

**Remark 3.14** *(Knowledge of $K$ and Assumption 2) The theoretical results of the LCLS algorithm of this section have been stated under Assumption 2 and the knowledge of a tight upper bound on the second-order partial derivatives: $K$. This tightness is in fact not necessary and all result pertaining to LCLS hold for any upper bound $K' \geq K$. In this case, the LCLS algorithm simply ensures convergence for a larger class of functions, $C^2(\mathcal{X},K) \subset C^2(\mathcal{X},K')$, then required at a slightly slower rate of convergence. Furthermore, while knowing an upper bound on $K$ is necessary in order for the theoretical properties of the LCLS algorithm to hold, the algorithm can still be implemented heuristically in practice without it.*

We conclude this section by stating the asymptotic sample complexity rates of the LCLS algorithm under Gaussian noise assumptions and providing a finite sample guarantee.

**Theorem 3.15** *(Asymptotic Sample Complexity – Gaussian Noise) Let $M \in \mathbb{R}_+$, $d \in \mathbb{N}$, $p \in \{1,2\}$ and $(\mathcal{H}_I)_{I \in \mathbb{N}}$ denote the regular partition of sub-hypercubes of $\mathcal{X}$ with side-length $\frac{M}{I}$. Assume that Assumption (2) holds, that one has access to a noisy oracle $\tilde{\Omega} : \mathcal{X} \to \mathbb{R}$ as specified in Assumption (3) and that the sample inputs are uniformly and independently sampled on $\mathcal{X}$. Setting the LCLS algorithm with a hypercube partition indexed by $I = \lceil C_1(d) \frac{MK}{\epsilon} \rceil$ for $\epsilon > 0$ (see below for definition of $C_1(d)$), there exists $C > 0$ such that if*

$$n \geq C \frac{\sigma^2 M^d K^{d+2} \log(\frac{MK}{\epsilon})}{\epsilon^{d+4}}$$

*Then,*

$$\lim_{\epsilon \to 0^+} \sup_{f \in C^2(\mathcal{X},K)} \mathbb{P}(Loss(x^{\hat{L}_I(f)}, f) > \epsilon) = 0,$$

*where $x^{\hat{L}_I(f)}$ denotes the the center of the hypercube associated to $argmax_{H \in \mathcal{H}_I} \|\hat{\beta}^H\|_q$ computed in Algorithm 2. Here, $C_1(d) = \frac{16d^2 \sqrt{d} d^{\max\{\frac{1}{q} - \frac{1}{2}, 0\}}}{\sqrt{\eta}}$ with $\eta = \frac{vol(B_1(0))}{2^{3d+1} 3^d} 11$.*

---

[10]I.e. which upper bound the best Lipschitz constant and satisfy the Lipschitz continuity condition.

[11]$vol(B_1(0))$ denotes the d-dimensional unit ball.

The asymptotic sample complexity rates derived in Theorem 3.15 match exactly the rates derived in Theorem 2.5. This implies that $\Theta\left(\frac{\sigma^2 M^d K^{d+2} \log(\frac{MK}{\epsilon})}{\epsilon^{d+4}}\right)$ is the optimal asymptotic sample complexity rate of the Lipschitz learning (search) problem and that the LCLS algorithm is sample optimal in the noisy setting when the noise is assumed to follow a Gaussian distribution. As done in Corollary 2.6, we can modify Theorem 3.15 in order to show that the optimal asymptotic convergence rate is $\Theta\left(MK\left(\frac{\log(MKn)}{nM^4K^2\sigma^2}\right)^{\frac{1}{d+4}}\right)$. We note that Theorem 3.15 holds more generally for any sub-Gaussian noise assumption on the sampling noise. In particular, the same convergence rate holds in the settings where this noise is assumed to be bounded which are often considered in Lipschitz-constant based applications (Canale et al. (2014), Sergeyev et al. (2020)).

Replacing the random uniform sampling assumption of Theorem 3.15 with Assumption 4 as done in Theorem 3.10, a small modification of the proof of Theorem 3.15 yields the following result on the finite-sample guarantees of the LCLS algorithm in the noisy sampling setting with sub-Gaussian noise.

**Corollary 3.16** *(Finite Sample Guarantee – Gaussian Noise) Consider the setting of Theorem 3.15. Assume that Assumptions (2), (4) (for a given $\eta \in (0,1]$) hold, that one has access to a noisy oracle $\tilde{\Omega} : \mathcal{X} \to \mathbb{R}$ as specified in Assumption (3). $\forall \epsilon \in (0, \frac{C_1(d)MK}{3})$, $\delta \in (0, \frac{1}{2})$, setting $I = \left\lceil C_1(d)\frac{MK}{\epsilon}\right\rceil$ and $\forall H \in \mathcal{H}_I$ :*
$$N_I^H \geq C^*(\eta, d)\frac{\sigma^2 K^2}{\epsilon^4}\log\left(\frac{2^{\frac{2}{d}}I}{\log(\frac{1}{1-\delta})^{\frac{1}{d}}}\right) \text{ implies}$$

$$\sup_{f \in C^2(\mathcal{X}, K)} \mathbb{P}(|\hat{L}_I(f) - L_p^*(f)| > \epsilon) \leq \delta.$$

*Here, $C_1$ is defined as in Theorem 3.15 and $\tilde{C}^*(\eta, d) := \frac{2^{10}n_q^2 C_1(d)^2 d^2}{\eta}$ however these constants have not been optimized.*

As done for the general noise setting in Corollary 3.13, convergence rates for the LCLS in the Gaussian noise setting can be obtained by reformulating the finite sample guarantees stated in Corollary 3.16. Then, following the approach described above, a sequence of feasible Lipschitz constant estimates converging to the best Lipschitz constant can be constructed;

- $\hat{L}_{up}^{Gauss} := \hat{L}(f) + C\frac{M^{\frac{d}{d+4}}K^{\frac{d+2}{d+4}}}{\sqrt[d+4]{n\sigma^2\log(MKn)^{-1}}}$ in the Gaussian noise setting

where $\hat{L}(f)$ denotes the LCLS algorithm with the hyperparameters set in Corollary 3.16 and $C \in \mathbb{R}_+$ is a constant that can be computed from $C_1(d), \tilde{C}^*(d, \eta)$. We observe that this sequence of feasible Lipschitz constant estimates converges significantly faster than the one constructed above for the general noisy setting.

### 3.4 Empirical Performance:

The focus so far in this section has been on developing the theoretical properties of the LCLS algorithm. While that discussion is useful in itself as it provides performance guarantees for LCLS as well as upper bounds on the sample complexity of the general Lipschitz learning problem, we are also interested in how the proposed algorithm performs empirically. In particular, we would like to compare the convergence speed of the LCLS algorithm to other theoretically well-behaved methods and to verify whether the theoretical computational advantage of LCLS (see Proposition 3.4) is observed in practice. In this subsection, we investigate these questions by illustrating the convergence rate and computation time of the proposed Lipschitz constant estimation method and comparing it against existing Strongin-based algorithms on a set of test functions with interesting properties in noiseless, bounded noise and unbounded noise sampling settings.

### 3.4.1 Experimental Setup

Table 1 provides an overview of the four test functions that are used in the experiments discussed in this section. The choice of these functions represents different testing points that are of interest: Function (a) reaches the maximum of the normed gradient in a single unique point of the input space, Function (b) is a classical optimisation testing function which we have also defined to have large second degree partial

| **Function** | **Error (Log Scale)** | **Estimation** | **Computation time** |
|---|---|---|---|

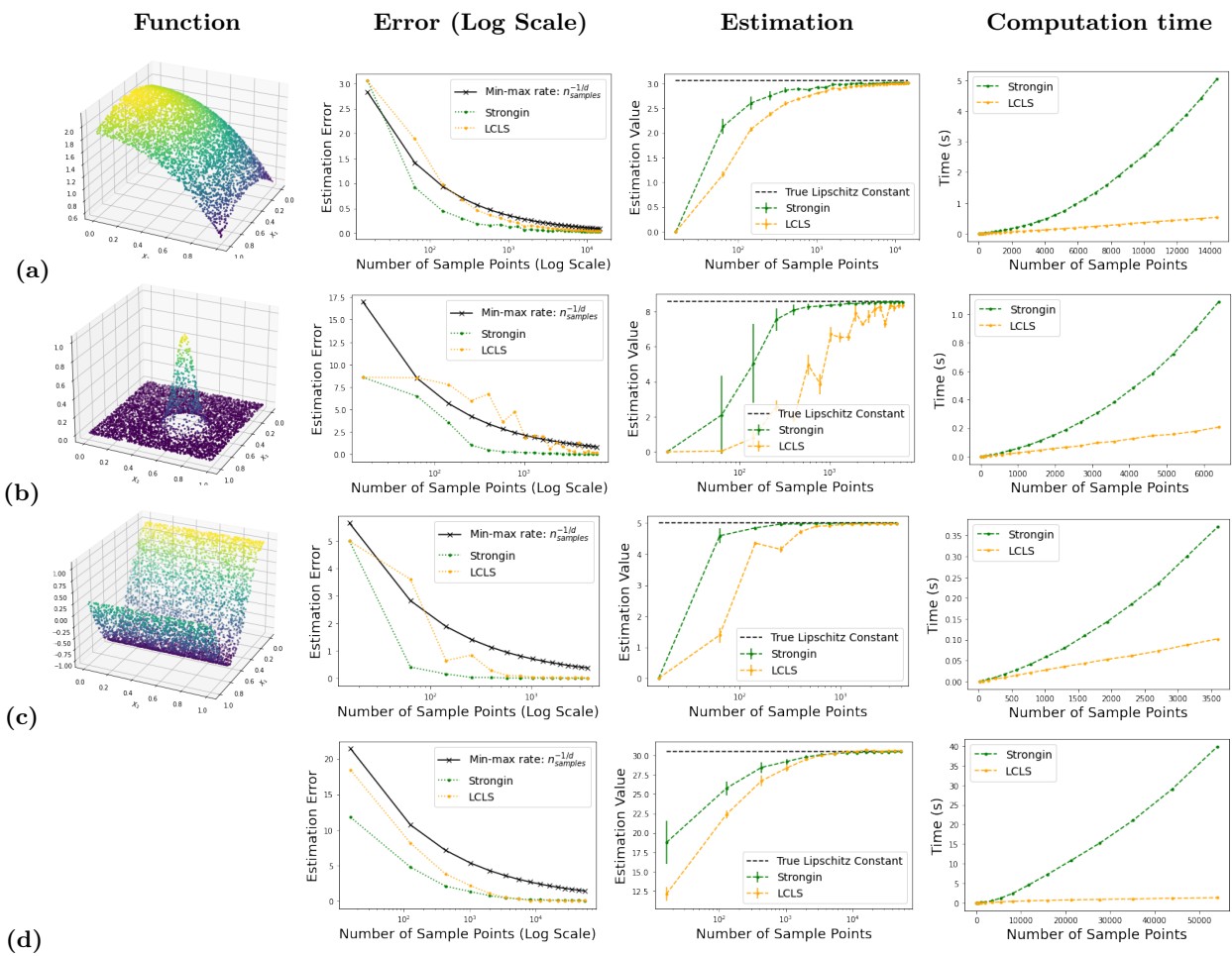

Figure 2: Comparison between the performance of the LCLS algorithm (in orange) and the classical Strongin algorithm (in green) in the noiseless setting. Each row corresponds to a different test function ((a) - (d)) and each column represents a different point of comparison between the two algorithms. From left to right: Column 1: Illustration of the target function where applicable. Column 2: Error of Lipschitz constant estimate - the bound on the sample complexity rate derived in Corollary 2.3 is plotted (in black). Column 3: Behaviour of the sequence of Lipschitz constant estimates. Column 4: Computation time required for each algorithm.

| **Function** | **Expression** | **Lipschitz Const.** | **Key Property** |
|---|---|---|---|
| (a) | See Lemma A.1 | 3.054 | Lipschitz constant reached in a unique point. |
| (b) | $e^{-(x_1^2+x_2^2)}cos(x_1)cos(x_2)$ | 8.5776 | Large second degree partial derivatives (K). |
| (c) | $cos(5x_1)$ | 5 | Simple test function. |
| (d) | See Lemma A.1 | 30.5399 | Higher dimensional input ($\mathbb{R}^3$). |

**Table 1:** Test Functions. Note: Functions (a), (d) are based on the function set utilised in proofs of the sample complexity lower bounds of Section 2.

derivatives, Function (c) is a trigonometric function which provides an illustration of convergence for simple target functions and finally, Function (d) is a higher dimensional version of Function (a) with 3 dimensional inputs. We do not explore higher dimensional versions (>3) of Function (a) as the convergence speed with respect to computation time of the Strongin-based benchmark algorithms is already very slow for Function (d) - see Figure 4 and ensuing discussion.

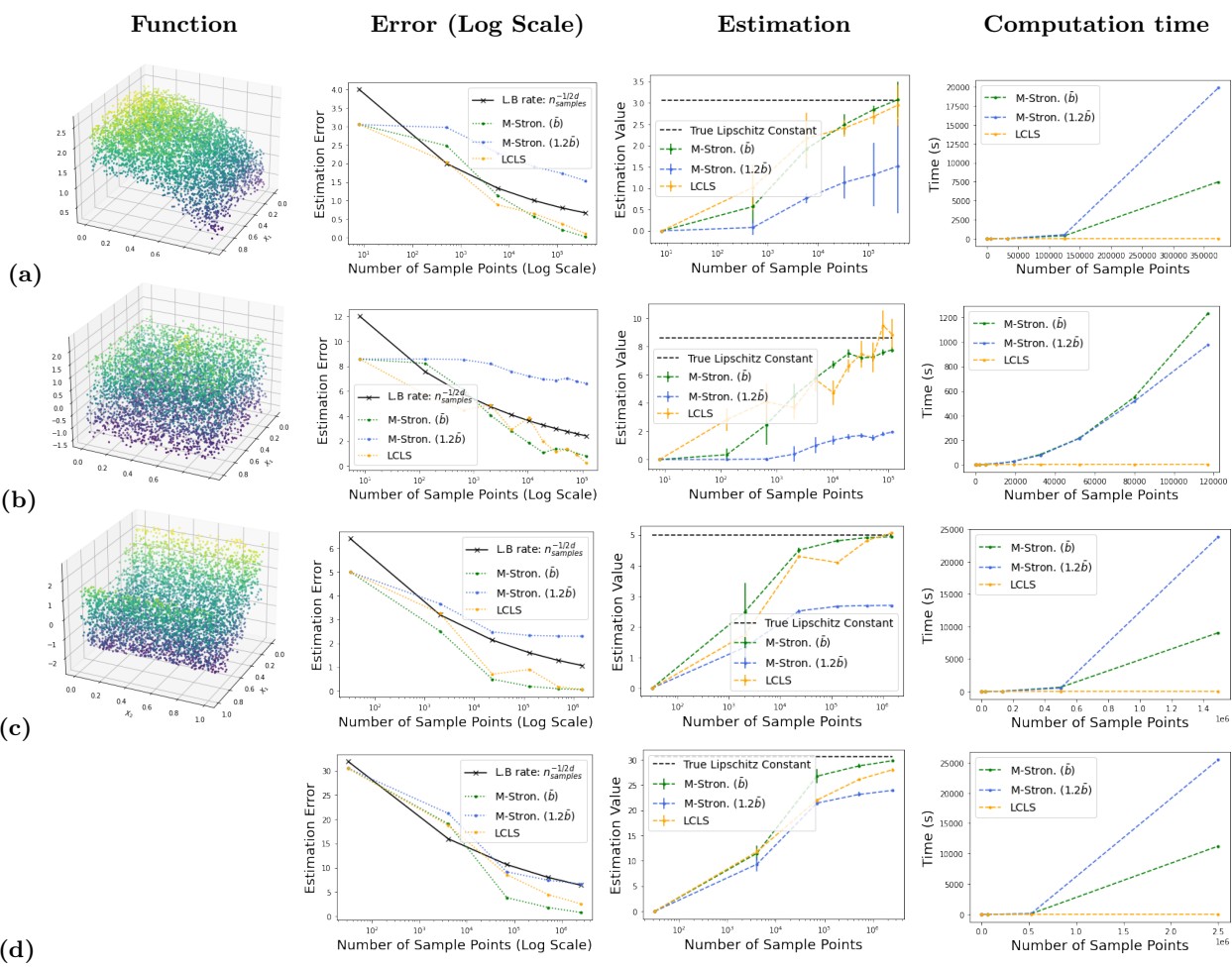

Figure 3: Comparison between the performance of the LCLS algorithm (in orange) and the modified-Strongin algorithm with a correctly (in green) and incorrectly (in blue) specified hyper-parameter in the noisy setting. ach row corresponds to a different test function ((a) - (d)) and each column represents a different point of comparison between the two algorithms. From left to right: Column 1: Illustration of the target function where applicable. Column 2: Error of Lipschitz constant estimate - the bound on the sample complexity rate derived in Corollary 2.6 is plotted (in black). Column 3: Behaviour of the sequence of Lipschitz constant estimates. Column 4: Computation time required for each algorithm.

As benchmarks we utilise the classical Strongin Lipschitz learning algorithm (Strongin (1973)) in the noiseless setting and the popular modified Strongin-based Lipschitz constant estimator in the bounded noise setting (see in particular Novara et al. (2013), Calliess et al. (2020) and Khajenejad et al. (2021) for applications in control problems). We note that this modified Strongin estimator is strongly dependent on a precise estimate of the smallest upper bound of the noise $\bar{b} > 0$ in order to properly specify $\bar{e} \in \mathbb{R}_+$ hyper-parameter. Indeed, if $\bar{e}$ is smaller than $\bar{b}$, then the Lipschitz constant estimates generated by the modified Strongin estimator converge to $+\infty$ as the number of observations increases. In contrast, if $\bar{e}$ is bigger then $\bar{b}$ then the generated Lipschitz constant estimates will converge to an underestimate of $L_p^*(f)$ and never be feasible.

**Benchmarking algorithms:**

- *(Noiseless Setting) Strongin Estimator:*

$$\hat{L} := \max_{i \neq j} \frac{|\tilde{f}_i - \tilde{f}_j|}{\|x_i - x_j\|}.$$

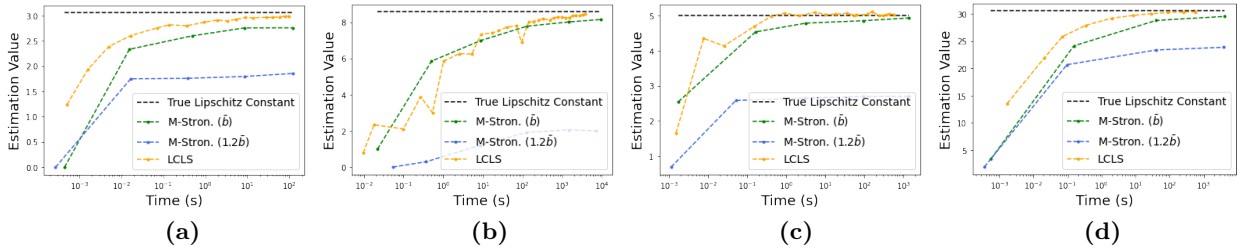

Figure 4: Illustration of convergence speed relative to computation time in the bounded noise setting using the set of test functions given in Table 1. We compare the LCLS algorithm (in orange) and the modified-Strongin algorithm with a correctly (in green) and incorrectly (in blue) specified hyper-parameter. We observe that the LCLS algorithm performs better on all test functions.

- *(Noisy Setting) Modified Strongin Estimator:*

$$\hat{L} := \max_{i \neq j} \frac{|\tilde{f}_i - \tilde{f}_j| - 2\bar{e}}{\|x_i - x_j\|}$$

*where $\bar{e}$ is a hyper-parameter that estimates the tightest upper bound $\bar{b}$ on the noise. We consider modified Strongin Lipschitz estimators with a correctly specified hyper-parameter ($\bar{e} = \bar{b}$) and a hyper-parameter that is slightly larger than the true upper bound ($\bar{e} = 1.2\bar{b}$) as benchmarks.*

### 3.4.2 Discussion

In Figure 2, we illustrate the performance of the LCLS algorithm against the classical Strongin algorithm on the proposed set of test functions. We plot the theoretical lower bounds on the sample complexity rate found in Section 2 in order to provide an intuition for the theoretical bounds. As one would expect, due to the fact that the Strongin algorithm was specifically designed for the noiseless setting, our proposed approach converges more slowly in terms of number of samples on all four test functions. However the difference in convergence speed is not significant and is mitigated by the substantial divergence in computation time. We also note that the plotted sample complexity rate implied by the lower bound of Section 2 does not appear to be tight which is unsurprising as it represents a min-max type bound.

**Remark 3.17** *(Link between the proof of Theorem 3.7 and convergence of LCLS) From the proof of Theorem 3.7, we have that the convergence of the LCLS algorithm depends on two factors:*

1. *($I \in \mathbb{N}$) the diameter of the subsets of the regular partition (upper bounded theoretically using a Taylor expansion).*
2. *($N_I$, $I \in \mathbb{N}$) the number of samples in each subset (upper bounded theoretically using a multivariate Chebyshev inequality).*

*The relation between these two factors is essential for ensuring quick convergence of the LCLS algorithm. In particular, for cases where the second derivatives of the target function $f$ are large, $N_I$ can be decreased and $I$ increased so that the LCLS algrithm considers a finer partition of $\mathcal{X}$ (without having to increase the number of sample points). This type of modification improves the linear approximation of the gradient of $f$ at the cost of increasing the noise in the local least squares estimates (see Function (b) - Easom function in Figure 3).*

In Figure 3, we observe the performance of the LCLS algorithm in the bounded noise setting. Here, the convergence speed relative to sample size of the LCLS method differs more significantly from the convergence speed of the correctly specified modified Strongin benchmark algorithm. This is again unsurprising as the correctly specified modified Strongin algorithm makes use of additional information on the noise distribution and the choice of a uniform noise distribution in the experiment is beneficial towards its convergence speed[12].

---

[12]If a truncated Gaussian distribution had been used instead, the convergence speed of the modified Strongin estimator could have been arbitrarily slowed by decreasing the variance of the distribution.

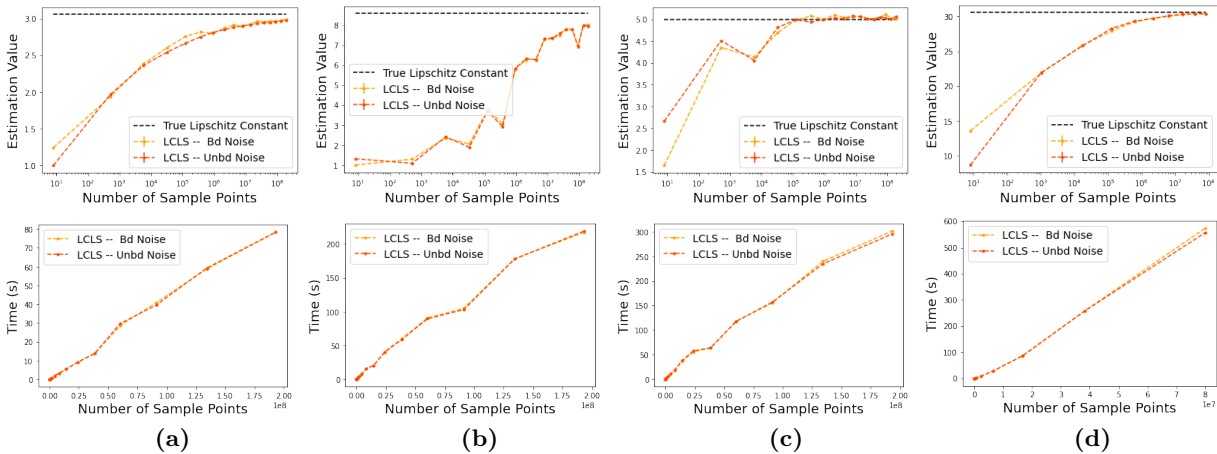

Figure 5: Illustration of the LCLS algorithm in bounded (in light orange) and unbounded noise settings (in dark orange) using the set of test functions given in Table 1. We observe no significant impact of the unboundedness of the noise distribution on the Lipschitz constant estimates produced by LCLS.

We note that the modified Strongin algorithm with a slightly incorrectly specified tightest upper bound fails to show any sign of convergence and that the difference in computation time is more significant than in the noiseless setting. The relation between computational complexity and convergence rate of the LCLS and modified Strongin Lipschitz constant estimators is illustrated more precisely in Figure 4 by plotting the convergence rate relative to computation time. We observe that the LCLS estimator performs better on all functions in the test set despite the fact that the modified Strongin estimator utilises additional information on the noise distribution. In particular, for Function (d) which takes inputs in $\mathbb{R}^3$, the LCLS algorithm needs 8.5 seconds to generate estimates with an estimation error $< 0.5$, while the Strongin approach needs approximately 4000 seconds. This suggests that for application settings with high sampling capacity and time constraints, the LCLS method should be used even when the modified Strongin algorithm can be properly specified.

In Figures 2, 3 and 4, the performance of the LCLS method seems to be more dependent on the value of the maximum of the second degree partial derivatives than the Strongin-based methods. This can be observed by noting the difference in convergence performance for Function (b) relative to the other test functions[13] and is due to the fact that the LCLS algorithm depends on the maximum to define a sufficiently refined partition of the input space in order to "localise" the computations and generate local Lipschitz constant estimates, see Remark 3.8 and Theorem 3.10 for a precise characterisation of the relationship. In some sense, the stronger dependency on the maximum of the second degree partial derivatives of the target function can be interpreted as a trade-off for the improvement in computation time obtained by the LCLS algorithm.

The last illustration, provided in Figure 5, shows the convergence and computation time of the LCLS algorithm in the unbounded noise setting. We do not provide a benchmark as no alternative theoretically backed approaches exist in this setting: the approaches of Beliakov (2005) and Calliess (2017) could be used but do not have any asymptotic convergence guarantees. Instead, we compare the convergence rate to the one obtained by LCLS in the bounded noise setting and observe the fact that no significant performance loss has occurred when the noise is unbounded on any of the test functions.

We conclude this section by remarking that throughout the experiments, our proposed method has been relatively unaffected by the changes in sample setting assumptions and can be used with minimal fine-tuning. Indeed, only the relation between the number of samples in each subset and the diameter of each of these subsets needs to be modified (see Remark 3.17). This relation can be set in a theoretically principled manner by considering the results given in Remarks 3.11, 3.12 and Corollary 3.16 or treated as a hyper-parameter and set more heuristically. The flexibility of the LCLS algorithm is in contrast to existing asymptotically

---

[13]See also Figure 7 and ensuing discussion.

consistent Lipschitz learning algorithms such as the benchmark approaches used in this section which either only consider noiseless sampling settings or require prior knowledge of the noise distribution in order to be applied.

# 4 Connections to Machine Learning and Related Fields

The theoretical results derived in Section 2 are fundamental in nature. They can be used as a benchmark when developing novel Lipschitz constant estimators or more generally to provide a better theoretical understanding of algorithms that depend explicitly on Lipschitz constant estimates of an underlying target function. Utilising Corollaries 2.3 and 2.6 the worst-case estimation errors of Lipschitz constant estimation can be better understood and their negative impact on overall performance mitigated. This is particularly important as Lipschitz constant dependent algorithms often rely on heuristic or experimental arguments which might not always hold in practice to justify the Lipschitz constant estimation step.

In some settings, the LCLS estimator developed in Section 3 can be directly applied to improve existing computational frameworks in which case the finite sample guarantees derived in Theorem 3.10 and Corollary 3.13 can be used. In particular, when a (loose) bound on the second order partial derivatives of the target function is known, a sequence of feasible Lipschitz constants converging to the best Lispchitz constant at a known convergence speed is obtainable. Unfortunately, while this approach is possible in all the sampling set-ups considered in this paper, the convergence rates obtained for the noisy sampling set-up (see Corollaries 3.13 and 3.16) can be too slow to be useful in some practical applications. In these cases, the LCLS estimator can be applied directly to estimate the Lipschitz constant without feasibility guarantees.

In the section below, we briefly discuss how the results and algorithms derived in this paper can be used in the fields of system identification and global optimisation.

## 4.1 Global Optimisation:

A major subfield of global optimisation research focuses on sequential search methods that explicitly utilise the Lipschitz constant of the target function to remove large sets in the search space and enhance the efficiency of exploration (Shubert (1972), Mladineo (1986)). As a good estimate of the Lipschitz constant is not always available in practice, work arounds must be be found (Jones et al. (1993)). In particular, a number of these optimisation frameworks make use of a Lipschitz constant estimator (Kvasov and Sergeyev (2012) and references therein, D'Agostino (2022)). The computation of these estimates is generally done heuristically without convergence analysis or error-certificate of the Lipschitz constant estimates. Therefore, the minimax bounds derived in Theorem 2.2 of Section 2 provide a context for the expected performance of these methods. More precisely, given recent work by Malherbe and Vayatis (2017) and Bachoc et al. (2021) which derives optimal sample complexity rates for Lipschitz Optimisation when a Lipschitz constant is known, it becomes possible to derive a lower bound on the sample complexity of adaptive Lipschitz Optimisation algorithms that separate the optimisation procedure and the Lipschitz constant estimation. We derive such a lower bound below as an example of how this can be done.

Following the set-up of certified online learning algorithms described in Bachoc et al. (2021), we assume that we have access to a black-box target function $f$ that can be queried to obtain noiseless observations. The goal of certified global optimisation is to design an algorithm that systematically queries $f$ in order to generate an output sequence $((x_n, f(x_n^*), \zeta_n))_{n \in \mathbb{N}}$ where $x_n$ is the n-th query point, $f(x_n^*)$ is the generated estimate of $\max_{x \in \mathcal{X}} f(x)$ after $n$ queries and $\zeta_n \geq 0$ is an error certificate that guarantees: $\max_{x \in \mathcal{X}} f(x) - f(x_n^*) \leq \zeta_n$.

Given an accuracy $\epsilon \in \mathbb{R}_+$, we can then define the sample complexity [14] $N(A, f, \epsilon)$ of a certified global optimisation algorithm $A$ as the smallest number of queries needed in order to obtain an error certificate smaller than $\epsilon$ for all $f$ belonging to a function class $\mathcal{C}$, or in other words:

$$N(A, f, \epsilon) := \min\{n \in \mathbb{N} \cup \{+\infty\} | \zeta_n < \epsilon\}.$$

---

[14]Note: this differs slightly from the definition used in Bachoc et al. (2021).

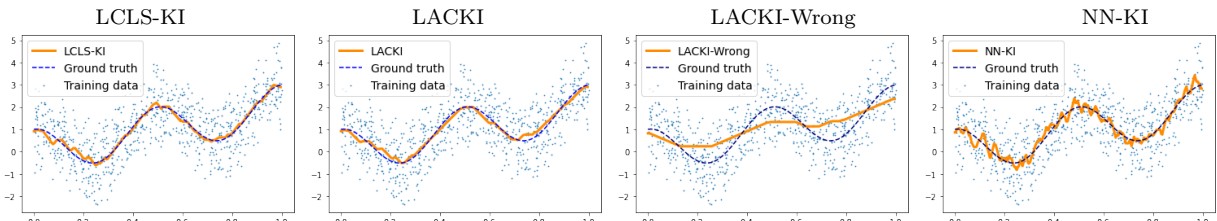

Figure 6: Illustration of several nonparametric methods applied to noisy data. The target function is $f : x \mapsto cos(4\pi x) + 2x$ and the noise is distributed according to a truncated Gaussian distribution (std: 1, upper/lower bound: $-2/2$). The predictions of the trained methods are plotted in orange and the training data in light blue (800 observations). From left to write: LCLS-KI: Kinky inference using the Lipschitz constant estimate generated by the LCLS algorithm. LACKI: Adaptive Kinky Inference proposed by Calliess et al. (2020) with correctly set error bounds. LACKI-wrong: LACKI method with error bounds set at the wrong observational error bound (i.e. at $1.3\times$ the true error bound). NN-KI: Kinky inference method using the Lipschitz constant of a fitted Neural Network model with sigmoid activation as proposed by Milanese and Novara (2004).

Utilising this theoretical set-up, we can then combine the theoretical results of Bachoc et al. (2021) with Corollary 2.3 in order to obtain the following statement on the worst case lower sample complexity bound of the adaptive Lipschitz optimisation problem.

**Proposition 4.1** *(Sample Complexity - Adaptive Lipschitz Optimisation) Assume that $\mathcal{X}$ is the hypercube and consider the set $\mathcal{A}$ of adaptive Lipschitz optimisation algorithms which combine classical Lipschitz optimisation methods with a separable[15] feasible Lipschitz constant estimator. There exists constants $C_1, C_2 > 0$ such that $\forall L^* \geq 0$, $A \in \mathcal{A}$ and $\epsilon \in (0, \epsilon_0)$ where $\epsilon_0 \in (0, 2^{d-1}ML^*)$ :*

$$\sup_{f \in C^2(\mathcal{X},K) \cap \mathcal{F}_p(L^*)} N(A, f, \epsilon)$$

$$\geq C_1 \alpha_d(M, L^*, K)((1 + C_2 \max(\min(\frac{3}{C_2}, \frac{1}{\lceil (1 + \log_2(\frac{\epsilon_0}{\epsilon}) \rceil)^{\frac{1}{d}} + \beta(L^*, K, \epsilon)} ), \frac{\gamma_d(M, L^*, \epsilon)}{\beta(L^*, K, \epsilon)}))^{\frac{1}{2}} - 1)^d \tag{2}$$

*where $m := \max_{y \in \mathcal{X}} f(y)$, $V_{\mathcal{X}} = M^d$, $K$ is as defined in Assumption 2 and*

- $\alpha_d(M, L^*, K) := (\frac{ML^*}{K})^d$ *represents the problem's general dependency on the input space size, true best Lipschitz constant of the target function and desired precision of the optimisation algorithm and second degree partial derivatives.*
- $\beta(L^*, K, \epsilon) := (1 + \lceil \log_2(\frac{\epsilon_0}{\epsilon}) \rceil)^{1/d} \frac{K\epsilon}{L^{*2}}$ *represents the dependency on the true best Lipschitz constant of the target function, the second degree partial derivatives and and desired precision of the optimisation algorithm.*
- $\gamma_d(M, L^*, \epsilon) := \sqrt[d]{\frac{L^*\epsilon(d-1)}{M}}$ *represents the dependency on the input space size, true best Lipschitz constant of the target function and desired precision of the optimisation algorithm and second degree partial derivatives.*

To our knowledge, (2) is the first lower bound on the sample complexity of adaptive Lipschitz optimisation frameworks (see Malherbe and Vayatis (2017) for a possible sample complexity upper bound provided by the adaLIPO algorithm). It depends on the input space, desired precision and upper bounds on the first two orders of differentiation of $f$. The structure of the proof of Proposition 4.1 as well as the two terms contained in the max expression of the lower bound can be interpreted as a comparison between the sample complexity arising from the optimisation procedure and the one arising from the Lipschitz constant estimation. In particular, $\frac{\gamma_d(L^*, M, \epsilon)}{\beta(L^*, K, \epsilon)}$ is computed by considering the subset of linear functions of $C^2(\mathcal{X}, K) \cap \mathcal{F}_p(L^*)$ which is trivial to optimise in the case where the Lipschitz constant is known but becomes complicated to certify if the Lipschitz constant estimation is difficult.

---

[15]In other words, only knowledge of the Lipschitz constant estimate is used in the optimisation part of the algorithm.

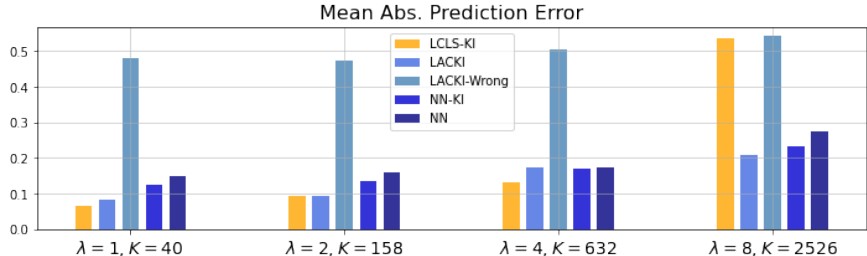

Figure 7: Mean absolute error of the nonparametric methods discussed in Figure 6 and the neural network utilised to estimate the Lipschitz constant in the NN-KI method. The target functions are given by $f_\lambda : x \mapsto cos(2\lambda\pi x) + \lambda x$, for $\lambda = 1, 2, 4, 10$ and associated maximum second derivative: $K := \max_{x\in\mathcal{X}} f_\lambda''(x) = 40, 158, 632, 2526$. The values shown in the plot are computed on a test set containing 500 independently sampled observations.

Unfortunately, the proposed bound is loose as the sampling scheme for the Lipschitz optimisation algorithm can differ significantly from the sampling scheme of Lipschitz constant estimator and is of moderate interest as it only considers a subset of adaptive Lipschitz optimisation algorithms. It does however provide an example of how the lower bounds derived in Section 2 of this paper can be utilised to theoretically analyse existing computational frameworks that rely on Lipschitz learning and future work will consider refining the lower bound given in (2).

Finally, we note that the lower bounds derived in Section 2 can also be considered in the application of recently proposed batch Bayesian optimisation frameworks (González et al. (2016), Alvi et al. (2019)). Indeed, while these methods provide interesting experimental results, the convergence bound stated in Corollary 2.3 shows that in the worst case the Lipschitz constant estimate generated from the fitted Gaussian Process can differ significantly from the true Lipschitz constant - severely impacting the performance of the algorithm in high dimensional settings. At best, the Lipschitz constant estimate used in these papers: $\max_{x\in\mathcal{X}} \|\mu_\nabla(x)\|$ must be replaced by $\max_{x\in\mathcal{X}} \|\mu_\nabla(x)\| + C(d,p)\frac{MK}{\sqrt[d]{n}}$ in order to ensure that the estimated value is a feasible Lipschitz constant. Here $\mu_\nabla$ denotes the mean function of the gradient function estimate associated to the fitted GP which can be computed efficiently using the covariance function of the GP.

## 4.2   Nonparametric Regression for System Identification:

A popular system identification method in control settings known as *Nonlinear Set Membership* (Milanese and Novara (2004)) and also referred to as *Lipschitz interpolation* (Beliakov (2006)) or *Kinky inference* (Calliess et al. (2020)) by subsequent authors, explicitly utilises the Lipschitz constant of an underlying Lipschitz continuous target function to define the smallest set of all possible systems that is consistent with the observed data and to provide optimal[16] point estimates. In the relevant literature, a number of approaches have been used to estimate the Lipschitz constant however these either rely on heuristic estimation (Milanese and Novara (2004), Calliess (2017)) or on knowledge of often unavailable hyper-parameters such as tight bounds on the noise (Novara et al. (2013),Calliess et al. (2020)) which underestimate the true Lipschitz constant. Utilising the LCLS algorithm developed in Section 3 would therefore be an interesting alternative approach to constructing an adaptive Nonlinear Set Membership framework. As noted at the beginning of the section, we directly utilise the Lipschitz estimate produced by the LCLS estimator as the worst-case error guarantees stated in Corollaries 3.13 and 3.16 are too conservative to be useful in the considered use case.

In Figure 6, we illustrate the performance of a hybrid LCLS - Kinky Inference method in comparison to other nonparametric methods that depend explicitly on the Lipschitz constant of the target function. The variation of the plotted nonparametric predictors is a direct function of the Lipschitz constant estimated from the data – when the Lipschitz constant estimate underestimates the true Lipschitz constant flatter prediction curves that do not fully capture the nonlinearity of the target function are produced while Lipschitz constant estimates that overestimate the true Lipschitz constant produce overly input sensitive predictions. In fact,

---

[16]See Milanese and Novara (2004).

the kinky inference framework converges to a nearest neighbour estimator as the Lispchitz constant goes to infinity (Maddalena and Jones (2020)).

In Figure 7, we observe that under the truncated Gaussian noise assumptions, the proposed LCLS-KI approach seems to perform best in comparison to the other nonparametric methods as long as the bound on the second derivative $K$ (see Assumption 2) is not too large relative to the number of observations in the training data. As noted in Section 3.4, this is due to the fact that the LCLS algorithm is more dependent on $K$ than other classes of Lipschitz learning algorithms and can significantly underestimate the true Lipschitz constant when $K$ is too large. Therefore, when the second order derivatives are moderate and upper bounds on the noise on the noise are not precisely known, the LCLS-KI algorithm provides an interesting alternative to existing nonlinear set membership/Lipschitz interpolation methods. Applications of LCLS-KI to the common-use case of such methods, e.g. in learning-based model predictive control (Canale et al. (2014), Limon et al. (2017)), could be pursued in future work.

## 5  Conclusions

In this work, we have established precise lower and upper bounds on the sample complexity of the estimation of Lipschitz constants under minimal parametric constraints on the target function. Instead, our bounds rely on the assumption of $C^2$ regularity of the target function which, given a compact input space, implies the existence of an upper bound on the second degree partial derivatives (this type of assumption is unavoidable as if the second degree partial derivatives are not assumed bounded, then the sample complexity can not be guaranteed to be finite and any theoretical characterisation of the general Lipschitz learning problem is trivial). The obtained bounds on the sample complexity are shown to be optimal in the noiseless sampling setting and in the noisy sampling setting for a slightly modified but generally equivalent version of the problem under Gaussian noise. These results can be used to provide a theoretical baseline for the Lipschitz learning problem and to help drive the design of future black-box Lipschitz constant estimators.

In order to derive the upper bound on the sample complexity, we have proposed a new algorithm for Lipschitz learning based on local least squares regression that is sample-optimal in the noiseless setting and in the noisy setting with Gaussian noise. We have thoroughly investigated the theoretical properties of this algorithm showing asymptotic consistency, guarantees on finite sample behaviour and computational complexity in both noiseless and general noisy sampling settings.

A series of brief empirical experiments illustrate how these theoretical results could translate into practice and how the LCLS algorithm can compare to existing classical Lipschitz constant estimators. The proposed method provides a suitable solution for Lipschitz constant estimation when a theoretically principled and computationally flexible approach is desired.

Forthcoming work on LCLS will focus on extending the algorithm to recursively compute local Lipschitz constants on observed data and to provide theoretical guarantees on this extension. In addition, future work on theoretical Lipschitz learning could look to improve the sample complexity bound in the noisy setting derived in Section 2 under stronger assumptions on the target function and by restricting the class of Lipschitz learning algorithms.

## 6  Acknowledgements

We gratefully acknowledge support from the Oxford-Man Institute of Quantitative Finance (OMI) and thank the reviewers whose feedback improved both the quality and the clarity of this paper.

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

## Appendix: Contents

## A   Proofs: Lower bounds on Sample Complexity

**Lemma A.1** *(Properties of $\mathcal{F}$) For $C_1, C_2 \in \mathbb{R}$, define the function $g_0 : \mathbb{R}^d \to \mathbb{R}$,*

$$g_0(x) = \begin{cases} C_1 e^{-\frac{1}{1-C_2 \sum_{j=1}^d x_j^2}} & \text{if } C_2 \sum_{j=1}^d x_j^2 < 1 \\ 0 & \text{otherwise.} \end{cases} \quad .$$

*The following properties of $g_0$ can be shown;*

1. $\max_{x \in \mathbb{R}^d} \|\nabla g_0(x)\|_2 \approx 0.8 C_1 \sqrt{C_2}$

2. $\max_{x \in \mathbb{R}^d, i,j \in \{1,...,d\}} \left| \frac{\partial^2 g_0}{\partial x_i x_j}(x) \right| \approx 7.75 C_1 C_2.$

**Proof**   Let $g_0$ be as described above. It follows from construction that $g_0$ is a radial function and that there exists $u : [0, +\infty) \to \mathbb{R}$, such that $\forall x \in \mathbb{R}^d$, $u(\sum_{i=1}^d x_i^2) = g_0(x)$ (and in other terms, $\forall r \in [0, +\infty)$, $u(r) := g_0(\sqrt{r}, 0, ..., 0)$ ). We can compute the maximum magnitude (in $\|.\|_2$) of the gradient of $g_0$ as follows;

$$\max_{x \in \mathbb{R}^d} \|\nabla g_0(x)\|_2 = \max_{x \in \mathbb{R}^d} \| \left( 2x_1 u'\left(\sum_{i=1}^d x_i^2\right), 2x_2 u'\left(\sum_{i=1}^d x_i^2\right), ... \right) \|_2 = \max_{x \in \mathbb{R}^d} \left\{ \left| 2u'\left(\sum_{i=1}^d x_i^2\right) \right| \|x\|_2 \right\}$$

$$= \max_{r \in \mathbb{R}_+} |2u'(r^2) r| = \max_{0 \leq r \leq \frac{1}{\sqrt{C_2}}} \left\{ 2C_1 C_2 r \frac{e^{-\frac{1}{1-C_2 r^2}}}{(1-C_2 r^2)^2} \right\} = C_1 \sqrt{C_2} \max_{0 \leq r \leq 1} \left\{ 2r \frac{e^{-\frac{1}{1-r^2}}}{(1-r^2)^2} \right\}.$$

Computing $\max_{0 \leq r \leq 1}\{2r \frac{e^{-\frac{1}{1-r^2}}}{(1-r^2)^2}\}$ gives $\max_{0 \leq r \leq 1}\{2r \frac{e^{-\frac{1}{1-r^2}}}{(1-r^2)^2}\} = \frac{6\sqrt[4]{3^3} e^{-\frac{1}{1-\frac{1}{\sqrt{3}}}}}{(\sqrt{3}-3)^2}$.   Since $g_0$ is continuously differentiable and the support of $\nabla g_0$ is compact, we have that there exists $x^* \in \mathbb{R}^d$ such that $\|\nabla g_0(x^*)\|_2 = C_1 \sqrt{C_2} \frac{6\sqrt[4]{3^3}}{(\sqrt{3}-3)^2} e^{-\frac{\sqrt{3}}{\sqrt{3}-1}} \approx 0.8 C_1 \sqrt{C_2}$.

Similarly, we have that for $i \in \{1, ..., d\}$, $x \in \mathbb{R}^d$;

$$\frac{\partial^2 g_0}{\partial x_i^2}(x) = 2u'\left(\sum_{i=1}^d x_i^2\right) + 4x_i^2 u''\left(\sum_{i=1}^d x_i^2\right).$$

Here it is clear that for $x^* \in \text{argmax}_{x \in \mathbb{R}^d} |\frac{\partial^2 g_0}{\partial x_i^2}(x)|$ either (1) $x_i^* = 0$ or (2) $x_j^* = 0$, $\forall i \neq j$. In the first case; we can compute $\max_{r \in \mathbb{R}_+} |2u'(r)| = \frac{8C_1 C_2}{e^2} \approx 1.08 C_1 C_2$. In the second case, setting $x = re_i$, we consider the computation of $\max_{r \in \mathbb{R}_+} |2u'(r^2) + 4r^2 u''(r^2)|$. We have

$$2u'(r^2) + 4r^2 u''(r^2) = 2C_1 C_2 e^{-\frac{1}{1-r^2}} \frac{3C_2^2 r^4 - 1}{(1-C_2 r^2)^4}$$

and can compute

$$\max_{r \in \mathbb{R}_+} |2u'(r^2) + 4r^2 u''(r^2)| = C_1 C_2 \max_{r \in \mathbb{R}_+} |2e^{-\frac{1}{1-r^2}} \frac{3C_2^2 r^4 - 1}{(1-C_2 r^2)^4}| \approx 7.75 C_1 C_2.$$

Therefore, we have $\max_{x \in \mathbb{R}^d} |\frac{\partial^2 g_0}{\partial x_i^2}(x)| \approx 7.75 C_1 C_2$. Finally, we check $\forall i \neq j \in \{1, ..., d\}$, $\max_{x \in \mathbb{R}^d} |\frac{\partial^2 g_0}{\partial x_i x_j}(x)| = \max_{x \in \mathbb{R}^d} |4x_i x_j u''(\sum_{i=1}^d x_i^2)|$. Clearly, we can set $x = re_i + se_j$ for $r, s \in \mathbb{R}_+$. Computing this quantity gives;

$$\max_{x \in \mathbb{R}^d} \left| 4x_i x_j u''(\sum_{i=1}^d x_i^2) \right| = C_1 C_2^2 \max_{(r,s) \in \mathbb{R}_+ \times \mathbb{R}_+} \left| \frac{4rs e^{-\frac{1}{1-C_2(r^2+s^2)}}}{(1-C_2(r^2+s^2))^3} \right| = C_1 C_2 \frac{8\sqrt{2} e^{-2-\sqrt{2}}}{(\sqrt{2}-2)^3}.$$

We obtain $\forall i \neq j \in \{1, ..., d\}$, $\max_{x \in \mathbb{R}^d} |\frac{\partial^2 g_0}{\partial x_i x_j}(x)| \approx 1.85 C_1 C_2 \leq \max_{x \in \mathbb{R}^d} |\frac{\partial^2 g_0}{\partial x_i^2}(x)|$.

∎

**Proof of Theorem 2.2 (Sample Complexity Bound – Noiseless).**
Let $p = 2$. If we can show that there exists a set $\mathcal{F} \subset C^2(\mathcal{X}, K) \cap \mathcal{F}_p(L^*)$ of functions that can be constructed such that

$$\forall \hat{L} \in \mathcal{L}_{n,p}(\mathcal{X}), \sup_{f \in \mathcal{F}} |\hat{L}(f) - L_p^*(f)| > \epsilon$$

when $n < \left(C(d, p) \frac{MK}{\epsilon}\right)^d$, then Theorem 2.2 follows directly. This expression can be simplified to the equivalent statement;

$$\forall \hat{L} \in \mathcal{L}_{n,p}(\mathcal{X}), \exists f \in \mathcal{F} \text{ such that } |\hat{L}(f) - L_p^*(f)| > \epsilon.$$

Consider the following functional family. For $C_1 \in \mathbb{R}, C_2 \in \mathbb{R}_+$,

$$\mathcal{F}_0(C_1, C_2) := \left\{ g_z : \mathbb{R}^d \to \mathbb{R} | z \in \mathcal{X}, g_z(x) = \begin{cases} C_1 e^{-\frac{1}{1 - C_2 \sum_{j=1}^d (x_j - z_j)^2}} & \text{if } C_2 \sum_{j=1}^d (x_j - z_j)^2 < 1 \\ 0 \text{ otherwise.} \end{cases} \right\}.$$

For any $L^*$, we can consider the family $\mathcal{F}_{L^*}(C_1, C_2)$ by adding a linear component, e.g. $L^* x_1$ to $g_z \in \mathcal{F}_0(C_1, C_2)$. In this case, we have by the construction of $\mathcal{F}_0(C_1, C_2)$ that for all $g_z^0 \in \mathcal{F}_0(C_1, C_2)$ with support in $\mathcal{X}$ and $g_z^{L^*} \in \mathcal{F}_{L^*}(C_1, C_2)$, $z \in \mathcal{X}$, $\max_{x \in \mathbb{R}^d} \|\nabla g_z^{L^*}(x)\|_2 = \max_{x \in \mathbb{R}^d} \|\nabla g_z^0(x)\|_2 + L^*$ and $\max_{x \in \mathbb{R}^d, i,j \in \{1,...,d\}} \left| \frac{\partial^2 g_z^{L^*}(x)}{\partial x_i x_j} \right| = \max_{x \in \mathbb{R}^d, i,j \in \{1,...,d\}} \left| \frac{\partial^2 g_z^0(x)}{\partial x_i x_j} \right|$. The second relation is obvious while the first follows from the fact that all $g_z^0 \in \mathcal{F}(C_1, C_2)$ are radial functions which implies that all gradients of $g_z^0$ are either pointing towards or away from $z$ with equal magnitude along any hypersphere of fixed radius. Therefore, by the properties $g_z^0$ shown in Lemma A.1, for any choice of linear component $l : \mathbb{R}^d \to \mathbb{R}$, $l(x) = ax + b$ for $a \in \mathbb{R}^d, b \in \mathbb{R}$ such that $\|a\|_2 = L^*$, there exists $x^* \in \mathcal{X}$ such that $\max_{x \in \mathbb{R}^d} \|\nabla g_z^0(x)\|_2 = \|\nabla g_z^0(x^*)\|_2$ and $\nabla g_z^0(x^*)$ and $a$ have the same direction (if they have opposite direction it suffices to take $x^*$ that is diametrically opposed on the same hypersphere). With this construction,

$$\max_{x \in \mathbb{R}^d} \|\nabla g_z^{L^*}(x)\|_2 = \|\nabla g_z^0(x^*) + a\|_2 = \|\nabla g_z^0(x)\| + \|a\|_2 = \max_{x \in \mathbb{R}^d} \|\nabla g_z^{L^*}(x)\| + L^*.$$

We can therefore restrict our proof to considering the case where $L^* = 0$.

In the first part of the proof, we will show that for carefully selected values $C_1^*, C_2^* \in \mathbb{R}$, $\mathcal{X}$ contains $\sim \left(\frac{MK}{\epsilon}\right)^d$ disjointed $\|.\|_2$-hyperspheres $\mathcal{B} := \{B_i\}_{i \in \{1,...,\left(\frac{MK}{\epsilon}\right)^d\}}$ of radius $\frac{1}{\sqrt{C_2^*}}$ such that; $\forall B_i, B_j \in \mathcal{B}$, $B_i \subset \mathcal{X}$, $B_i \cap B_j = \emptyset$ if $i \neq j$ and a set $\mathcal{F} \subset \mathcal{F}_0$ of associated functions with the following properties; $\forall g_{\bar{z}_i} \in \mathcal{F}$ associated to $B_i \in \mathcal{B}$,

1. $supp(g_{\bar{z}_i}) = B_i$,

2. $\max_{x \in \mathcal{X}} \|\nabla g_{\bar{z}_i}(x)\|_2 \geq 2\epsilon$ $(+L^*$ if $L^* \neq 0)$ ,

3. $\forall k, j \in \{1, ..., d\}$, $\max_{x \in \mathcal{X}} \left| \frac{\partial^2 g_{\bar{z}_i}}{\partial x_k x_j}(x) \right| \leq K$.

To do so we consider the gradient and second order partial derivatives of the functions in $\mathcal{F}$. Let $g \in \mathcal{F}$, applying Lemma A.1, we have :

1. $\max_{x \in \mathbb{R}^d} \|\nabla g(x)\|_2 \approx 0.8 C_1 \sqrt{C_2}$ $(+L^*$ if $L^* \neq 0)$

2. $\max_{x \in \mathbb{R}^d, i,j \in \{1,...,d\}} \left| \frac{\partial^2 g(x)}{\partial x_i x_j} \right| \approx 7.75 C_1 C_2$.

Using these values, we can define the values of $C_1^*$ and $C_2^*$ discussed earlier in the proof. Firstly, in order to have $g \in C^2(\mathbb{R}^d, K)$, we need $\max_{x \in \mathbb{R}^d, i,j \in \{1,...,d\}} \left| \frac{\partial^2 g}{\partial x_i x_j}(x) \right| \leq K$. This implies the relation $C_1 = \frac{K}{7.75 C_2}$. Secondly, we set $C_2$ such that $\max_{x \in \mathcal{X}} \|\nabla g(x)\|_2 = 0.8 C_1 \sqrt{C_2} = 2\epsilon$. Plugging in the relation for $C_1$ given above;

$$\frac{0.1K}{\sqrt{C_2}} = 2\epsilon \Leftrightarrow \left(\frac{K}{20\epsilon}\right)^2 = C_2^* \text{ and } C_1^* = \frac{51\epsilon^2}{K}.$$

Setting $l = \frac{1}{\sqrt{C_2^*}} = \frac{20\epsilon}{K}$ we have

$$supp(g)^{17} := \left\{ x \in \mathbb{R}^d | C_2^* \sum_{i=1}^d x_i^2 < 1 \right\} = B_l(c)$$

where $B_l(z)$ denotes the d-dimensional ball of radius $l$ defined with respect to $\|.\|_2$ and centered in $c \in \mathcal{X}$. The last step before defining $\mathcal{F}$ is to count how many sphere of radius $l$ can fit[18] in $\mathcal{X} = [0, M]^d$. Here, we use a lower bound that is obtained by considering the regular hypercube partition of $\mathcal{X}$ of side-length $\tilde{l}$, defined by; $N := \lfloor \frac{M}{l} \rfloor$ and $\tilde{l} = \frac{M}{N}$. Let $\mathcal{B}$ denote the set of balls of radius $l$ that can be inscribed in a subset belonging to the hypercube partition of $\mathcal{X}$. Then, for all $B_i, B_j \in \mathcal{B}$, $B_i \subset \mathcal{X}$ and $B_i \cap B_j = \emptyset$. Furthermore, we have $|\mathcal{B}| = (\frac{M}{\tilde{l}})^d \approx \left( \frac{MK}{20\epsilon} \right)^d$ (+ constant).

The associated set $\mathcal{F}$ of functions can be constructed by utilising the set $\mathcal{Z}$ of ball centers $z_i$ for $B_i \in \mathcal{B}$ and the values $C_1^*, C_2^*$ computed above to define

$$\mathcal{F} := \{ g_z \in \mathcal{F}_0(C_1^*, C_2^*) | z \in \mathcal{Z} \} \cup \{ f^0 \}$$

where $f^0 \equiv 0$. Suppose that $n < \frac{1}{20^d} \left( \frac{MK}{\epsilon} \right)^d$ and consider an arbitrary $\hat{L} \in \mathcal{L}_{n,p}(\mathcal{X})$. By construction, there exists a ball $B$ in the set $\mathcal{B}$ with associated ball center $z \in \mathcal{Z}$ (as defined above) such that no observations are sampled in $B$. Therefore, if the unknown target function $f \in \{ g_z^0, f^0 \}$ then $\forall (x, \Omega(x)) \in G_{\mathcal{X}}^{\hat{L}}$, $\Omega(x) = 0$. This implies that we can freely set the target function to either $g_z^0$ or $f^0$ with no change to the Lipschitz constant estimate generated by $\hat{L}$. It then suffices to select $g_z^0$ if $\hat{L}$ generates a prediction that is smaller than $\epsilon$ and $f^0$ otherwise. As the choice of $\hat{L}$ was arbitrary, this implies that:

$$\forall \hat{L} \in \mathcal{L}_{n,p}(\mathcal{X}), \exists f \in \mathcal{F} \text{ such that } |\hat{L}(f) - L_p^*(f)| > \epsilon$$

and therefore:

$$\inf_{\hat{L} \in \mathcal{L}_{n,p}(\mathcal{X})} \sup_{f \in C^2(\mathcal{X}, K) \cap \mathcal{F}_p(L^*)} |\hat{L}(f) - L_p^*(f)| > \epsilon.$$

Utilising norm equivalences and Lemma B.2, we can apply similar arguments to the ones given above to obtain that in the case: $p = 1$, the sample complexity of the Lipschitz learning problem can be lower bounded by $\left( C(d) \frac{MK}{\epsilon} \right)^d$, where $C(d) = \frac{1}{20d^{\frac{1}{2}}}$.

∎

**Proof of Theorem 2.5 (Sample Complexity Bound − Noisy).**
Let $\epsilon > 0$ be sufficiently small such that $\frac{40\epsilon}{K} < M$ (which implies that the packing number $N(\mathcal{X}, \frac{20\epsilon}{K}) > 0$). Consider the maximal packing $\mathcal{B}_\epsilon$ of $\mathcal{X}$ of radius $\frac{20\epsilon}{K}$ with respect to $\|.\|_2$ and the associated class of functions $\mathcal{F}_0$ defined in Lemma A.1 which we denote $\mathcal{F}_\epsilon$ in this proof in order to explicitly mark the dependence on $\epsilon$ (we only consider $\mathcal{F}_L^*$ defined above for $L^* = 0$). We recall that for all $B \in \mathcal{B}_\epsilon$ and associated $f_B \in \mathcal{F}_\epsilon$; $\max_{x \in B} \|\nabla f_B(x)\|_q = 2\epsilon$ and $\max_{x \in \mathcal{X} \setminus B} \|\nabla f_B(x)\|_q = 0$. Therefore, by construction of $\mathcal{F}_\epsilon$, we have for any distinct pair of functions $f_1, f_2 \in \mathcal{F}_\epsilon$,

$$\forall x \in \mathcal{X}, \ \max \{ Loss(x, f_1), Loss(x, f_2) \} = \max \{ |\|\nabla f_1(x)\|_q - L^*|, |\|\nabla f_2(x)\|_q - L^*| \} \geq \epsilon.$$

with $L^* := 2\epsilon$. This implies that

$$\inf_{\hat{L} \in \mathcal{L}_{n,p}(\mathcal{X})} \sup_{f \in C^2(\mathcal{X}, K)} \mathbb{P}(Loss(x^{\hat{L}(f)}, f) > \epsilon) \geq \inf_{\hat{A} \in \mathcal{H}} \sup_{f \in \mathcal{F}_\epsilon} \mathbb{P}(f^{\hat{A}} \neq f).$$

---

[17]Which we define as the subset of $\mathcal{X}$ where $g$ is non-zero.
[18]This is often referred to as a "packing" of $\mathcal{X}$ by balls of radius $l$ and the maximum cardinality of such a set is called the "packing number" denoted $N(\mathcal{X}, l)$.

where $\mathcal{H}$ denotes the class of algorithms that utilise the data samples $G_{\mathcal{X}}^{\hat{L}}$ in order to select the correct $f$ in $\mathcal{F}_\epsilon$. In order to lower bound the right hand side of the above equation, Fano's Lemma can be applied. To do so, we first estimate $\log(|\mathcal{F}_\epsilon|)$ and $\sup_{f_1,f_2 \in \mathcal{F}_\epsilon} \mathrm{KL}\,(p_{f_1}\|p_{f_2})$ where $p_f$ denotes the density defined on $(\mathcal{X},\mathcal{Y})$ of a noisy sample $(x,\tilde{f}_x)$ associated to $f \in \mathcal{F}_\epsilon$ (defined more precisely below). The first term: $\log(|\mathcal{F}_\epsilon|)$ follows directly from the proof of Theorem 2.2 where we obtained that $|\mathcal{F}_\epsilon| = |\mathcal{B}_\epsilon| \geq \left(\frac{MK}{20\epsilon}\right)^d$ which implies

$$\log(|\mathcal{F}_\epsilon|) \geq d \log(\frac{MK}{20\epsilon}).$$

Let $f \in \mathcal{F}_\epsilon$. Denoting the density of the uniform measure on $\mathcal{X}$ as $\lambda_u$ and the density of the Gaussian measure on $\mathbb{R}$ with mean $\mu \in \mathbb{R}$ and variance $\sigma^2$ as $\nu_{\mu,\sigma^2}$, we have $\forall x \in \mathcal{X}$, $y \in \mathbb{R}$, $p_f(x,y) = \lambda_u(x)\nu_{f(x),\sigma^2}(y)$. Then, the second term can be upper bounded as follows:

$$\sup_{f_1,f_2 \in \mathcal{F}_\epsilon} \mathrm{KL}\,(p_{f_1}\|p_{f_2}) = \int_{\mathcal{X} \times \mathbb{R}} p_{f_1}(x,y) \log\left(\frac{p_{f_1}(x,y)}{p_{f_2}(x,y)}\right) d(x,y)$$

$$= \int_{\mathcal{X}} \int_{\mathbb{R}} \nu_{f_1(x),\sigma^2}(y) \log\left(\frac{\nu_{f_1(x),\sigma^2}(y)}{\nu_{f_2(x),\sigma^2}(y)}\right) dy \lambda_u(x) dx \stackrel{(i)}{=} \frac{1}{2\sigma^2} \int_{\mathcal{X}} |f_1(x) - f_2(x)|^2 \lambda_u(x) dx$$

$$\stackrel{(ii)}{\leq} \frac{1}{\sigma^2} \|f_1 - f_2\|_\infty^2 \int_{vol(B_\epsilon)} \lambda_u(x) dx$$

where $B_\epsilon$ denotes an arbitrary ball in the packing defined by $\mathcal{B}_\epsilon$. **(i)** follows from the well-known KL-divergence of univariate Gaussians and **(ii)** follows by construction of $\mathcal{F}_\epsilon$. By the proof of Theorem 3.7, we have for all $B \in \mathcal{B}_\epsilon$ and associated $f_B \in \mathcal{F}_\epsilon$, $\sup_{x \in B} |f_B(x)| = C_1^* = \frac{51\epsilon^2}{K}$. Furthermore, $\int_{vol(B_\epsilon)} \lambda_u(x) dx = \tilde{c}\frac{vol(B_\epsilon)}{M^d} = \tilde{c}' \left(\frac{20\epsilon}{MK}\right)^d$ for some constants $\tilde{c},\tilde{c}' := c(d) > 0$. Therefore, there exists a constant $c > 0$ such that

$$\sup_{f_1,f_2 \in \mathcal{F}_\epsilon} \mathrm{KL}\,(p_{f_1}\|p_{f_2}) \leq \frac{c}{\sigma^2} \frac{\epsilon^{d+4}}{M^d K^{d+2}}.$$

Applying Fano's Lemma, we obtain for an arbitrary ordering of $\mathcal{F}_\epsilon$:

$$\inf_{\hat{L} \in \mathcal{L}_{n,p}(\mathcal{X})} \sup_{f \in C^2(\mathcal{X},K)} \mathbb{P}(Loss(x^{\hat{L}(f)}, f) > \epsilon)$$

$$\geq 1 - \frac{\log(2) + n \sup_{f_1,f_2 \in \mathcal{F}_\epsilon} \mathrm{KL}\,(p_{f_1}\|p_{f_2})}{\log(|\mathcal{F}_\epsilon|)} \geq 1 - \frac{\log(2) + n \frac{c}{\sigma^2} \frac{\epsilon^{d+4}}{M^d K^{d+2}}}{d \log(\frac{MK}{20\epsilon})}.$$

Therefore, for $\epsilon$ sufficiently small and any arbitrary $\delta \in (0,1)$,

$$\inf_{\hat{L} \in \mathcal{L}_{n,p}(\mathcal{X})} \sup_{f \in C^2(\mathcal{X},K)} \mathbb{P}(Loss(x^{\hat{L}(f)}, f) > \epsilon) \leq \delta \implies 1 - \frac{\log(2) + n \frac{c}{2\sigma^2} \frac{\epsilon^{d+4}}{M^d K^{d+2}}}{d \log(\frac{MK}{20\epsilon})} \leq \delta.$$

Taking the limit as $\epsilon$ goes to 0, we have that if $n \notin \Omega\left(\frac{M^d K^{d+2} \log(\frac{MK}{\epsilon})}{\epsilon^{d+4}}\right)$, then $\lim_{\epsilon \to 0^+} 1 - \frac{\log(2) + n \frac{c}{2\sigma^2} \frac{\epsilon^{d+4}}{M^d K^{d+2}}}{d \log(\frac{MK}{20\epsilon})} = 1 > \delta$. This implies that

$$n \in \Omega\left(\frac{\sigma^2 M^d K^{d+2} \log(\frac{MK}{\epsilon})}{\epsilon^{d+4}}\right)$$

must necessarily hold in order for $\inf_{\hat{L} \in \mathcal{L}_{n,p}(\mathcal{X})} \sup_{f \in C^2(\mathcal{X},K)} \mathbb{P}(Loss(x^{\hat{L}(f)}, f) > \epsilon) \leq \delta$ to hold and concludes the proof.

■

# B Proofs: Theoretical Properties of LCLS

**Proof of Proposition 3.4 (Computational Complexity of LCLS).**
Follows directly from the computational complexity of the linear least squares regression algorithm which is $O(n_{samples})$.

∎

## B.1 Technical Lemmas

The proof of Theorem 3.7 relies on the following technical lemmas.

**Lemma B.1** *(Fundamental logarithm inequalities) For all $x > 0$,*

$$1 - \frac{1}{x} \leq \log(x) \leq x - 1.$$

**Lemma B.2** *Let $\mathcal{X}$ be as described in Assumption 1. If $f$ verifies Assumption 2, then $f$ is $L_p^*$-Lipschitz with respect to $\|.\|_p$. Furthermore, for $p = 1, 2$, $L_p^* = \max_{x \in \mathcal{X}}\{\|\nabla f(x)\|_q\}$ and for $p > 2$, $L_p^* \leq \max_{x \in \mathcal{X}}\{\|\nabla f(x)\|_q\}$ where $q$ is the Hölder conjugate of $p$.*

**Proof** $f$ is $L_p^*$-Lipschitz follows directly from the fact that $\mathcal{X}$ is compact and $f \in C^1(\mathcal{X})$.

$\forall p \in \mathbb{N}$, $L_p^* \leq \max_{x \in \mathcal{X}}\{\|\nabla f(x)\|_q\}$ follows from the multidimensional mean-value theorem and an application of the Hölder inequality.

For $p = 1, 2$, we show $L_p^* \geq \max_{x \in \mathcal{X}}\{\|\nabla f(x)\|_q\}$. Consider : $\forall x \in \mathcal{X}$ consider the Frechet derivative of $f$ at $x$; $\lim_{t \to 0} \frac{|f(x+th) - f(x) - \nabla f(x)^\top (th)|}{t} = 0 \ \forall h \in \mathbb{R}^d$. Then, choosing $h = \nabla f(x)$ for $p = 2$ and $h = e_{i^*}$ such that $|\nabla f(x)^\top e_{i^*}| = \|\nabla f(x)\|_\infty$ for $p = 1$ gives $L_p^* \geq \max_{x \in \mathcal{X}}\{\|\nabla f(x)\|_q\}$. (Note that this reasoning is well defined because $f \in C^1(\tilde{\mathcal{X}})$ and $\tilde{\mathcal{X}}$ is an open set that contains $\mathcal{X}$ which implies that $f(x+th)$ is well-defined for any $h \in \mathbb{R}^d$ and small enough $t$).

∎

**Lemma B.3** *Let $\delta \in (0, 1)$, then $\forall x > 2\log(\frac{1}{1 - \frac{\delta}{2}})$:*

$$\frac{1}{1 - \sqrt[x]{1 - \frac{\delta}{2}}} \leq \frac{2x}{\log(\frac{1}{1 - \frac{\delta}{2}})}.$$

**Proof** Let $x > 2\log(\frac{1}{1 - \frac{\delta}{2}})$ be arbitrary, we have

$$\frac{1}{1 - \sqrt[x]{1 - \frac{\delta}{2}}} \leq \frac{2x}{\log(\frac{1}{1 - \frac{\delta}{2}})} \iff 1 \leq \frac{2x}{\log(\frac{1}{1 - \frac{\delta}{2}})}(1 - \sqrt[x]{1 - \frac{\delta}{2}}).$$

Then,

$$\frac{2x}{\log(\frac{1}{1 - \frac{\delta}{2}})}(1 - \sqrt[x]{1 - \frac{\delta}{2}}) = \frac{2x}{\log(\frac{1}{1 - \frac{\delta}{2}})}(1 - e^{-\frac{1}{x}\log(\frac{1}{1 - \frac{\delta}{2}})})$$

Utilising the fact that $e^y \leq 1 + y + y^2$ for all $y < 1$, we have that the above expression is greater or equal to:

$$\frac{2x}{\log(\frac{1}{1 - \frac{\delta}{2}})}\left(\frac{1}{x}\log(\frac{1}{1 - \frac{\delta}{2}}) - \frac{1}{x^2}\log(\frac{1}{1 - \frac{\delta}{2}})^2\right) = 2\left(1 - \frac{1}{x}\log(\frac{1}{1 - \frac{\delta}{2}})\right) \geq 1$$

where last inequality follows from the fact that $x > 2\log(\frac{1}{1 - \frac{\delta}{2}})$.

■

**Lemma B.4** *Consider a sequence of partitions $(\mathcal{H}_I)_{I \in \mathbb{N}}$ used by LCLS and assume that for a given $\eta \in (0, 1]$ the sampling distribution satisfies Assumption 4. Then,*

$$\forall I \in \mathbb{N}, \, \forall H \in \mathcal{H}_I, \, \|\left(X_I^{H\top} X_I^H\right)^{-1}\|_2 \leq \frac{16}{\eta \delta_I^{H^2} N_I^H}.$$

**Proof** Let $\lambda_{max}(M)$ denote the maximum eigenvalue of a matrix $M$ if it exists. $\forall I \in \mathbb{N}, \, \forall H \in \mathcal{H}_I$, we have $\|\left(X_I^{H\top} X_I^H\right)^{-1}\|_2 = \frac{1}{\sigma_{min}(X_I^{H\top} X_I^H)}$ where $\sigma_{min}(X_I^{H\top} X_I^H)$ denotes the smallest singular value of $X_I^{H\top} X_I^H$. Therefore, we can focus on showing the following relation that implies the Lemma statement:

$$\sigma_{min}(X_I^{H\top} X_I^H) \geq \frac{\eta N_I^H}{16} \delta_I^{H^2}.$$

Let $\bar{X}_I^H$ be the design matrix without the first column of ones, ie. $\bar{X}_I^H = \begin{bmatrix} x_{H_1}^\top \\ x_{H_2}^\top \\ \vdots \\ x_{H_{N_I^H}}^\top \end{bmatrix}$, we have

$$\sigma_{min}(X_I^{H\top} X_I^H) = \sigma_{min}(X_I^{H\top})^2 = \min_{\|u\|_2=1} \|X_I^{H\top} u\|_2^2 = \min_{\|u\|_2=1} \|\begin{bmatrix} \mathbf{1}_{N_I^H}^\top \\ \bar{X}_I^{H\top} \end{bmatrix} u\|_2^2$$

$$\geq \min_{\|u\|_2=1} \|\begin{bmatrix} \mathbf{0}_{N_I^H}^\top \\ \bar{X}_I^{H\top} \end{bmatrix} u\|_2^2 = \min_{\|u\|_2=1} \|\bar{X}_I^{H\top} u\|_2^2 = \sigma_{min}(\bar{X}_I^{H\top} \bar{X}_I^H).$$

Therefore we can consider the smallest singular value of $\bar{X}_I^H$ instead of $X_I^H$ which allows for a direct use of Assumption 4. We have

$$\sigma_{min}(\bar{X}_I^{H\top} \bar{X}_I^H) = \lambda_{min}(\bar{X}_I^{H\top} \bar{X}_I^H) = \min_{u \in \mathbb{R}^d, \|u\|_2=1} u^\top \bar{X}_I^{H\top} \bar{X}_I^H u = \min_{u \in \mathbb{R}^d, \|u\|_2=1} \sum_{i=1}^{N_I^H} \langle x_{H_i}, u \rangle^2.$$

Let $u_{min}^* \in \mathbb{R}^d$ denote the eigenvector associated to $\lambda_{min}(\bar{X}_I^{H\top} \bar{X}_I^H)$ with $\|u_{min}^*\|_2 = 1$. Then, $u_{min}^*$ satisfies $\sum_{i=1}^{N_I^H} \langle x_{H_i}, u_{min}^* \rangle^2 = \min_{\|u\|_2=1} \sum_{i=1}^{N_I^H} \langle x_{H_i}, u \rangle^2$.

Using $u_{min}^*$, we can construct an orthonormal basis of $\{u_{min}^*, u_2, ..., u_d\}$ of $\mathbb{R}^d$ (such a basis can be constructed using the Gramm-Schmitt algorithm). Then, since $D_I^H$ contains $\eta N_I^H$ points in each ball associated to an element of an $\frac{\delta_I^H}{8}$-cover of $H$, we have that there exists at least $\eta N_I^H$ pairs of datapoints $(x_{H_i}, \tilde{f}_{H_j})$, $(x_{H_j}, \tilde{f}_{H_j}) \in D_I^H$ such that $\exists \{\alpha_i\}_{i \in \{1,...,d\}}, \alpha_i \in \mathbb{R}$ with $|\alpha_1| > \frac{\delta_I^H}{2}$ and $(x_{H_i} - x_{H_j}) = \alpha_1 u_{min}^* + \sum_{k=2}^{d} \alpha_k u_k$. This implies that $\max(|\langle x_{H_i}, u_{min}^* \rangle|, |\langle x_{H_j}, u_{min}^* \rangle|) \geq \frac{\delta_I^H}{4}$. Indeed, if $|\langle x_{H_i}, u_{min}^* \rangle| < \frac{\delta_I^H}{4}$, then

$$|\langle x_{H_j}, u_{min}^* \rangle| = |\langle x_{H_j} - x_{H_i} + x_{H_i}, u_{min}^* \rangle| = |\langle x_{H_j} - x_{H_i}, u_{min}^* \rangle + \langle x_{H_i}, u_{min}^* \rangle| \geq \frac{\delta_I^H}{2} - \frac{\delta_I^H}{4} = \frac{\delta_I^H}{4}$$

Using this inequality $\eta N_I^H$ times we can conclude, $\sigma_{min}(\bar{X}_I^{H\top} \bar{X}_I^H) = \sum_{i=1}^{N_I^H} \langle x_{H_i}, u_{min}^* \rangle^2 \geq \eta N_I^H (\frac{\delta_I^H}{4})^2$.

■

**Lemma B.5** *Consider the constructions of Definition 3.3. The following relationship holds for all $I \in \mathbb{N}$,*

$$\frac{V(\mathcal{X})\Gamma(\frac{d}{2} + 1)2^d}{\pi^{\frac{d}{2}} \max_{H \in \mathcal{H}_I} (\Delta_I^H)^d} \leq |\mathcal{H}_I| \leq \frac{V(\mathcal{X})\Gamma(\frac{d}{2} + 1)2^d}{\pi^{\frac{d}{2}} \min_{H \in \mathcal{H}_I} (\delta_n)^d}$$

*where $V(\mathcal{X})$ denotes the volume of $\mathcal{X}$.*

**Proof** Follows directly from the definition of $\{\delta_I^H\}_{H\in\mathcal{H}_I}$, $\{\Delta_I^H\}_{H\in\mathcal{H}_I}$ and volume formula for the d-dimensional ball.

∎

**Lemma B.6** *Let the notation and assumptions be as described in Theorem 3.7 and define $x^* \in \mathcal{X}$ as $x^* \in argmax_{x\in\mathcal{X}}\|\nabla f(x)\|_q$. Then, $\forall I \in \mathbb{N}$,*

$$\left| \|\nabla f(x^*)\|_q - \max_{H\in\mathcal{H}_I}\{\|\mathbb{E}[\hat{\beta}_I^H]\|_q\}\right| \leq \frac{4\sqrt{d}dn_qK}{\sqrt{\eta}}a_I.$$

*where $n_q = d^{\max\{\frac{1}{q}-\frac{1}{2},0\}}$.*

**Proof** Note: such an $x^*$ exists by compactness of $\mathcal{X}$ and the fact that $f \in C^2(\mathcal{X})$.

By definition, $[\hat{b}_I^H, \hat{\beta}_I^H]^\top = (X_I^{H\top}X_I^H)^{-1}X_I^{H\top}\tilde{f}_I^H$. Computing the expectation of this expression yields

$$\mathbb{E}\left[[\hat{b}_I^H, \hat{\beta}_I^H]^\top\right] = \mathbb{E}\left[(X_I^{H\top}X_I^H)^{-1}X_I^{H\top}\tilde{f}_I^H\right]$$

$$= \mathbb{E}\left[(X_I^{H\top}X_I^H)^{-1}X_I^{H\top}f_I^H\right] + \mathbb{E}\left[(X_I^{H\top}X_I^H)^{-1}X_I^{H\top}\gamma_I^H\right] = (X_I^{H\top}X_I^H)^{-1}X_I^{H\top}f_I^H.$$

$\forall H \in \mathcal{H}_I$, let $c_H \in \overline{H}$ (closure of $H$) be such that $\|\nabla f(c_H)\|_q = \max_{x\in\bar{H}}\{\|\nabla f(x)\|_q\}$ which exists by compactness of $\bar{H}$ and the fact that $f \in C^2(\mathcal{X})$. Then, using the second order Taylor expansion of $f$ around $c_h$, every coordinate $f_{H_k}$ of $f_I^H$ can be re-expressed as

$$f_{H_k} = f(c_H) + (x_{H_k} - c_H)^\top\nabla f(c_H) + (x_{H_k} - c_H)^\top Hess(c_H + r_{H_k}(x_{H_k} - c_H))(x_{H_k} - c_H), \quad (3)$$

where $r_{H_k} \in [0,1]$ and $Hess$ denotes the Hessian matrix of $f$. To alleviate notation, let $\|.\|_{\tilde{q}}$ denote a pseudo-norm on $\mathbb{R}^{d+1}$ defined by; $x \in \mathbb{R}^{d+1}$, $\|x\|_{\tilde{q}} := \sqrt[q]{\sum_{i=2}^{d+1}x_i^q}$ if $q < \infty$ and $\|x\|_{\tilde{\infty}} := \max_{i\in\{2,...,d+1\}}|x_i|$ otherwise. Then, using the definition of $X_I^H$,

$$\|\mathbb{E}[\hat{\beta}_I^H]\|_q = \left\|\begin{bmatrix}\mathbb{E}[\hat{b}_I^H]\\\mathbb{E}[\hat{\beta}_I^H]\end{bmatrix}\right\|_{\tilde{q}} = \|(X_I^{H\top}X_I^H)^{-1}X_I^{H\top}f_I^H\|_{\tilde{q}}$$

$$= \left\|(X_I^{H\top}X_I^H)^{-1}X_I^{H\top}(X_I^H\begin{bmatrix}f(c_H) - c_H^\top\nabla f(c_H)\\0\\\vdots\\0\end{bmatrix} + X_I^H\begin{bmatrix}0\\\nabla f(c_H)\end{bmatrix} + \begin{bmatrix}(x_{H_1}-c_H)^\top Hess(r_{H_1})(x_{H_1}-c_H)\\(x_{H_2}-c_H)^\top Hess(r_{H_2})(x_{H_2}-c_H)\\\vdots\end{bmatrix})\right\|_{\tilde{q}}$$

$$= \left\|\begin{bmatrix}f(c_H) - c_H^\top\nabla f(c_H)\\0\\\vdots\\0\end{bmatrix} + \begin{bmatrix}0\\\nabla f(c_H)\end{bmatrix} + \underbrace{(X_I^{H\top}X_I^H)^{-1}X_I^{H\top}(\begin{bmatrix}(x_{H_1}-c_H)^\top Hess(r_{H_1})(x_{H_1}-c_H)\\(x_{H_2}-c_H)^\top Hess(r_{H_2})(x_{H_2}-c_H)\\\vdots\end{bmatrix})}_{=:J(H)}\right\|_{\tilde{q}}$$

Plugging this expression into the theorem statement yields:

$$\left|\|\nabla f(x^*)\|_q - \max_{H\in\mathcal{H}_I}\{\|\mathbb{E}[\hat{\beta}_I^H]\|_q\}\right| = \|\nabla f(x^*)\|_q - \max_{H\in\mathcal{H}_I}\{\|\mathbb{E}[\hat{\beta}_I^H]\|_q\}$$

$$\leq \|\nabla f(x^*)\|_q - (\max_{H\in\mathcal{H}_I}\{\|\begin{bmatrix}0\\\nabla f(c_H)\end{bmatrix}\|_{\tilde{q}}\} - \max_{H\in\mathcal{H}_I}\{\|J(H)\|_{\tilde{q}}\}) \leq \|\nabla f(x^*)\|_q - \max_{H\in\mathcal{H}_I}\{\|\nabla f(c_H)\|_q\}$$

$$+ \max_{H\in\mathcal{H}_I}\{\|J(H)\|_q\} = \max_{H\in\mathcal{H}_I}\{\|J(H)\|_q\}$$

where the last equality follows from the fact that there exists $H \in \mathcal{H}_I$ such that $x^* \in H$. As $f \in C^2(\mathcal{X},K)$, we have $\forall i,j \in \{1,...,d\}$, $\forall x \in \mathcal{X}$ that $|\frac{\partial^2 f}{\partial x_i\partial x_j}(x)| < K$. This implies that $\|Hess(x)\|_1 \leq dK \ \forall x \in \mathcal{X}$ and by

matrix norm equivalence; $\|Hess(x)\|_2 \leq \sqrt{d}\|Hess(x)\|_1 \leq d\sqrt{d}K$, $\forall x \in \mathbb{R}^d$. Therefore, since matrix p-norms are sub-multiplicative;

$$\|J(H)\|_q \leq n_q\|J(H)\|_2 \leq n_q\|(X_I^{H^\top}X_I^H)^{-1}X_I^{H^\top}\|_2\|\begin{bmatrix} (x_{H_1}-c_1)^\top Hess(r_{H_1})(x_{H_1}-c_H) \\ (x_{H_2}-c_2)^\top Hess(r_{H_2})(x_{H_2}-c_H) \\ \vdots \end{bmatrix}\|_2$$

where $n_q = d^{\max\{\frac{1}{q}-\frac{1}{2},0\}}$. Using Lemma B.4 we have

$$\|(X_I^{H^\top}X_I^H)^{-1}X_I^{H^\top}\|_2 = \|X_I^H((X_I^{H^\top}X_I^H)^\top)^{-1}\|_2 = \sqrt{\lambda_{max}((X_I^{H^\top}X_I^H)^{-1})}$$

$$= \sqrt{\|(X_I^{H^\top}X_I^H)^{-1}\|_2} \leq \frac{4}{\delta_I^H\sqrt{\eta N_I^H}}$$

where $\lambda_{max}$ denotes the maximum eigenvalue of $(X_I^{H^\top}X_I^H)^{-1}$. Furthermore, the components of the vector on the right-hand side can be upper bounded by using Cauchy-Schwartz and matrix-vector inequalities;

$$|(x_{H_1}-c_1)^\top Hess(r_{H_1})(x_{H_1}-c_H)| \leq \|(x_{H_1}-c_1)^\top\|_2\|Hess(r_{H_1})(x_{H_1}-c_H)\|_2$$

$$\leq \|(x_{H_1}-c_1)^\top\|_2\|Hess(r_{H_1})\|_2\|(x_{H_1}-c_H)\|_2 \leq \Delta_I^{H^2}\sqrt{d}dK.$$

Combining these two upper bounds we can conclude:

$$\left|\|\nabla f(x^*)\|_q - \max_{H \in \mathcal{H}_I}\{\|\mathbb{E}[\hat{\beta}_I^H]\|_q\}\right| \leq \max_{H \in \mathcal{H}_I}\frac{4\sqrt{d}dKn_q}{\sqrt{\eta}}\frac{\Delta_I^{H^2}}{\delta_I^H} = \frac{4\sqrt{d}dn_qK}{\sqrt{\eta}}a_I.$$

$\blacksquare$

**Lemma B.7** *If the Assumptions of Theorem 3.7 hold, then $\forall I \in \mathbb{N}$, the difference between the Lipschitz estimate generated by the LCLS method with noisy sampling $\hat{L}_I$ and the Lipschitz estimate generated by the LCLS method with noiseless sampling $\bar{L}_I$ can be upper bounded by;*

$$\mathbb{P}(|\bar{L}_I - \hat{L}_I| > \frac{\epsilon}{2}) \leq 1 - \prod_{H \in \mathcal{H}_I}(1 - \frac{2^6\sigma^2 d^{\max\{\frac{2}{q},1\}}}{\eta\epsilon^2}\frac{1}{N_I^H\delta_I^{H^2}}). \tag{4}$$

**Proof** Let $I \in \mathbb{N}$. $\forall H \in \mathcal{H}_I$ denote by $[b_I^H, \beta_I^H]$ the least squares coefficients computed using $(X_I^H, f_I^H)$ (instead of $(X_I^H, \tilde{f}_I^H)$), i.e. the noiseless least squares coefficients. Then,

$$\mathbb{E}\left[[\hat{b}_I^H, \hat{\beta}_I^H]^\top\right] = \mathbb{E}\left[(X_I^{H^\top}X_I^H)^{-1}X_I^{H^\top}f_I^H\right] = [b_I^H, \beta_I^H]^\top$$

Therefore, we can write (with $n_q = d^{\max\{\frac{1}{q}-\frac{1}{2},0\}}$)

$$\mathbb{P}\left(\left|\bar{L}_I - \hat{L}_I\right| > \frac{\epsilon}{2}\right) = \mathbb{P}\left(|\max_{H \in \mathcal{H}_I}\{\|[\beta_I^H]\|_q\} - \max_{H \in \mathcal{H}_I}\{\|\hat{\beta}_I^H\|_q\}| > \frac{\epsilon}{2}\right)$$

$$= \mathbb{P}\left(\left|\max_{H \in \mathcal{H}_I}\{\|\mathbb{E}\left[\hat{\beta}_I^H\right]\|_q\} - \max_{H \in \mathcal{H}_I}\{\|\hat{\beta}_I^H\|_q\}\right| > \frac{\epsilon}{2}\right) \leq \mathbb{P}\left(\left|\max_{H \in \mathcal{H}_I}\{\|\mathbb{E}\left[\hat{\beta}_I^H\right]\|_q - \|\hat{\beta}_I^H\|_q\}\right| > \frac{\epsilon}{2}\right)$$

$$\leq \mathbb{P}\left(\max_{H \in \mathcal{H}_I}\{\|\mathbb{E}\left[\hat{\beta}_I^H\right] - \hat{\beta}_I^H\|_q\} > \frac{\epsilon}{2}\right) \leq \mathbb{P}\left(\max_{H \in \mathcal{H}_I}\{\|\mathbb{E}\left[\hat{\beta}_I^H\right] - \hat{\beta}_I^H\|_2\} > \frac{\epsilon}{2n_q}\right)$$

$$= 1 - \mathbb{P}\left(\max_{H \in \mathcal{H}_I}\{\|\mathbb{E}\left[\hat{\beta}_I^H\right] - \hat{\beta}_I^H\|_2\} < \frac{\epsilon}{2n_q}\right) \leq 1 - \prod_{H \in \mathcal{H}_I}\mathbb{P}\left(\|\mathbb{E}\left[\hat{\beta}_I^H\right] - \hat{\beta}_I^H\|_2 < \frac{\epsilon}{2n_q}\right)$$

$$= 1 - \prod_{H \in \mathcal{H}_I}\left(1 - \mathbb{P}\left(\|\mathbb{E}\left[\hat{\beta}_I^H\right] - \hat{\beta}_I^H\|_2 \geq \frac{\epsilon}{2n_q}\right)\right)$$

In order to upper bound the term given in product: $\mathbb{P}(\|\mathbb{E}[\hat{\beta}_I^H] - \hat{\beta}_I^H\|_2 \geq \frac{\epsilon}{2n_q})$, we use the covariance matrix: $var([\hat{b}_I^H, \hat{\beta}_I^H]) = \sigma^2(X_I^{H\top}X_I^H)^{-1}$ which follows from the fact that the components of $\gamma_I^H$ are assumed to be uncorrelated with mean 0 and variance $\sigma^2$. We also denote by $Tr(M)$ the trace of a matrix $M \in \mathbb{R}^{d \times d}$. Then, by applying an extension of Chebyshev's inequality to finite dimensional vectors (Ferentios (1982)) and Lemma B.4, we have

$$\mathbb{P}\left(\|\mathbb{E}\left[\hat{\beta}_I^H\right] - \hat{\beta}_I^H\|_2 \geq \frac{\epsilon}{2n_q}\right) \overset{\underset{Inequality}{Chebychev's}}{\leq} \frac{4n_q^2\sigma^2 Tr((X_I^{H\top}X_I^H)^{-1})}{\epsilon^2}$$

$$\leq \frac{4n_q^2\sigma^2 d\|\left(X_I^{H\top}X_I^H\right)^{-1}\|_2}{\epsilon^2} \overset{Lemma\ B.4}{\leq} \frac{4n_q^2\sigma^2 d}{\epsilon^2}\frac{16}{\eta\delta_I^{H^2}N_I^H} = \frac{2^6 n_q^2\sigma^2 d}{\eta\epsilon^2}\frac{1}{N_I^H \delta_I^{H^2}}.$$

Plugging this expression into the product given above concludes the proof.

$$\mathbb{P}(|\bar{L}_I - \hat{L}_I| > \frac{\epsilon}{2}) \leq 1 - \prod_{H \in \mathcal{H}_I}\left(1 - \frac{2^6\sigma^2 d^{\max\{\frac{2}{\hat{q}}-1,0\}}d}{\eta\epsilon^2}\frac{1}{N_I^H \delta_I^{H^2}}\right)$$

∎

## B.2 Proof of Main Theoretical Properties of LCLS

**Proof of Theorem 3.7 (General Convergence Rate).**
We recall that $\forall I \in \mathbb{N}$, the Lipschitz estimate $\hat{L}_I$ is obtained by considering the partition $\mathcal{H}_I$ and computing $\max_{H \in \mathcal{H}_I}\{\|\mathbb{E}[\hat{\beta}_I^H]\|_q\}$. Let $\epsilon > 0$ be arbitrary. We need to show for $p = 1, 2$:

$$\lim_{I \to \infty} \mathbb{P}(|L_p^* - \hat{L}_I| > \epsilon) = 0$$

and for $p > 2$

$$\lim_{I \to \infty} \mathbb{P}(|L_p - \hat{L}_I| > \epsilon) = 0 \text{ with } L_p \in \mathbb{R}_{\geq L_p^*}.$$

Since $f$ verifies assumption 2 and $\mathcal{X}$ is convex and compact, Lemma B.2 guarantees the existence of $x^* \in \mathcal{X}$ such that $\|\nabla f(x^*)\|_q = L_p^*$ for $p = 1, 2$ and $L_p := \|\nabla f(x^*)\|_q = \max_{x \in \mathcal{X}}\|\nabla f(x)\|_p \geq L_p^*$ for $p > 2$.

Therefore, for all $p \geq 1$, we can consider the statement;

$$\lim_{I \to \infty} \mathbb{P}(|\|\nabla f(x^*)\|_q - \hat{L}_I| > \epsilon) = 0$$

Let $I \in \mathbb{N}$ and consider $\mathbb{P}(|\|\nabla f(x^*)\|_q - \hat{L}_I| > \epsilon)$. This expression can be split into two terms:

$$\mathbb{P}(|\|\nabla f(x^*)\|_q - \hat{L}_I| > \epsilon)$$
$$\leq \underbrace{\mathbb{P}\left(|\|\nabla f(x^*)\|_q - \max_{H \in \mathcal{H}_I}\{\|\mathbb{E}\left[\hat{\beta}_I^H\right]\|_q\}| > \frac{\epsilon}{2}\right)}_{(I)} + \underbrace{\mathbb{P}\left(|\max_{H \in \mathcal{H}_I}\{\|\mathbb{E}\left[\hat{\beta}_I^H\right]\|_q\} - \hat{L}_I| > \frac{\epsilon}{2}\right)}_{(II)}.$$

In the following, we show that both (I) and (II) converge to 0 when $I$ goes to infinity.

**(I):** From Lemma B.6, $\left|\|\nabla f(x^*)\|_q - \max_{H \in \mathcal{H}_I}\{\|\mathbb{E}\left[\hat{\beta}_I^H\right]\|_q\}\right| \leq \frac{4\sqrt{d}dn_qK}{\sqrt{\eta}}a_I$. Plugging this upper bound into the above expression, we have

$$\mathbb{P}\left(|\|\nabla f(x^*)\|_q - \max_{H \in \mathcal{H}_I}\{\|\mathbb{E}\left[\hat{\beta}_I^H\right]\|_q\}| > \frac{\epsilon}{2}\right) \leq \mathbb{P}(\frac{4\sqrt{d}dn_qK}{\sqrt{\eta}}a_I > \frac{\epsilon}{2}).$$

By hypothesis 2. $\lim_{I\to\infty} a_I = 0$ and therefore there exists $I_1 \in \mathbb{N}$ sufficiently large such that $\frac{4\sqrt{d}dn_qK}{\sqrt{\eta}}a_{I_1} \leq \frac{\epsilon}{2}$ and therefore $\mathbb{P}(\frac{4\sqrt{d}dn_qK}{\sqrt{\eta}}a_{I_1} > \frac{\epsilon}{2}) = 0$.

**(II):** We show that $\mathbb{P}(|\max_{H\in\mathcal{H}_I}\{\|\mathbb{E}[\hat{\beta}_I^H]\|_q\} - \hat{L}_I| > \frac{\epsilon}{2})$ converges to 0 as $I$ goes to infinity. Let $\bar{L}$ denote the Lipschitz estimate generated by LCLS with noiseless samples. Then, applying Lemma B.7, we have the following upper bound on $\mathbb{P}(|\max_{H\in\mathcal{H}_I}\{\|\mathbb{E}[\hat{\beta}_I^H]\|_q\} - \hat{L}_I| > \frac{\epsilon}{2})$:

$$\mathbb{P}(|\max_{H\in\mathcal{H}_I}\{\|\mathbb{E}[\hat{\beta}_I^H]\|_q\} - \hat{L}_I| > \frac{\epsilon}{2}) = \mathbb{P}(|\bar{L}_I - \hat{L}_I| > \frac{\epsilon}{2}) \leq 1 - \prod_{H\in\mathcal{H}_I}(1 - \frac{16\sigma^2 d^{\max\{\frac{2}{q},1\}}}{\eta\epsilon^2}\frac{1}{N_I^H\delta_I^{H2}})$$

$$\leq 1 - (1 - \frac{2^6\sigma^2 d^{\max\{\frac{2}{q},1\}}}{\eta\epsilon^2\min_{H\in\mathcal{H}_I}(N_I^H\delta_I^{H2})})^{|\mathcal{H}_I|}.$$

As by Theorem hypothesis 2 $\lim_{I\to\infty}\max_{H\in\mathcal{H}_I}(\Delta_I^H) = 0$, applying Lemma B.5 implies that $\lim_{I\to\infty}|\mathcal{H}_I| = \infty$. Therefore using the fact that $\lim_{I\to\infty} b_I = 0$, we have $\lim_{I\to\infty}\max_{H\in\mathcal{H}_I}\left(\frac{1}{N_I^H\delta_I^{H2}}\right) = \lim_{I\to\infty}\frac{1}{\min_{H\in\mathcal{H}_I}N_I^H\delta_I^{H2}} = 0$.

To alleviate notation, let $(\alpha_I)_{I\in\mathbb{N}}$ be the sequence defined by $\alpha_I := \frac{2^6\sigma^2 d^{\max\{\frac{2}{q},1\}}}{\eta\epsilon^2\min_{H\in\mathcal{H}_I}N_I^H\delta_I^{H2}}$, then $\lim_{I\to\infty}\frac{1}{\min_{H\in\mathcal{H}_I}N_I^H\delta_I^{H2}} = 0$ implies that $\exists\bar{I}\in\mathbb{N}$ such that $\forall I \geq \bar{I}$, $\alpha_I < 0.5$. Utilising fundamental logarithm inequalities, we obtain:

$$1 - (1-\alpha_I)^{|\mathcal{H}_I|} \leq |\mathcal{H}_I|\log(\frac{1}{1-\alpha_I}) \leq |\mathcal{H}_I|\frac{\alpha_I}{1-\alpha_I} \leq |\mathcal{H}_I|\frac{\alpha_I}{2}$$

$$= \frac{2^5\sigma^2 d^{\max\{\frac{2}{q},1\}}}{\eta\epsilon^2}\frac{|\mathcal{H}_I|}{\min_{H\in\mathcal{H}_I}N_I^H\delta_I^{H2}} = (\frac{2^5\sigma^2 d^{\max\{\frac{2}{q},1\}}}{\eta\epsilon^2})b_I \xrightarrow{I\to\infty} 0.$$

∎

**Proof of Corollary 3.9 (Noiseless Oracle).**
As in the proof of Theorem 3.7, we can consider the statement; $p \in \mathbb{N}$,

$$\lim_{I\to\infty}\mathbb{P}(|\|\nabla f(x^*)\|_q - \hat{L}_I| > \epsilon) = 0$$

where $x^* \in \text{argmax}_{x\in\mathcal{X}}\|\nabla f(x)\|_p$. Since the data samples contain no noise, $\hat{L}_I = \max_{H\in\mathcal{H}_I}\{\|\mathbb{E}[\hat{\beta}_I^H]\|_q\}$ and

$$\mathbb{P}(|\|\nabla f(x^*)\|_q - \hat{L}_I| > \epsilon) = \mathbb{P}(|\|\nabla f(x^*)\|_q - \max_{H\in\mathcal{H}_I}\{\|\mathbb{E}[\hat{\beta}_I^H]\|_q\}| > \epsilon).$$

Then, applying Lemma B.6 and using $\lim_{I\to\infty} a_I = 0$ as in the proof of Theorem 3.7 gives the desired convergence result.
(Note: that the least squares estimation is well defined as $N_I^H \geq d + 1$ and Assumption 4 holds.)

∎

**Proof of Theorem 3.10 (Finite Sample Guarantee).**
We show the equivalent statement; $\mathbb{P}(|L_p - \hat{L}_I| > \epsilon) \leq \delta$. where as in proof of Theorem 3.7, $L_p := \|\nabla f(x^*)\|_q$ with $x^* := \text{argmax}_{x\in\mathcal{X}}\|\nabla f(x)\|_p$ and $L_p = L_p^*$ for $p = 1, 2$, $L_p \geq L_p^*$ for $p > 2$.
In the hypercube set-up, we have $\forall I \in \mathbb{N}_{>1}$, $\forall H \in \mathcal{H}_I$, $\Delta_I^H = \frac{\sqrt{d}M}{I}$, $\delta_I^H = \frac{M}{I}$ and $|\mathcal{H}_I| = I^d$. Let $\epsilon > 0$, $\delta \in (0, \frac{1}{2}]$: From the proof of Theorem 3.7 we have that three following inequalities need to be satisfied in order for (1) to hold. From **(I)** we need $\frac{4\sqrt{d}dn_qK}{\sqrt{\eta}}\frac{\Delta_I^{H2}}{\delta_I^H} \leq \frac{\epsilon}{2}$ in order for $\mathbb{P}(|\|\nabla f(x^*)\|_q - \max_{H\in\mathcal{H}_I}\{\|\mathbb{E}[\hat{\beta}_I^H]\|_q\}| > \frac{\epsilon}{2}) = 0$. This implies that;

$$I \geq \frac{8d^2\sqrt{d}n_q}{\sqrt{\eta}}\frac{MK}{\epsilon}.$$

From **(II)**, we have the following two inequalities that need to be satisfied;

$$(1) \qquad \alpha_I = \frac{2^6 \sigma^2 d^{\max\{\frac{2}{q},1\}}}{\eta \epsilon^2 \min_{H \in \mathcal{H}_I} N_I^H \delta_I^{H^2}} < 0.5$$

$$(2) \qquad \frac{2^5 \sigma^2 d^{\max\{\frac{2}{q},1\}}}{\eta \epsilon^2} \frac{|\mathcal{H}_I|}{\min_{H \in \mathcal{H}_I} N_I^H \delta_I^{H^2}} < \delta$$

The first implies that

$$\frac{2^7 d^{\max\{\frac{2}{q},1\}} \sigma^2}{\eta} \frac{I^2}{M^2 \epsilon^2} < \min_{H \in \mathcal{H}_I} N_I^H$$

and the second expression gives

$$\frac{2^5 d^{\max\{\frac{2}{q},1\}} \sigma^2}{\eta} \frac{I^2}{M^2 \epsilon^2} \frac{|\mathcal{H}_I|}{\delta} < \min_{H \in \mathcal{H}_I} N_I^H.$$

Since $|\mathcal{H}_I| = I^d$, $I \in \mathbb{N}_{>1}$ and $\delta \in (0, \frac{1}{2}]$, we have that if the $\min_{H \in \mathcal{H}_I} N_I^H$ satisfies (2) then (1) is true as well. Therefore, we have $\forall H \in \mathcal{H}_I$;

$$\frac{2^5 d^{\max\{\frac{2}{q},1\}} \sigma^2}{\eta} \frac{I^{d+2}}{\delta M^2 \epsilon^2} < \min_{H \in \mathcal{H}_I} N_I^H.$$

Setting $C_1(d) = 8d^2 \sqrt{d} d^{\max\{\frac{1}{q} - \frac{1}{2}, 0\}}$ and $C_2(d, q) = 2^5 d^{\max\{\frac{2}{q},1\}} d$ concludes the proof.

$\blacksquare$

**Proof of Theorem 3.15 (Asymptotic Sample Complexity – Gaussian Noise).**
Consider the setting described by Theorem 3.15 with $I = \lceil C_1(d) \frac{MK}{\epsilon} \rceil$ when $\epsilon > 0$. As described in the proof of Theorem 3.10: in the the hypercube set-up we have $\forall H \in \mathcal{H}_I$, $\Delta_I^H = \frac{\sqrt{d}M}{I}$, $\delta_I^H = \frac{M}{I}$ and $|\mathcal{H}_I| = I^d$.

For all $\epsilon > 0$, let $\mathcal{A}_\epsilon$ denote the event that every $H \in \mathcal{H}_I$ contains a number of samples $N_I^H$ equal or greater than $C(d) \frac{\log(\frac{MK}{\epsilon}) \sigma^2 K^2}{\epsilon^4}$ for a constant $C(d) > 0$ that depends on $d$ (see $(\star)$ for the explicit definition of $C(d)$) and is $(\frac{\delta_I^H}{8}, \eta)$-covered where $\eta = \frac{vol(B_1(0))}{2^{3d+1}3^d}$ where $vol(B_1(0))$ denotes the volume of the d-dimensional unit ball. More precisely,

$$\mathcal{A}_\epsilon := \{\forall H \in \mathcal{H}_I : N_I^H \geq C(d) \frac{\log(\frac{MK}{\epsilon}) \sigma^2 K^2}{\epsilon^4} \land H \text{ is } (\frac{\delta_I^H}{8}, \eta)\text{-covered}\}.$$

Let us assume that there exists $\bar{\epsilon} > 0$ such that $\forall \epsilon \in (0, \bar{\epsilon})$, $\mathbb{P}(\mathcal{A}_\epsilon) > 0$ (this will follow from $(\star\star)$ given at the end of the proof). Then,

$$\sup_{f \in C^2(\mathcal{X}, K)} \mathbb{P}(Loss(x^{\hat{L}_I(f)}, f) > \epsilon) \leq \sup_{f \in C^2(\mathcal{X}, K)} \mathbb{P}(Loss(x^{\hat{L}_I(f)}, f) > \epsilon | \mathcal{A}_\epsilon) + \mathbb{P}(\mathcal{A}_\epsilon^c).$$

Therefore, in order to show Theorem 3.15, it suffices to show that both terms of the right-hand expression given above converge to 0 as $\epsilon$ goes to 0. The first part of the proof considers $\sup_{f \in C^2(\mathcal{X}, K)} \mathbb{P}(Loss(x^{\hat{L}_I(f)}, f) > \epsilon | \mathcal{A}_\epsilon)$. We will show that for all $\delta > 0$, there exists $\bar{\epsilon}^*$ such that $\forall \epsilon \in (0, \bar{\epsilon}^*)$,

$$\sup_{f \in C^2(\mathcal{X}, K)} \mathbb{P}(Loss(x^{\hat{L}_I(f)}, f) > \epsilon | \mathcal{A}_\epsilon) < \delta$$

**Notation B.8** *To alleviate notation, we omit the conditional dependence on $\mathcal{A}_\epsilon$ in the following computations.*

Fix an arbitrary $\delta > 0$ and define $\forall H \in \mathcal{H}_I$, $\bar{\beta}_I^H := [b_I^H, \beta_I^H]^\top = (X_I^{H\top} X_I^H)^{-1} X_I^{H\top} f_I^H$. As the noise and the sampling distribution are independent and every input sample is selected independently, we have

$$\bar{\beta}_I^H = \mathbb{E}\left[[\hat{b}_I^H, \hat{\beta}_I^H]^\top \Big| X_I^H\right] = \mathbb{E}\left[[\hat{b}_I^H, \hat{\beta}_I^H]^\top \Big| G_I^{\mathcal{X}}\right]$$

where $G_I^{\mathcal{X}}$ denotes the set of all sample inputs utilised by the LCLS algorithm.. (Note that $\bar{\beta}_I^H$ is a random variable as the sample inputs are randomly sampled). We have

$$\mathbb{P}(Loss(x^{\hat{L}_I(f)}, f) > \epsilon) = \mathbb{P}\left(\left|L_p^* - \|\nabla f(x^{\hat{L}_I(f)})\|_q\right| > \epsilon\right)$$

$$\leq \mathbb{P}\left(\left|L_p^* - \hat{L}_I(f)\right| \geq \frac{\epsilon}{2}\right) + \mathbb{P}\left(\left|\hat{L}_I(f) - \|\nabla f(x^{\hat{L}_I(f)})\|_q\right| \geq \frac{\epsilon}{2}\right)$$

$$\leq \mathbb{P}\left(\left|\|\nabla f(x^*)\|_q - \max_{H \in \mathcal{H}_I}\{\|\mathbb{E}\left[\hat{\beta}_I^H \Big| G_I^{\mathcal{X}}\right]\|_q\}\right| > \frac{\epsilon}{4}\right) + \mathbb{P}\left(\left|\max_{H \in \mathcal{H}_I}\{\|\mathbb{E}\left[\hat{\beta}_I^H \Big| G_I^{\mathcal{X}}\right]\|_q\} - \hat{L}_I(f)\right| > \frac{\epsilon}{4}\right)$$

$$+\mathbb{P}\left(\left|\|\nabla f(x^{\hat{L}_I(f)})\|_q - \|\mathbb{E}\left[\hat{\beta}_I^{H^{\hat{L}_I(f)}} \Big| G_I^{\mathcal{X}}\right]\|_q\right| > \frac{\epsilon}{4}\right) + \mathbb{P}\left(\left|\|\mathbb{E}\left[\hat{\beta}_I^{H^{\hat{L}_I(f)}} \Big| G_I^{\mathcal{X}}\right]\|_q - \hat{L}_I(f)\right| > \frac{\epsilon}{4}\right).$$

where $x^* := \text{argmax}_{x \in \mathcal{X}} \|\nabla f(x)\|_q = L_p^*$ by Lemma B.2 (for $p = 1, 2$) and $\hat{\beta}_I^{H^{\hat{L}_I(f)}}$ denotes parameters of the linear regression associated to the hypercube $\text{argmax}_{H \in \mathcal{H}_I} \|\hat{\beta}^H\|_q$. By the arguments given at the beginning of Lemma B.7, we have

$$\left|\max_{H \in \mathcal{H}_I}\{\|\mathbb{E}\left[\hat{\beta}_I^H \Big| G_I^{\mathcal{X}}\right]\|_q\} - \hat{L}_I(f)\right| \leq n_q \max_{H \in \mathcal{H}_I}\{\|\mathbb{E}\left[\hat{\beta}_I^H \Big| G_I^{\mathcal{X}}\right] - \hat{\beta}_I^H\|_2\}$$

where we recall $n_q = d^{\max\{\frac{1}{q} - \frac{1}{2}, 0\}}$. Similarly, by construction of LCLS, the reverse triangle inequality and norm equivalence,

$$\left|\|\mathbb{E}\left[\hat{\beta}_I^{H^{\hat{L}_I(f)}} \Big| G_I^{\mathcal{X}}\right]\|_q - \hat{L}_I\right| = \left|\|\mathbb{E}\left[\hat{\beta}_I^{H^{\hat{L}_I(f)}} \Big| G_I^{\mathcal{X}}\right]\|_q - \|\hat{\beta}_I^{H^{\hat{L}_I(f)}}\|_q\right|$$

$$\leq n_q\|\mathbb{E}\left[\hat{\beta}_I^{H^{\hat{L}_I(f)}} \Big| G_I^{\mathcal{X}}\right] - \hat{\beta}_I^{H^{\hat{L}_I(f)}}\|_2 \leq n_q \max_{H \in \mathcal{H}_I}\{\|\mathbb{E}\left[\hat{\beta}_I^H \Big| G_I^{\mathcal{X}}\right] - \hat{\beta}_I^H\|_2\}.$$

Therefore,

$$\mathbb{P}(Loss(x^{\hat{L}_I(f)}, f) > \epsilon) \leq \underbrace{\mathbb{P}\left(\left|\|\nabla f(x^*)\|_q - \max_{H \in \mathcal{H}_I}\{\|\mathbb{E}\left[\hat{\beta}_I^H \Big| G_I^{\mathcal{X}}\right]\|_q\}\right| > \frac{\epsilon}{4}\right)}_{(i)}$$

$$+\underbrace{\mathbb{P}\left(\left|\|\nabla f(x^{\hat{L}_I(f)})\|_q - \|\mathbb{E}\left[\hat{\beta}_I^{H^{\hat{L}_I(f)}} \Big| G_I^{\mathcal{X}}\right]\|_q\right| > \frac{\epsilon}{4}\right)}_{(ii)} + \underbrace{2\mathbb{P}\left(\max_{H \in \mathcal{H}_I}\{\|\mathbb{E}\left[\hat{\beta}_I^H \Big| G_I^{\mathcal{X}}\right] - \hat{\beta}_I^H\|_2\} > \frac{\epsilon}{4n_q}\right)}_{(iii)}.$$

The terms $(i), (ii)$ in the above expression can be shown to be equal to 0 with similar arguments. For $(i)$, Lemma B.6 can be utilised (as $\mathcal{A}_\epsilon$ is assumed to hold) to obtain:

$$\left|\|\nabla f(x^*)\|_q - \max_{H \in \mathcal{H}_I}\{\|\mathbb{E}\left[\hat{\beta}_I^H \Big| G_I^{\mathcal{X}}\right]\|_q\}\right| \leq \max_{H \in \mathcal{H}_I} \frac{4\sqrt{d}dKn_q}{\sqrt{\eta}} \frac{\Delta_I^{H^2}}{\delta_I^H} \leq \frac{4\sqrt{d}dn_qK}{\sqrt{\eta}} a_I$$

and it follows from applying the same approach as the one used in the proof of Lemma B.6 (as $\mathcal{A}_\epsilon$ is assumed to hold), that

$$\left|\|\nabla f(x^{\hat{L}_I(f)})\|_q - \|\mathbb{E}\left[\hat{\beta}_I^{H^{\hat{L}_I(f)}} \Big| G_I^{\mathcal{X}}\right]\|_q\right| \leq \frac{4\sqrt{d}dKn_q}{\sqrt{\eta}} \frac{\Delta_I^{H^{\hat{L}_I(f)^2}}}{\delta_I^{H^{\hat{L}_I(f)}}} \leq \frac{4\sqrt{d}dn_qK}{\sqrt{\eta}} a_I.$$

Note that $\eta$ is defined in the omitted conditioning on $A_\epsilon$. Then, we have by definition of $I = \lceil C_1(d) \frac{MK}{\epsilon} \rceil$,

$$\frac{4\sqrt{d}dn_qK}{\sqrt{\eta}} a_I = \frac{4\sqrt{d}d^2n_qK}{\sqrt{\eta}} \frac{M}{I} \leq \frac{4\sqrt{d}d^2n_q}{\sqrt{\eta}} \frac{\epsilon}{C_1(d)} = \frac{\epsilon}{4}.$$

where the last line follows from the fact $C_1(d) = \frac{16d^2\sqrt{d}n_q}{\sqrt{\eta}}$. Therefore, conditional on $\mathcal{A}_\epsilon$, we have

$$\mathbb{P}\left(\left|\|\nabla f(x^*)\|_q - \max_{H \in \mathcal{H}_I}\{\|\mathbb{E}\left[\hat{\beta}_I^H\Big|G_I^\mathcal{X}\right]\|_q\}\right| > \frac{\epsilon}{4}\right) = \mathbb{P}\left(\left|\|\nabla f(x^{\hat{L}_I(f)})\|_q - \|\mathbb{E}\left[\hat{\beta}_I^{H^{\hat{L}_I(f)}}\Big|G_I^\mathcal{X}\right]\|_q\right| > \frac{\epsilon}{4}\right) = 0.$$

In order to show that **(iii)** converges to 0 as $\epsilon$ goes to 0, we define for all $H \in \mathcal{H}_I$: $E^H := \{\|\mathbb{E}[\hat{\beta}_I^H\big|G_I^\mathcal{X}] - \hat{\beta}_I^H\|_2 \leq \frac{\epsilon}{4n_q}\}$, consider an arbitrary ordering of $\mathcal{H}_I := \{H_1, ..., H_{|\mathcal{H}_I|}\}$ and apply similar arguments as the ones utilised in the proof of Lemma B.7 to obtain

$$\mathbb{P}\left(\max_{H \in \mathcal{H}_I}\{\|\mathbb{E}\left[\hat{\beta}_I^H\Big|G_I^\mathcal{X}\right] - \hat{\beta}_I^H\|_2\} > \frac{\epsilon}{4n_q}\Big|\mathcal{A}_\epsilon\right) = 1 - \mathbb{P}(\forall H \in \mathcal{H}_I, \ E^H|\mathcal{A}_\epsilon)$$

$$= 1 - \mathbb{P}\left(E^{H_1}\big|\mathcal{A}_\epsilon\right)\prod_{i=2}^{|\mathcal{H}_I|}\mathbb{P}\left(E^{H_i}\big|\mathcal{A}_\epsilon, E^{H_1}, ..., E^{H_{i-1}}\right)$$

The computation of the local linear regressions parameters is done independently with no data overlap implying that the conditioning expression: $\{E^{H_1}, ..., E^{H_{i-1}}\}$ (for $i = 1, ..., |\mathcal{H}_I|$) can only impact the probability by affecting the number of samples contained in $H_i$: $N_I^{H_i}$ which are utilised in the local linear regression. As the probabilities are also each conditioned on $\mathcal{A}_\epsilon$ which provides a fixed lower bound on $N_I^H$ for all $H \in \mathcal{H}_I$ and the remaining arguments for this part of the proof will only utilise this fact, we use a slight abuse of notation in order to alleviate notation and omit the dependencies on $\{E^{H_1}, ..., E^{H_{i-1}}\}$ (for $i = 1, ..., |\mathcal{H}_I|$) in the remainder of this part of the proof. Therefore, we can consider

$$1 - \prod_{H \in \mathcal{H}_I}\left(1 - \mathbb{P}\left(\|\bar{\beta}_I^H - [\hat{b}_I^H, \hat{\beta}_I^H]^\top\|_2 \geq \frac{\epsilon}{4n_q}\Big|\mathcal{A}_\epsilon\right)\right).$$

In order to upper bound $\mathbb{P}(\|\bar{\beta}_I^H - [\hat{b}_I^H, \hat{\beta}_I^H]^\top\|_2 \geq \frac{\epsilon}{4n_q}|\mathcal{A}_\epsilon)$ a more refined bound than the general Chebyshev inequality used in the proof of Lemma B.7 is utilised. Instead, we apply Corollary 3 of (Pinelis and Sakhanenko (1986)) to obtain an alternative bound (see **(iv)** below).

Remarking that $0 < \epsilon < \frac{C_1(d)MK}{3}$ implies $I = \lceil C_1(d) \frac{MK}{\epsilon} \rceil \leq \sqrt{2}C_1(d) \frac{MK}{\epsilon}$, we set $\bar{\epsilon}_1 := \frac{C_1(d)MK}{3}$. Then, for all $\epsilon \in (0, \bar{\epsilon}_1)$,

$$\mathbb{P}\left(\|\bar{\beta}_I^H - [\hat{b}_I^H, \hat{\beta}_I^H]^\top\|_2 \geq \frac{\epsilon}{4n_q}\Big|\mathcal{A}_\epsilon\right) = \mathbb{P}\left(\|(X_I^{H^\top}X_I^H)^{-1}X_I^{H^\top}\gamma_L^H\|_2 \geq \frac{\epsilon}{4n_q}\Big|\mathcal{A}_\epsilon\right)$$

$$\overset{\textbf{(iv)}}{\leq} 2e^{-\frac{(\frac{\epsilon}{4n_q})^2}{2\mathbb{E}[\|(X_I^{H^\top}X_I^H)^{-1}X_I^{H^\top}\gamma_L^H\|_2^2|\mathcal{A}_\epsilon]}}.$$

As the Gaussian vector $(X_I^{H^\top}X_I^H)^{-1}X_I^{H^\top}\gamma_L^H$ has covariance matrix $\sigma^2(X_I^{H^\top}X_I^H)^{-1}$, we can utilise the tower rule to observe that

$$\mathbb{E}\left[\|(X_I^{H^\top}X_I^H)^{-1}X_I^{H^\top}\gamma_L^H\|_2^2\Big|\mathcal{A}_\epsilon\right] = \mathbb{E}\left[\mathbb{E}[\|(X_I^{H^\top}X_I^H)^{-1}X_I^{H^\top}\gamma_L^H\|_2^2|G_I^\mathcal{X}]\Big|\mathcal{A}_\epsilon\right]$$

$$= \mathbb{E}\left[\sigma^2 Tr((X_I^{H^\top}X_I^H)^{-1})\Big|\mathcal{A}_\epsilon\right] \leq \mathbb{E}\left[d\sigma^2\|(X_I^{H^\top}X_I^H)^{-1}\|_2\Big|\mathcal{A}_\epsilon\right] \leq \frac{16d\sigma^2}{\eta\delta_I^{H^2}\bar{N}_I}.$$

where $\bar{N}_I := C(d)\frac{\log(\frac{MK}{\epsilon})\sigma^2 K^2}{\epsilon^4}$ is given by $\mathcal{A}_\epsilon$ (with $C(d)$ explicitly determined below). The last inequality follows from Lemma B.4 which can be applied by definition of $\mathcal{A}_\epsilon$. This implies:

$$\mathbb{P}\left(\|\bar{\beta}_I^H - [\hat{b}_I^H, \hat{\beta}_I^H]^\top\|_2 \geq \frac{\epsilon}{4n_q}|\mathcal{A}_\epsilon\right) \leq 2e^{-\frac{\epsilon^2 \eta \delta H_I^2 \bar{N}_I}{2^9 n_q^2 d\sigma^2}} = 2e^{-\frac{\epsilon^2 \eta M^2 \bar{N}_I}{2^9 n_q^2 I^2 d\sigma^2}} \leq 2e^{-\frac{\epsilon^4 \eta \bar{N}_I}{2^{10} n_q^2 K^2 C_1(d)^2 d\sigma^2}}.$$

Therefore, denoting $C_2(d) := \frac{\eta}{2^{10} n_q^2 C_1(d)^2 d}$ and plugging the above expression into the initial upper bound, we obtain

$$\mathbb{P}\left(Loss(x^{\hat{L}_I(f)}, f) > \epsilon|\mathcal{A}_\epsilon\right) \leq 2 - 2\prod_{H\in\mathcal{H}_I}(1 - 2e^{-C_2(d)\frac{\epsilon^4 \bar{N}_I}{\sigma^2 K^2}}) = 2 - 2(1 - 2e^{-C_2(d)\frac{\epsilon^4 \bar{N}_I}{\sigma^2 K^2}})^{|\mathcal{H}_I|}.$$

Then, setting $2 - 2(1 - 2e^{-C_2(d)\frac{\epsilon^4 \bar{N}_I}{\sigma^2 K^2}})^{|\mathcal{H}_I|} \leq \delta$, we obtain that if

$$\bar{N}_I \geq \frac{\sigma^2 K^2}{C_2(d)\epsilon^4}\log(\frac{2}{1 - {}^{|\mathcal{H}_I|}\sqrt{1 - \frac{\delta}{2}}})$$

then $\mathbb{P}(Loss(x^{\hat{L}_I(f)}, f) > \epsilon|\mathcal{A}_\epsilon) \leq \delta$. As $|\mathcal{H}_I|$ is monotonically increasing and converges to infinity as $\epsilon$ goes to 0, there exists $\bar{\epsilon}_2 > 0$ such that $\forall \epsilon \in (0, \bar{\epsilon}_2)$, $|\mathcal{H}_I| \geq 2\log(\frac{1}{1-\frac{\delta}{2}})$. This implies that we can apply Lemma B.3 to obtain:

$$\frac{2}{1 - {}^{|\mathcal{H}_I|}\sqrt{1 - \frac{\delta}{2}}} \leq \frac{4|\mathcal{H}_I|}{\log(\frac{1}{1-\frac{\delta}{2}})}.$$

Therefore, we have that the following stronger condition on $\bar{N}_I$ implies $\mathbb{P}(Loss(x^{\hat{L}_I(f)}, f) > \epsilon) \leq \delta$:

$$\bar{N}_I \geq \frac{\sigma^2 K^2}{C_2(d)\epsilon^4}\log(\frac{2}{1 - {}^{|\mathcal{H}_I|}\sqrt{1 - \frac{\delta}{2}}}) \Longleftarrow \bar{N}_I \geq \frac{\sigma^2 K^2}{C_2(d)\epsilon^4}\log(\frac{4|\mathcal{H}_I|}{\log(\frac{1}{1-\frac{\delta}{2}})}).$$

We can rewrite this lower bound with $|\mathcal{H}_I|$ expressed in terms of $\epsilon$:

$$\frac{\sigma^2 K^2}{C_2(d)\epsilon^4}\log(\frac{4|\mathcal{H}_I|}{\log(\frac{1}{1-\frac{\delta}{2}})}) = \frac{\sigma^2 K^2}{C_2(d)\epsilon^4}\log(2^{d+2}C_1(d)^d(\frac{MK}{\epsilon})^d\log(\frac{1}{1-\frac{\delta}{2}})^{-1})$$

$$= \frac{\sigma^2 K^2}{C_2(d)\epsilon^4}d\log(C_3(d)\log(\frac{1}{1-\frac{\delta}{2}})^{-\frac{1}{d}}\frac{MK}{\epsilon})$$

where $C_3(d) := 2^{1+\frac{2}{d}}C_1(d)$. Finally, there exists $\bar{\epsilon}_3 > 0$ such that $\forall \epsilon \in (0, \bar{\epsilon}_3)$,

$$\frac{C_3(d)}{\log(\frac{1}{1-\frac{\delta}{2}})^{\frac{1}{d}}} \leq \frac{MK}{\epsilon}$$

which implies

$$\bar{N}_I \geq C^*(d)\frac{\sigma^2 K^2}{\epsilon^4}\log(\frac{MK}{\epsilon}) \implies \mathbb{P}(Loss(x^{\hat{L}_I(f)}, f) > \epsilon) \leq \delta$$

where $C^*(d) := \frac{2d}{C_2(d)}$. Therefore, as $C^*(d)$ only depends on $d$ (note that $\eta$ depends only on $d$) we can set $C(d) = C^*(d)$ $(\star)$.

Selecting $\bar{\epsilon}^* := \min(\epsilon_1, \epsilon_2, \epsilon_3)$, we have $\forall \epsilon \in (0, \bar{\epsilon}^*)$

$$\sup_{f\in C^2(\mathcal{X}, K)}\mathbb{P}(Loss(x^{\hat{L}_I(f)}, f) > \epsilon|\mathcal{A}_\epsilon) < \delta.$$

As the choice of $\delta > 0$ was arbitrary, this concludes the first part of the proof.

($\star\star$) We now show $\lim_{\epsilon \to 0^+} \mathbb{P}(\mathcal{A}_\epsilon^c) = 0$ with $C^*(d)$ as defined above in ($\star$). Let $\epsilon \in (0, C_1(d)MK)$ be arbitrary and define the following events:

$$\mathcal{A}_\epsilon^1 := \left\{ \forall H \in \mathcal{H}_I : N_I^H \geq C^*(d) \frac{\log(\frac{MK}{\epsilon})\sigma^2 K^2}{\epsilon^4} \right\}$$

$$\mathcal{A}_\epsilon^2 := \left\{ \forall H \in \mathcal{H}_I : H \text{ is } (\frac{\delta_I^H}{8}, \eta)\text{-covered} \right\}.$$

We recall:

$$\mathcal{A}_\epsilon := \left\{ \forall H \in \mathcal{H}_I : N_I^H \geq C^*(d) \frac{\log(\frac{MK}{\epsilon})\sigma^2 K^2}{\epsilon^4} \wedge H \text{ is } (\frac{\delta_I^H}{8}, \eta)\text{-covered} \right\} = \left\{ \mathcal{A}_\epsilon^1 \wedge \mathcal{A}_\epsilon^2 \right\}.$$

We can write:

$$\mathbb{P}(\mathcal{A}_\epsilon^c) = 1 - \mathbb{P}(\mathcal{A}_\epsilon) = 1 - \mathbb{P}(\mathcal{A}_\epsilon^2 | \mathcal{A}_\epsilon^1)\mathbb{P}(\mathcal{A}_\epsilon^1)$$

which is well defined as we will show that $\mathbb{P}(\mathcal{A}_\epsilon^1) > 0$ for sufficiently small $\epsilon$. Therefore, if we can show that

$$\lim_{\epsilon \to 0^+} \mathbb{P}(\mathcal{A}_\epsilon^2 | \mathcal{A}_\epsilon^1) = \lim_{\epsilon \to 0^+} \mathbb{P}(\mathcal{A}_\epsilon^1) = 1$$

then the proof of ($\star\star$) is concluded.

**(I)** We begin by showing $\lim_{\epsilon \to 0^+} \mathbb{P}(\mathcal{A}_\epsilon^1) = 1$. For all $H \in \mathcal{H}_I$, we define

$$\mathcal{E}_\epsilon^H(n) := \left\{ H \text{ contains } \geq C^*(d) \frac{\log(\frac{MK}{\epsilon})\sigma^2 K^2}{\epsilon^4} \text{ for total sample points equal to } n \right\}$$

where $n$ denotes the total number of samples which was assumed to satisfy: $n \geq C \frac{\sigma^2 M^d K^{d+2} \log(\frac{MK}{\epsilon})}{\epsilon^{d+4}}$ for a fixed constant $C > 0$ (defined explicitly below). Then, considering an arbitrary ordering of $\mathcal{H}_I := \{H_1, ..., H_{|\mathcal{H}_I|}\}$, we have

$$\mathbb{P}(\mathcal{A}_\epsilon^1) = \mathbb{P}\left(\forall H \in \mathcal{H}_I, \mathcal{E}_\epsilon^H(n)\right) = \left(\mathcal{E}_\epsilon^{H_1}(n)\right) \prod_{i=2}^{|\mathcal{H}_I|} \mathbb{P}\left(\mathcal{E}_\epsilon^{H_i}(n) | \mathcal{E}_\epsilon^{H_1}(n), ..., \mathcal{E}_\epsilon^{H_{i-1}}(n)\right)$$

As the inputs are sampled independently and the elements of $\mathcal{H}_I$ are disjointed by construction, we have $\forall i \in \{2, ..., |\mathcal{H}_I|\}$

$$\mathbb{P}\left(\mathcal{E}_\epsilon^{H_i}(n) | \mathcal{E}_\epsilon^{H_1}(n), ..., \mathcal{E}_\epsilon^{H_{i-1}}(n)\right) = \mathbb{P}\left(\mathcal{E}_\epsilon^{H_i}(n - (i-1)C^*(d)\frac{\log(\frac{MK}{\epsilon})\sigma^2 K^2}{\epsilon^4})\right).$$

It trivial to see that $\forall i \in \{1, ..., |H_i|\}$, $\mathbb{P}(\mathcal{E}_\epsilon^{H_i}(n))$ is increasing in $n$. Thus, we have

$$\mathbb{P}(\mathcal{A}_\epsilon^1) \geq \prod_{i=1}^{|\mathcal{H}_I|} \mathbb{P}\left(\mathcal{E}_\epsilon^{H_i}(n - |\mathcal{H}_I|C^*(d)\frac{\log(\frac{MK}{\epsilon})\sigma^2 K^2}{\epsilon^4})\right).$$

For $\epsilon > 0$ satisfying $\epsilon < C_1(d)MK$, we have $I \leq 2C_1(d)\frac{MK}{\epsilon}$. Therefore, using $|\mathcal{H}_I| = I^d \leq \frac{(2C_1(d)MK)^d}{\epsilon^d}$, we have

$$|\mathcal{H}_I| \frac{C^*(d)\log(\frac{MK}{\epsilon})\sigma^2 K^2}{\epsilon^4} \leq \frac{(2C_1(d)MK)^d}{\epsilon^d} \frac{C^*(d)\log(\frac{MK}{\epsilon})\sigma^2 K^2}{\epsilon^4}$$

$$= (2C_1(d))^d C^*(d) \frac{M^d K^{d+2}\log(\frac{MK}{\epsilon})}{\epsilon^{d+4}} \leq \frac{(2C_1(d))^d C^*(d)}{C} n$$

where the last inequality follows from the theorem assumption: $n \geq C \frac{\sigma^2 M^d K^{d+2} \log(\frac{MK}{\epsilon})}{\epsilon^{d+4}}$. Therefore, defining $\bar{C}_1 := 2(2C_1(d))^d C^*(d)$, setting $C \geq \bar{C}_1$, and utilising the upper bound derived above, we obtain

$$\prod_{i=1}^{|\mathcal{H}_I|} \mathbb{P}\left(\mathcal{E}_\epsilon^{H_i}(n - |\mathcal{H}_I|C^*(d)\frac{\log(\frac{MK}{\epsilon})\sigma^2 K^2}{\epsilon^4})\right) \geq \prod_{i=1}^{|\mathcal{H}_I|} \mathbb{P}\left(\mathcal{E}_\epsilon^{H_i}(n - \frac{n}{2})\right) = \prod_{i=1}^{|\mathcal{H}_I|} \mathbb{P}\left(\mathcal{E}_\epsilon^{H_i}(\frac{n}{2})\right).$$

We now consider for all $H \in \mathcal{H}_I$ the computation of $\mathbb{P}(\mathcal{E}_\epsilon^H(\frac{n}{2}))$ for which we will derive a lower bound.

For all $H \in \mathcal{H}_I$, denote $M_I^H(n) := |\{i \in \{0, ..., n\} : x_i \in H\}|$ the random variable[19] that counts the number of sample inputs in $H$. As every sample input is sampled uniformly on $\mathcal{X}$ and for all $H \in \mathcal{H}_I$ $vol(H) = (\frac{M}{I})^d \geq \frac{\epsilon^d}{(2C_1(d)K)^d}$, we have that the probability of a sample input being in $H \in \mathcal{H}_I$ can be modelled using a Bernouilli random variable with success probability $p = \frac{vol(H)}{vol(\mathcal{X})} \geq \frac{\epsilon^d}{(2C_1(d)MK)^d}$. Therefore, $M_I^H(n)$ can be modelled as a sum of independent Bernoulli variables with success probability $p$. From Lemma 1 of (Stone (1982)), we have

$$\mathbb{P}(M_I^H(n) \leq \frac{\mathbb{E}[M_I^H(n)]}{2}) \leq (\frac{2}{e})^{\frac{\mathbb{E}[M_I^H(n)]}{2}}.$$

In order to apply this result, we observe that as $C \geq \bar{C}_1 = 2(2C_1(d))^d C^*(d)$ (by construction), the following relations hold:

$$\mathbb{E}[M_I^H(\frac{n}{2})] = \frac{n}{2}p \geq \frac{\sigma^2 K^2 \log(\frac{MK}{\epsilon})}{\epsilon^4} \frac{C}{C_1(d)^d 2^{d+1}} \geq C^*(d)\frac{\log(\frac{MK}{\epsilon})\sigma^2 K^2}{\epsilon^4}$$

where the rightmost term corresponds to the bound stated in the definition of the $\mathcal{E}_\epsilon^H(\frac{n}{2})$ events. This implies that $\mathbb{P}(\mathcal{E}_\epsilon^H(\frac{n}{2}))$ can be lower bounded as follows:

$$\mathbb{P}\left(\mathcal{E}_\epsilon^H(\frac{n}{2})\right) = \mathbb{P}\left(M_I^H(\frac{n}{2}) \geq C^*(d)\frac{\log(\frac{MK}{\epsilon})\sigma^2 K^2}{\epsilon^4}\right)$$

$$= 1 - \mathbb{P}\left(M_I^H(\frac{n}{2}) \leq C^*(d)\frac{\log(\frac{MK}{\epsilon})\sigma^2 K^2}{\epsilon^4}\right) \geq 1 - \mathbb{P}\left(M_I^H(\frac{n}{2}) \leq \frac{\mathbb{E}[M_I^H(\frac{n}{2})]}{2}\right) \geq 1 - (\frac{2}{e})^{\frac{\mathbb{E}[M_I^H(\frac{n}{2})]}{2}}$$

Plugging this expression into the initial bound, we obtain:

$$\prod_{i=1}^{|\mathcal{H}_I|} \mathbb{P}\left(\mathcal{E}_\epsilon^{H_i}(\frac{n}{2})\right) \geq \prod_{i=1}^{|\mathcal{H}_I|} \left(1 - (\frac{2}{e})^{\frac{\mathbb{E}[M_I^H(\frac{n}{2})]}{2}}\right).$$

As $\mathbb{E}[M^H(\frac{n}{2})] \geq \frac{\sigma^2 K^2 \log(\frac{MK}{\epsilon})}{\epsilon^4} \frac{C}{C_1(d)^d 2^{d+1}}$ and $|\mathcal{H}_I| = I^d$ (i.e. both terms increase polynomially with respect to $\frac{1}{\epsilon}$), the above expression can be shown to go to 1 as $\epsilon$ goes to 0. This implies that if $C \geq \bar{C}_1$, then

$$\lim_{\epsilon \to 0^+} \mathbb{P}(\mathcal{A}_\epsilon^1) = 1.$$

**(II)** We now show that $\lim_{\epsilon \to 0^+} \mathbb{P}(\mathcal{A}_\epsilon^2 | \mathcal{A}_\epsilon^1) = 1$.

By the law of total probability, we can derive:

$$\mathbb{P}(\mathcal{A}_\epsilon^2 | \mathcal{A}_\epsilon^1) = \sum_{\{\bar{N}_I^H\}_{H \in \mathcal{H}_I} \in V_I(n)} \mathbb{P}\left(\mathcal{A}_\epsilon^2 | \{\bar{N}_I^H\}_{H \in \mathcal{H}_I}\right) \mathbb{P}\left(\{N_I^H\}_{H \in \mathcal{H}_I} = \{\bar{N}_I^H\}_{H \in \mathcal{H}_I} | \mathcal{A}_\epsilon^1\right)$$

$$= \sum_{\{\bar{N}_I^H\}_{H \in \mathcal{H}_I} \in V_I(n)} \left(\prod_{H \in \mathcal{H}_I} \mathbb{P}(H \text{ is } (\frac{\delta_I^H}{8}, \eta)\text{-covered} | \bar{N}_I^H)\right) \mathbb{P}\left(\{N_I^H\}_{H \in \mathcal{H}_I} = \{\bar{N}_I^H\}_{H \in \mathcal{H}_I} | \mathcal{A}_\epsilon^1\right)$$

---

[19]In essence, $M_I^H(n) = N_I^H$ but makes explicit the dependency on $n$.

where $V_I(n) := \{\{\bar{N}_I^H\}_{H \in \mathcal{H}_I} \in \mathbb{N}_{\geq c(\epsilon)}^{|\mathcal{H}_I|} : \sum_{H \in |\mathcal{H}_I|} N_I^H = n\}$ with $c(\epsilon) := C^*(d)\frac{\log(\frac{MK}{\epsilon})\sigma^2 K^2}{\epsilon^4}$ defined as the bound stated in $\mathcal{A}_\epsilon^1$. The second equality follows from the definition of $(\frac{\delta_I^H}{8}, \eta)$-covered and the disjointness of the partition. Note that $V_I(n)$ is non-empty, i.e. $n \geq c(\epsilon)|\mathcal{H}_I|$, due to the inequality: $C \geq \bar{C}_1$ set in **(I)**.

For an arbitrary $\{\bar{N}_I^H\}_{H \in \mathcal{H}_I} \in V_I(n)$, we now focus on lower bounding $\prod_{H \in \mathcal{H}_I} \mathbb{P}(H \text{ is } (\frac{\delta_I^H}{8}, \eta)\text{-covered } |\bar{N}_I^H)$.

In order to do so, for each $H \in \mathcal{H}_I$ we define $\mathcal{C}_I^H$: the minimal cover of $H$ with balls of radius $\frac{\delta_I^H}{8}$ with respect to $\|.\|_2$ and the associated set of hyperballs: $\mathcal{B}_I^H$.

Let $I \in \mathbb{N}$, $H \in \mathcal{H}_I$ be arbitrary, without loss of generality, we can assume that $\mathcal{C}_I^H \subset H$ as $H$ is a hypercube. This implies that for all $B \in \mathcal{B}_I^H$, $vol(B \cap H) \geq 2^{-d}vol(B) = 2^{-d}vol(B_1(0))(\frac{\delta_I^H}{8})^d$ where $vol(B_1(0))$ corresponds to the volume of the unit ball and is a constant[20] that depends on $d$. Utilising $\{\mathcal{B}_I^H\}_{H \in \mathcal{H}_I}$, we construct the set $\tilde{\mathcal{B}}_I^H$ as follows

$$\tilde{\mathcal{B}}_I^H := \left\{ \tilde{B}_I^H \subset H : \exists B_I^H \in \mathcal{B}_I^H \text{ such that } \tilde{B}_I^H = H \cap B_I^H \right\}.$$

We have $\bigcup_{B \in \tilde{\mathcal{B}}_I^H} B = H$, $|\tilde{\mathcal{B}}_I^H| = |\mathcal{C}_I^H|$ and for all $B \in \tilde{\mathcal{B}}_I^H$, $vol(B) \geq vol(B_1(0))(\frac{\delta_I^H}{16})^d$. Furthermore, by Theorem 14.2 of (Wu (2017)), we have

$$|\tilde{\mathcal{B}}^H| = |\mathcal{C}_I^H| \leq (\frac{2^3 3}{\delta_I^H})^d \frac{\delta_I^{Hd}}{vol(B_1(0))} = \frac{2^{3d}3^d}{vol(B_1(0))}.$$

Clearly, if each set in $\tilde{\mathcal{B}}_I^H$ contains $\eta N_I^H$ sample inputs, then $H$ is $(\frac{\delta_I^H}{8}, \eta)$-covered.

For all $B^H \in \tilde{\mathcal{B}}^H$, we define the event:

$$\mathcal{E}^{B^H}(N) := \{B^H \text{ contains } \geq \eta N \text{ inputs if } N \text{ samples are in } H.\}.$$

Then, we can apply the same arguments as the ones given in **(I)** to obtain:

$$\mathbb{P}\left(H \text{ is } (\frac{\delta_I^H}{8}, \eta)\text{-covered}|\bar{N}_I^H\right) \geq \mathbb{P}\left(\forall B^H \in \tilde{\mathcal{B}}^H : \mathcal{E}^{B^H}(\bar{N}_I^H)\right) \geq \prod_{B^H \in \tilde{\mathcal{B}}^H} \mathbb{P}\left(\mathcal{E}^{B^H}(\bar{N}_I^H - \eta|\tilde{\mathcal{B}}^H|\bar{N}_I^H)\right)$$

$$\geq \prod_{B^H \in \tilde{\mathcal{B}}^H} \mathbb{P}\left(\mathcal{E}^{B^H}\left(\bar{N}_I^H(1 - \eta\frac{2^{3d}3^d}{vol(B_1(0))})\right)\right) = \prod_{B^H \in \tilde{\mathcal{B}}^H} \mathbb{P}\left(\mathcal{E}^{B^H}(\frac{1}{2}\bar{N}_I^H)\right)$$

where the last equality follows from the fact that by assumption, $\eta = \frac{vol(B_1(0))}{2^{3d+1}3^d}$. Following the same approach as in **(I)**: we consider the random variables $M^{B^H}(N) := |\{i \in \{1, ..., N\} : x_i \in B^H\}|$ where $x_i$ are samples that are selected uniformly on $H$. For all $B^H \in \tilde{\mathcal{B}}^H$, $M^{B^H}(N)$ can be modelled as the sum of independent Bernouilli variables with success probability: $p = \frac{vol(B^H)}{vol(H)} \geq \frac{vol(B_1(0))(\frac{\delta_I^H}{16})^d}{\delta_I^{Hd}} = \frac{vol(B_1(0))}{16^d}$ and satisfying

$$\mathbb{E}\left[M^{B^H}(\frac{\bar{N}_I^H}{2})\right] = \frac{\bar{N}_I^H}{2}p \geq \bar{N}_I^H\frac{vol(B_1(0))}{2^{4d+1}}.$$

Using this inequality, we observe:

$$\eta\bar{N}_I^H = \frac{vol(B_1(0))}{2^{3d+1}3^d}\bar{N}_I^H \leq \bar{N}_I^H\frac{vol(B_1(0))}{2^{4d+1}} \leq \mathbb{E}[M^{B^H}(\frac{\bar{N}_I^H}{2})].$$

Therefore, leveraging the same arguments as the ones utilised in **(I)**, we can apply Lemma 1 of Stone (1982) to obtain:

$$\mathbb{P}\left(\mathcal{E}^{B^H}(\frac{1}{2}\bar{N}_I^H)\right) \geq 1 - (\frac{2}{e})^{\bar{N}_I^H \frac{vol(B_1(0))}{2^{4d+2}}}.$$

---

[20]In fact, a closed form for $vol(B_1(0))$ is known, $vol(B_1(0)) = \frac{\pi^{\frac{d}{2}}}{\Gamma(\frac{d}{2}+1)}$.

By construction, we have that for all $\epsilon > 0$ and $I = I(\epsilon) \in \mathbb{N}$, $\{\bar{N}_I^H\}_{H \in \mathcal{H}_I} \in V_I(n)$ which implies that for all $H \in \mathcal{H}_I$, $\bar{N}_I^H \geq c(\epsilon)$. Combining this bound with the lower bound derived above, we obtain for all $H \in \mathcal{H}_I$:

$$\mathbb{P}(H \text{ is } \left(\frac{\delta_I^H}{8}, \eta\right)\text{-covered}|\bar{N}_I^H) \geq \prod_{B^H \in \tilde{\mathcal{B}}^H} \mathbb{P}\left(\mathcal{E}^{B^H}(\frac{1}{2}\bar{N}_I^H)\right) \geq \prod_{B^H \in \tilde{\mathcal{B}}^H} 1 - (\frac{2}{e})^{\bar{N}_I^H \frac{vol(B_1(0))}{2^{4d+2}}}$$

$$\geq \left(1 - (\frac{2}{e})^{\bar{c}(\epsilon)\frac{vol(B_1(0))}{2^{4d+2}}}\right)^{|\tilde{\mathcal{B}}^H|} \geq \left(1 - (\frac{2}{e})^{\bar{c}(\epsilon)\frac{vol(B_1(0))}{2^{4d+2}}}\right)^{\frac{2^{3d}3^d}{vol(B_1(0))}}.$$

As the lower bound derived above does not depend on $\{\bar{N}_I^H\}_{H \in \mathcal{H}_I} \in V_I(n)$, we plug it into the initial expression to obtain

$$\sum_{\{\bar{N}_I^H\}_{H \in \mathcal{H}_I} \in V_I(n)} \left(\prod_{H \in \mathcal{H}_I} \mathbb{P}(H \text{ is } (\frac{\delta_I^H}{8}, \eta)\text{-covered}|\bar{N}_I^H)\right) \mathbb{P}(\{N_I^H\}_{H \in \mathcal{H}_I} = \{\bar{N}_I^H\}_{H \in \mathcal{H}_I}|\mathcal{A}_\epsilon^1)$$

$$\geq \prod_{H \in \mathcal{H}_I} \left(1 - (\frac{2}{e})^{\bar{c}(\epsilon)\frac{vol(B_1(0))}{2^{4d+2}}}\right)^{\frac{2^{3d}3^d}{vol(B_1(0))}} \sum_{\{\bar{N}_I^H\}_{H \in \mathcal{H}_I} \in V_I(n)} \mathbb{P}(\{N_I^H\}_{H \in \mathcal{H}_I} = \{\bar{N}_I^H\}_{H \in \mathcal{H}_I}|\mathcal{A}_\epsilon^1)$$

$$= \prod_{H \in \mathcal{H}_I} \left(1 - (\frac{2}{e})^{\bar{c}(\epsilon)\frac{vol(B_1(0))}{2^{4d+2}}}\right)^{\frac{2^{3d}3^d}{vol(B_1(0))}} \geq \left(1 - (\frac{2}{e})^{\bar{c}(\epsilon)\frac{vol(B_1(0))}{2^{4d+2}}}\right)^{\frac{2^{4d}3^d(C_1(d)MK)^d}{vol(B_1(0))\epsilon^d}}$$

where the last inequality follows from: for all $\epsilon > 0$ satisfying $\epsilon < C_1(d)MK$, we have $I \leq 2C_1(d)\frac{MK}{\epsilon}$ implying $|\mathcal{H}_I| = I^d \leq \frac{(2C_1(d)MK)^d}{\epsilon^d}$. It is relatively straightforward to see that the lower bound derived above converges to 1 as $\epsilon$ goes to 0. Therefore:

$$1 \geq \lim_{\epsilon \to 0^+} \mathbb{P}(\mathcal{A}_\epsilon^2|\mathcal{A}_\epsilon^1) \geq \lim_{\epsilon \to 0^+} \left(1 - (\frac{2}{e})^{\bar{c}(\epsilon)\frac{vol(B_1(0))}{2^{4d+2}}}\right)^{\frac{2^{4d}3^d(C_1(d)MK)^d}{vol(B_1(0))\epsilon^d}} = 1.$$

This shows:

$$\lim_{\epsilon \to 0^+} \mathbb{P}(\mathcal{A}_\epsilon^c) = 1 - \lim_{\epsilon \to 0^+} \mathbb{P}(\mathcal{A}_\epsilon^2|\mathcal{A}_\epsilon^1) \lim_{\epsilon \to 0^+} \mathbb{P}(\mathcal{A}_\epsilon^1) = 1 - 1 \cdot 1 = 0$$

and concludes the proof of Theorem 3.15.

■

**Proof of Corollary 3.16 (Finite Sample Guarantee – Gaussian Noise).**

The assumptions of Corollary 3.16 imply that the event $\mathcal{A}_\epsilon$ defined in the proof of Theorem 3.15 holds with constants specified in the statement of the corollary: $\eta \in (0, 1)$ and $C_1(d) = \frac{16d^2\sqrt{d}d^{\max\{\frac{1}{q} - \frac{1}{2}, 0\}}}{\sqrt{\eta}}$.

Therefore, the same arguments as the ones used in the first part of the proof of Theorem 3.15 can be applied in order to obtain $\forall \epsilon \in (0, \frac{C_1(d)MK}{3})$:

$$\sup_{f \in C^2(\mathcal{X}, K)} \mathbb{P}(|\hat{L}_I(f) - L_p^*(f)| > \epsilon) \leq \mathbb{P}\left(\max_{H \in \mathcal{H}_I} \{\|\mathbb{E}\left[\hat{\beta}_I^H \Big| G_I^{\mathcal{X}}\right] - \hat{\beta}_I^H\|_2\} > \frac{\epsilon}{4n_q}\right)$$

where we note that as we consider $\mathbb{P}(|\hat{L}_I(f) - L_p^*(f)| > \epsilon)$ instead of $\mathbb{P}(Loss(x^{\hat{L}_I(f)}, f) > \epsilon)$, a factor 2 disappears.

This implies that we can consider the statement $\forall \delta \in (0, \frac{1}{2})$:

$$\mathbb{P}\left(\max_{H \in \mathcal{H}_I} \{\|\mathbb{E}\left[\hat{\beta}_I^H \Big| G_I^{\mathcal{X}}\right] - \hat{\beta}_I^H\|_2\} > \frac{\epsilon}{4n_q}\right) \leq \delta$$

in order to show Corollary 3.16.

Let $\epsilon \in (0, \frac{C_1(d)MK}{3})$ and $\delta \in (0, \frac{1}{2})$ be arbitrary. Again applying the arguments of Theorem 3.15, with $\delta$, we obtain that the above expression holds if $\epsilon \leq \bar{\epsilon}^* := \min(\epsilon_1, \epsilon_2, \epsilon_3)$ where $\epsilon_1, \epsilon_2, \epsilon_3 > 0$ depend on $\delta$ and $\epsilon$. We consider each of the epsilon separately:

($\epsilon_1$). By construction, $\epsilon_1 = \frac{C_1(d)MK}{3}$ and by assumption: $\epsilon \in (0, \frac{C_1(d)MK}{3})$. Therefore, $\epsilon < \epsilon_1$ holds.

($\epsilon_2$). $\epsilon_2$ is set such that the relation: $|\mathcal{H}_I| \geq 2\log(\frac{1}{1-\delta})$ holds (in order to apply Lemma B.3). Substituting $|\mathcal{H}_I| = I^d$ and $\delta \leq \frac{1}{2}$ into the expression yields:

$$|\mathcal{H}_I| \geq 2\log(\frac{1}{1-\delta}) \impliedby I^d \geq 2\log(2).$$

As $\epsilon \in (0, \frac{C_1(d)MK}{3})$ and $I$ is defined to be $I = \lceil C_1(d)\frac{MK}{\epsilon} \rceil$, $I \geq 3$ and the above holds. Therefore, $\epsilon < \epsilon_2$ holds.

($\epsilon_3$). $\epsilon_3$ is defined such that the following relation holds:

$$\forall H \in \mathcal{H}_I, \ \bar{N}_I^H \geq \frac{\sigma^2 K^2}{C_2(d)\epsilon^4} \log(\frac{4|\mathcal{H}_I|}{\log(\frac{1}{1-\delta})})$$

where $\bar{N}_I^H$ is guaranteed number of samples in $H \in \mathcal{H}_I$. By the assumptions of Corollary 3.16, we have $\bar{N}_I^H = \tilde{C}^*(\eta, d)\frac{\sigma^2 K^2}{\epsilon^4} \log(\frac{4^{\frac{1}{d}}I}{\log(\frac{1}{1-\delta})^{\frac{1}{d}}})$ for all $H \in \mathcal{H}_I$. By construction, $\tilde{C}^*(d) = \frac{2^{10}n_q^2 C_1(d)^2 d^2}{\eta} = \frac{d}{C_2(d)}$. Therefore:

$$\bar{N}_I^H \geq \frac{\sigma^2 K^2}{C_2(d)\epsilon^4} \log(\frac{4|\mathcal{H}_I|}{\log(\frac{1}{1-\delta})}) \iff \bar{N}_I^H \geq \frac{d\sigma^2 K^2}{C_2(d)\epsilon^4} \log(\frac{4^{\frac{1}{d}}I}{\log(\frac{1}{1-\delta})^{\frac{1}{d}}})$$

holds by design for all $\epsilon \in (0, \frac{C_1(d)MK}{3})$ . Therefore, $\epsilon < \epsilon_3$ holds.

Thus, we have shown: $\forall \epsilon \in (0, \frac{C_1(d)MK}{3}), \delta \in (0, \frac{1}{2})$

$$\sup_{f \in C^2(\mathcal{X}, K)} \mathbb{P}(|\hat{L}_I(f) - L_p^*(f)| > \epsilon) \leq \delta.$$

$\blacksquare$

## C   Proofs: Sample Complexity of Adaptive Lipschitz Optimisation

In this section we prove the lower bound on the sample complexity of certified adaptive Lipschitz optimisation algorithms given in Section 4.

**Proof of Proposition 4.1 (Sample Complexity of Adaptive Lipschitz Optimisation).**

Fix $\epsilon > 0$, $L^* \geq 0$ and let $A$ be a non-adaptive certified optimisation algorithm which takes a given Lipschitz constant $\bar{L} > L^*$ as a hyperparameter. Using the notation given in Section 4: with $n$-queries to the oracle, $A$ outputs a triplet $((x_n, f(x_n^*), \zeta_n))_{n \in \mathbb{N}}$ where $x_n$ is the n-th query point, $f(x_n^*)$ is the generated estimate of $\max_{x \in \mathcal{X}} f(x)$ after $n$ queries and $\zeta_n \geq 0$ is an error certificate that guarantees: $\max_{x \in \mathcal{X}} f(x) - f(x_n^*) \leq \zeta_n$. From Theorem 3 of Bachoc et al. (2021) with $\epsilon_0 < 2^{d-1}ML^*$ (this follows from the fact that $\mathcal{X}$ is a hypercube), we have that for all $f \in \{h : \mathcal{X} \to | h$ is Lipschitz cont. and $L_p^*(h) < \bar{L}\}$:

$$N(A, f, \epsilon) \geq \frac{c_d L^{*d}(1 - \frac{L^*}{\bar{L}})^d}{1 + \lceil \log_2(\frac{\epsilon_0}{\epsilon}) \rceil} \int_{\mathcal{X}} \frac{dx}{(f(x^*) - f(x) + \epsilon)^d}. \tag{5}$$

where $c_d > 0$ (It is important to note that the term $c_d L^{*d}$ is not optimised in (Bachoc et al. (2021)) and could be improved in future work). Now, consider an adaptive Lipschitz optimisation algorithm $\tilde{A}$ with a

separable Lipschitz constant estimator $\tilde{L}_{\tilde{A}}(f)$. If $\tilde{L}_{\tilde{A}}(f)$ can be guaranteed to be feasible (e.g. see discussion after Corollary 3.13) then equation (5) holds for $\tilde{A}$ and $\forall f \in C^2(\mathcal{X}, K) \cap \mathcal{F}_p(L^*)$ with $\bar{L}$ replaced by $\tilde{L}_{\tilde{A}}(f)$ [21]. The precision at which $\tilde{L}_{\tilde{A}}(f)$ estimates $L^*(f)$ therefore directly impacts the lower bound on $N(\tilde{A}, f, \epsilon)$. From the Corollary 2.3 given in Section 2, we have that $\forall n \in \mathbb{N}$, any Lipschitz learning algorithm $\tilde{L} \in \mathcal{L}_{n,p}$ that guarantees feasible Lipschitz constants must satisfy

$$\sup_{f \in C^2(\mathcal{X}, K) \cap \mathcal{F}_p(L^*)} \tilde{L}(f) - L^* \geq C \frac{MK}{\sqrt[d]{n}}.$$

for some $C > 0$. This implies that for all $A \in \mathcal{A}$, there exists a non-empty set $\mathcal{G}_A \subset C^2(\mathcal{X}, K) \cap \mathcal{F}_p(L^*)$ such that $\forall f^* \in \mathcal{G}_A$, $\tilde{L}_A(f^*) - L^* \geq \frac{C}{2} \frac{MK}{\sqrt[d]{n}}$. Then, denoting $I(f) := \frac{c_d L^{*d}}{1 + \lceil \log_2(\frac{\epsilon_0}{\epsilon}) \rceil} \int_{\mathcal{X}} \frac{dx}{(f(x^*) - f(x) + \epsilon)^d}$ in order to alleviate notation, we have $\forall A \in \mathcal{A}$,

$$N(A, \epsilon) := \sup_{f \in C^2(\mathcal{X}, K) \cap \mathcal{F}_p(L^*)} N(A, f, \epsilon) \geq \sup_{f \in C^2(\mathcal{X}, K) \cap \mathcal{F}_p(L^*)} \left\{ (1 - \frac{L^*}{\tilde{L}_A(f)})^d I(f) \right\}$$

$$\geq \left( 1 - \frac{L^*}{L^* + \frac{C}{2} \frac{MK}{\sqrt[d]{N(A, \epsilon)}}} \right)^d \sup_{f \in \mathcal{G}_A} \left\{ I(f) \right\}.$$

Re-arranging the terms in the above expression, we can obtain:

$$\frac{C}{2} MK \sup_{f \in \mathcal{G}_A} \left\{ \sqrt[d]{I(f)} \right\} \leq L^* (\sqrt[d]{N(A, \epsilon)})^2 + \frac{C}{2} MK \sqrt[d]{N(A, \epsilon)}$$

which can be solved to give the lower bound

$$\sqrt[d]{N(A, \epsilon)} \geq C_1 \frac{MK}{L^*} \left( \sqrt{1 + C_1 \frac{L^* \sup_{f \in \mathcal{G}_A} \left\{ \sqrt[d]{I(f)} \right\}}{MK}} - 1 \right)$$

where $C_1 > 0$ is a constant. In order to conclude the proof, a lower bound on $\sup_{f \in \mathcal{G}_A} \left\{ \sqrt[d]{I(f)} \right\}$ is needed. To do so, we note that $I(f)$ is minimised when $f$ is constant. We therefore consider the set of functions $\mathcal{F}_0$ defined in the proof of Theorem 2.2. From the proof of Theorem 2.2, we have that if $N(A, \epsilon) \leq (\frac{MK}{L^*})^d (\frac{C_2}{2})^d$, then $L^* \leq \frac{C_2}{2} \frac{MK}{\sqrt[d]{N(A, \epsilon)}}$ which implies $\mathcal{F}_0(\frac{(L^*)^2}{0.8K}, \frac{0.8}{7.75}(\frac{K}{L^*})^2) \subset \mathcal{G}_A$. Using $f(x^*) = \frac{(L^*)^2}{0.8K}$, $\forall f \in \mathcal{F}_0(\frac{(L^*)^2}{0.8K}, \frac{0.8}{7.75}(\frac{K}{L^*})^2)$, we obtain the lower bound:

$$\sup_{f \in \mathcal{G}_A} \left\{ I(f) \right\} \geq \frac{c_d L^{*d}}{1 + \lceil \log_2(\frac{\epsilon_0}{\epsilon}) \rceil} \frac{\mathcal{V}_{\mathcal{X}}}{(\epsilon + \frac{(L^*)^2}{0.8K})^d}.$$

Therefore, if $N(A, \epsilon) \leq (\frac{MK}{L^*})^d (\frac{C_2}{2})^d$, the above expression can be plugged into the lower bound on $\sqrt[d]{N(A, \epsilon)}$. We obtain

$$\sqrt[d]{N(A, \epsilon)} \geq C_1 \frac{MK}{L^*} \left( \sqrt{1 + C_3 \frac{1}{\sqrt[d]{(1 + \lceil \log_2(\frac{\epsilon_0}{\epsilon}) \rceil)}(\frac{\epsilon K}{L^{*2}} + 1)}} - 1 \right)$$

(for some constant $C_3 > 0$) which corresponds to the first half of the lower bound stated in the Proposition 4.1. In order to derive the second part of the expression, we consider the case where $N(A, \epsilon) > (\frac{MK}{L^*})^d (\frac{C_2}{2})^d$. In this case, an alternative lower bound on $\sup_{f \in \mathcal{G}_A} \left\{ I(f) \right\}$ needs to be derived. In order to do so, we consider the following class of functions,

$$\left\{ g : \mathcal{X} \subset \mathbb{R}^d \to \mathbb{R} \mid \forall x \in \mathcal{X}, g(x) = f(x) + (L^* - \frac{C_2}{2} \frac{MK}{\sqrt[d]{N(A, \epsilon)}}) x_1 \text{ where } f \in \mathcal{F}_0 \left( \frac{(L^*)^2}{0.8K}, \frac{0.8}{7.75}(\frac{K}{L^*})^2 \right) \right\}$$

which belongs to $\mathcal{G}_A$ by construction. However, as obtaining a tight lower bound on $\sup_{f \in \mathcal{G}_A} \left\{ \sqrt[d]{I(f)} \right\}$ is technically infeasible for this class, we simplify the problem by removing the functional input from

---

[21]Note: this is only possible as we are considering adaptive Lipschitz optimization algorithms which are separable.

$\mathcal{F}_0(\frac{(L^*)^2}{0.8K}, \frac{0.8}{7.75}(\frac{K}{L^*})^2)$ and considering the simple linear function $f^* : \mathcal{X} \subset \mathbb{R}^d \to \mathbb{R}$, $f^*(x) = L^* x_1$ which belongs trivially to $\mathcal{G}_A$. In this case, we can compute the lower-bound

$$\sup_{f \in \mathcal{G}_A} \{I(f)\} \geq c_d \frac{L^{*d+1} M^{d-1}}{(1 + \lceil \log_2(\frac{\epsilon_0}{\epsilon}) \rceil) \epsilon^{d-1}} (d-1) \frac{(\frac{LM}{\epsilon} + 1)^{d-1} - 1}{(\frac{LM}{\epsilon} + 1)^{d-1}} \geq \frac{c_d(d-1)}{2} \frac{L^{*d+1} M^{d-1}}{(1 + \lceil \log_2(\frac{\epsilon_0}{\epsilon}) \rceil) \epsilon^{d-1}}$$

where the last inequality follows from the fact that $LM \geq \epsilon$ since $\epsilon \in (0, \epsilon_0)$. Plugging this expression into the lower bound on $\sqrt[d]{N(A, \epsilon)}$, we obtain

$$\sqrt[d]{N(A, \epsilon)} \geq C_1 \frac{MK}{L^*} \left( \sqrt{1 + C_4 \frac{L^{*2}}{\epsilon K} \sqrt[d]{\frac{L^*(d-1)\epsilon}{M(1 + \lceil \log_2(\frac{\epsilon_0}{\epsilon}) \rceil)}}} - 1 \right)$$

(for some constant $C_4 > 0$) which corresponds to the second half of the lower bounding expression. Note: in the statement of the proposition we simply set $C_2 = \min(C_3, C_4)$ as the used constant.

∎

