# OpenReview forum: "On the Sample Complexity of Lipschitz Constant Estimation"
_TMLR — Accepted by TMLR_

### Review · Reviewer_79hU · 2023-07-06

**Summary Of Contributions:**

The paper studies the problem where we have (potentially noisy) oracle access to a function $f : \mathcal{X} \rightarrow \mathbb{R}$ (where $X \subset \mathbb{R}^d$) and we want to learn the Lipschitz constant for $f$ to additive error $\varepsilon$. We are guaranteed that all second order partial derivatives exist and are bounded as $|\frac{\partial^2 f}{\partial x_i \partial x_j}| \leq K$. Upper and lower bounds on the oracle complexity of this problem are studied.

**Lower bounds**: Two different setting are considered for building lower bounds. One setting is the noiseless oracle, where we can exactly compute $f(x)$ for any $x\in\mathcal{X}$. When $\mathcal{X} = [0,M]^d$, i.e. when the domain of $f$ is the hypercube with side length $M$, the authors show a lower bound of roughly $n \geq (\frac{MK}{\varepsilon})^d$. In the noisy oracle setting, where the output of the oracle is perturbed by iid zero-mean bounded-variance noise, the authors show a lower bound of roughly $n \geq (\frac{MK}{\varepsilon})^{2d}$ under a pretty opaque "Assumption 4" (discussed in more detail later in this review).

**Upper Bounds**: An algorithm called LCLS is proposed. The algorithm works by partitioning the input space $\mathcal{X}$ into a very large number of partitions, sampling points within each partition, solving a least squares problem in each partition using the responses from the oracle to compute a local Lipschitz constant, and returning the largest Lipschitz constant computed amongst all the partitions.

The authors prove this algorithm matches the lower bound in the noiseless case, and nearly matches the lower bound in the noisy case. While the state-of-the-art algorithms have runtime scale quadratically in the number of points sampled, LCLS has a linear runtime. So, while existing algorithms achieve sample complexities similar to LCLS, the necessary number of samples is very large ($n = \Omega( (\frac{MK}{\varepsilon})^d )$), so this improvement in the runtime while maintaining optimal sample complexity is the key contribution for the upper bound.

Experiments are provided to explore the empirical performance of LCLS.

The above results were stated for the Lipschitz constant with respect to the $p=2$ norm. For the $p=1$ norm Lipschitz constant, similar results are presented.

**Audience:**

Yes

**Claims And Evidence:**

Yes

**Requested Changes:**

I'll list a combination of requested changes, typos, and just questions I have. They'll be ordered based on where in the paper the questions occur.

1. [Page 3, second bullet point under "2."] You stated "without any prior knowledge..." as a valuable upside, but you haven't really given a concrete example of what that means. What sorta prior knowledge does Strongin require?
1. [Page 3, last sentence of bullet point "4."] I don't understand what you mean by the last sentence. "specifically designed" for what?
1. [Page 4, second line] Add "any" before "$L \geq 0$"
1. [Page 5, after "In this paper"] Make the exponent of $d$ into $d^{\frac1p - \frac12}$. Since you only consider $p\in\\{1,2\\}$, it's just cleaner this way.
1. [Page 6, last line of Theorem 2.5] Again simplify the exponent. Also, $Q^*$ isn't defined. It seems to be only defined in the proof, but also it's defined in a way that depends on the noise distribution and the location of the ball? Makes me think that maybe it's more of a change to Assumption 4? Not sure how you'd rather clarify that.
1. [Page 6, Remark 2.6] Could you explain what would change quantitatively if the noise was (e.g.) subgaussian? Would the dependence on $\delta$ become logarithmic while everything else stays the same, or would it be more drastic with the exponent of $2d$ returning to be $d$?
1. [Page 8, Figure 1] Add $I\in\mathbb{N}$ subscript to $\\{\hat L_I\\}$
1. [Page 8, Defn 3.2] Make "we define" into the start of a new sentence.
1. [Page 8, Prop 3.3] Add $d^2$ into the big-Oh runtime of LCLS. It's personal preference I guess, but it strikes me as odd to not mention the impact of $d$ here in the runtime.
1. [Page 9, second line] Add the dimension dependence to Strongin's big-Oh as well.
1. [Page 9, Defn 3.5] Define $D_I^H$
1. [Page 9, Defn 3.5] This requirement of containing a multitude of many disjoint covers isn't very intuitive, but it sorta is? I'd add a few lines explaining this requirement, and the usual scale of both $N_I^H$ and $\eta$.
1. [Page 9, fist line after Assumption 5] Say "used" not "necessary". Necessary is a technically stronger claim, and I don't think you show it to be necessary. It's merely sufficient, afaik.
1. [Page 9, footnote 6] Don't use the appendix to define notation used in the body of the paper. I've no issue with the appendix summarizing all the notation that appears in the body, and introducing new notation unique to the appendix, but it shouldn't be defining things for the body.
1. [Page 10, Remark 3.7] I overall really like this remark btw! It's really nice to have this sorta explanation! But the second bullet point a bit unintuitive still here. It's worth writing in the remark that the second bullet point is really only for the noisy setting, and even then it's still worth explaining why the number of samples taken within partition $H$ should depend on the number of partitions. That feels rather unintuitive to me.
1. [Page 10, Corol 3.8] Drop the $\forall D_{\gamma}^n$ bit -- this is the noiseless setting.
1. [Page 11, Definition of $N(I)$] The curly brackets don't make use of the symbol $n$. Maybe make it intro $\min \\{|D\_I^H| ~ : ~ D\_I^H \text{contains a ...} \\}$?
1. [Page 12, before Section 3.4] At this point of reading, I felt unclear if we had any good reason to assume we know a good value for $K$. If we don't know the Lipschitz constant, how can we know a good bound on the second derivative, especially when the empirical performance of LCLS does seem to depend on our knowledge of $K$?
1. [Page 12, Section 3.4.1] Add more confidence intervals to these plots. It's a randomized algorithm, so I should be able to see those standard deviations, or maybe the 25th and 75th quartiles, on these plots.
1. [Page 12, Section 3.4.1] Optimization is misspelled, using "s" instead of "z"
1. [Page 13, Figure 2] The middle column should probably still use a log x-axis to legibility? Not suer sure why you have both the first and second columns, frankly.
1. [Page 13, Table 1] Point out somewhere (maybe in the table? maybe not in the table? not sure) that functions (a) and (d) are based on the lower bound construction
1. [Page 14, Figure 3] Use more points on the x-axis. There's waaaay too few, especially for the right two columns.
1. [Page 14, Figure 3] Consider interpreting somewhere why the computation time is only sometimes different for the two Strongins. It's not your algorithm, so it's not a big deal, but it's confusing to see.
1. [Page 14, Figure 3] LCLS completely violates the lower bound which it should be tracking on function (b). Interpret what went wrong. Also, this baaadly needs confidence intervals, the error plot for function (b) looks nonsensical.
1. [Page 15, Figure 4] I think the green and blue labels should be M-Stron instead of LACKI
1. [Page 16, sentence starting with "In some sense"] In what sense is this true? Does this really feel like necessary cost? I don't see the idea of this being "necessary" for any algorithm family other than LCLS. It's necessary for LCLS, sure, but the sentence feels like it's talking about what's necessary for any algorithm to achieve linear-in-$n$ runtime.
1. [Page 16, sentence starting with "The last illustration"] Add commas around ", provided in Figure 5,"
1. [Page 16, same paragraph as above] Still feels like you should include the algorithms of Beliakov and Calliess in those plots though. Might be worth using those algorithms in practice for the time being if they way outperform LCLS.
1. [Page 18, Prop 4.1] There's an extra parenthesis
1. [Page 18, Prop 4.1] If you can find a way to format it visually nicely, put the sup before the big equation
1. [Page 18, Prop 4.1] $\gamma$ and $\beta$ have the exact same words in their definitions.
1. [Page 19, Figure 6] Caption says "red" instead of "orange"
1. [Page 19, Last paragraph] "performs" should be "perform"
1. [Page 22, Lemma C.1] I think the $z_i$ should all be remove from $g_0$?
1. [Page 24, paragraph starting with "For any $L^*$"] How do you know that $\max\_{x \in \mathbb{R}^d} \|\| \nabla g\_z^{L^*} \|\|_2 = L^*+  \max\_{x \in \mathbb{R}^d} \|\| \nabla g\_z^{0} \|\|_2$. I believe it as a triangle inequality, but equality needs a bit more justification.
1. [Page 24, paragraph starting with "In this proof,"] What is $A$? Is it supposed to be $M$?
1. [Page 24, Bullet point 3] Double dipping on the symbol $i$. Let it be either $i,j\in[d]$ or let it be the entry of $\bar z\_i$, but not both.
1. [Page 24, paragraph starting with "Since every"] Should be "analytic" not "analytical"
1. [Page 24, paragraph starting with "Since every"] This paragraph feels like an intuition paragraph, and isn't super rigorous. If that's right, then say so upfront at the start of the paragraph.
1. [Page 24, paragraph starting with "Since every"] What is $A$? Is it supposed to be $M$?
1. [Page 25, paragraph starting with "As noted in"] This is not nearly formal enough. Add rigor here, since this feels important. Why couldn't an algorithm information-theoretically just output the common Lipschitz constant that all of these function have? Add some sorta formal argument about picking one of these functions uniformly at random, except picking the zero function with sufficiently high probability that any accurate algorithm has to account for it, and continue as such.


**Strengths And Weaknesses:**

I like this paper. I think it easily merits being accepted to the journal.

The paper studies an interesting problem in a largely compelling way. The story of the paper is nice -- that lower bounds for lipschitz learning haven't been proven in the past, and the authors can show the existence algorithms with essentially optimal sample complexity and can even present an algorithm with faster runtime than existing work. Further, the newly proposed algorithm has a fundamentally different structure from the Lipschitz learning algorithms they mention from the prior work (I'm no expert in Lipschitz learning, so I'm supposing that the authors cited all the relevant prior work). The algorithm presented is flexible, allowing for a high sample complexity that achieves the rigorous worst-case bound, while also providing flexibility for more reasonable implementations in practice (i.e. by using fewer partitions or fewer samples per partition).

The paper is also well written. The authors spend a good deal of time and effort explaining and interpreting their results. The writing is clear and easy to follow. I've got some minor confusions or edits which I'll list in the "Requested Changes" section, but nothing severe.

At a high level, it's hard to say more than that -- a well motivated problem given new upper and lower bounds, involving a new algorithm with a faster runtime while maintaining near-optimal sample complexity, all bundled up in a well written paper.

That said, I do have one particular gripe involving the proof of the lower bound of the sample complexity in the noisy oracle setting. The proof of the noisy oracle sample complexity lower bound requires a technical assumption on the noise being "noisy enough", but the statement tracks as a bit too general to me to be possible, but also shouldn't be necessary. The assumption is "Assumption 4" on the top of page 6. Roughly speaking, the assumption says the following:
> There exists a ball $\mathcal{B}$ of small radius in the hypercube $[0,M]^d$ and a threshold $Q>0$ such that: for all oracle noise distributions $\mathcal{D}$ with zero-mean and variance $\sigma^2$, and for all Lipschitz learning algorithms $\mathcal{Alg}$, we know that at least one of the two following statements hold:
> 1. All the queries made by $\mathcal{Alg}$ in the ball $\mathcal{B}$ are sufficiently perturbed (as measured by $Q$) with probability 1; or
> 2. There exists an $L$-Lipschitz function $f$ such that, conditioned on the event that the queries made by $\mathcal{Alg}$ in the ball $\mathcal{B}$ are sufficiently perturbed (as measured by $Q$), the probability that $\mathcal{Alg}$ returns an estimate $\hat L$ within additive error $\varepsilon$ of $L$ is at most constant (i.e. and not arbitrarily close to 1).

This assumption is very technical, and is maybe plausible? The fact that this technical dichotomy has to hold for all noise distributions with zero mean and bounded variance is a very strong claim. It feels plausible that there exists pedantic noise distributions that violate this assumption though (for instance, if $\cD$ takes value zero with overwheliming probability, but is very large with very low probability). The paper does claim that this technical assumption is written this way to allow for more generality in their lower bounds. However, it's hard to approach and is plausibly wrong. The second bullet point is especially harsh -- saying that every Lipschitz-learning algorithm has a function which is hard to $\varepsilon$-learn in the ball $\mathcal{B}$ with bounded noise.

I have two inclinations about this assumption:
1. It's not clear the assumption needs to hold _for all_ zero-mean bounded-variance noise distributions. Most lower bounds just pick a single noise distribution and show a lower bound that's tight for that choice of additive noise.
1. It would really help to give a concrete example (with formal proof) of a natural setting where this assumption holds. I don't see how to prove the second bullet point rigorously, and that feels like a vital part of the broader lower bound claim.

This is a bit of a sticking point to me, since it does feel a bit like moving the goalpost of what the authors need to prove a lower bound against. The reason I'm sorta okay overall though is that this is a sticking point for merely part of their contributions. For instance, the lower bound against noiseless oracles has no such annoying assumption.

Again, overall, I like the result a lot, and recommend accepting it. It just has this one assumption that really gets me. Oh, also, there's a nice and thorough experiments section!

---

> ### Author Response · Authors · 2023-07-24
> **Response to Reviewer 79hU**
>
> Dear Reviewer 79hU,
>
> Thank you for the in-depth and thoughtful review and suggestions. We went thoroughly through the feedback you provided and will incorporate it in the revised version of the manuscript (which will be uploaded shortly).
>
>
> $\underline{\text{Response to "Strengths and Weaknesses":}} $
>
> Thank you for the overall positive assessment.
>
> Regarding comments on Assumption 4 and Theorem 2.5:
>
> The main goal of the lower bounds derived in the paper was to show that the general Lipschitz learning problem suffers from the curse of dimensionality. As this was shown to already be true in the noiseless setting (Theorem 2.2 and Corollary 2.3), the lower bounds on the sample complexity (in the noisy sampling setting) provided by Assumption 4 and Theorem 2.5 were essentially for intuition and for comparison with the upper bounds provided by the LCLS algorithm and are, as you pointed out, quite technical in nature.
>
>
> In the response to your in-depth comments on this assumption, we have modified the paper by providing alternative lower bounds under the more usual Gaussian noise assumption rather than the general noise assumption initially used. With this Gaussian constraint, we can completely remove Assumption 4 of Subsection 2.3. More precisely:
>
> $\textbf{(1)}$ We modify Subsection 2.3 to consider noise that is assumed to be Gaussian and a slightly different formulation of the Lipschitz learning problem (which is equivalent in practice for the majority of existing algorithms). We obtain a lower bound on the sample complexity rate of the order $\Omega \left(\frac{M^d K^{d+2}\log(\frac{M K}{\epsilon})}{ \epsilon^{d+4}}\right)$.
>
> $\textbf{(2)}$ We propose an additional result on the sample complexity of the LCLS algorithm under the Gaussian noise assumption. The obtained sample complexity rate matches the lower bound of $\textbf{(1)}$ exactly implying that the derived sample complexity rates of the Lipschitz learning problem with Gaussian sampling noise are optimal.
>
>
> $\underline{\text{Response to Requested Changes:}}$
>
> Points 3-17, 19, 21-24, 26, 28, 30-42 are being incorporated into the new version. Response and clarifications to remaining 8 questions/points:
>
> 1. In the noisy sampling setting (bounded noise), most existing modified Strongin-based algorithms depend
> on knowing bounds on the noise in order to be implementable. Note that while the LCLS algorithm depends
> on knowing K in order for its theoretical properties to hold, it can still be implemented heuristically if K is
> not known.
>
> 2. The benchmark methods were specifically designed for application in either the noiseless or the noisy sampling setting with bounded noise. By contrast, the LCLS estimator can be applied in the noiseless, bounded noise and unbounded noise sampling setting without modification (apart from hyperparameter tuning).
>
> 6. It would be more drastic with the $2d$ exponent becoming $d+4$ (see new Subsection 2.3).
>
> 18. The empirical performance and the theoretical bounds of the LCLS estimator do indeed depend on $K$. However, in practice, $K$ does not need to be tight in order to ensure that the LCLS algorithm converges towards the best Lipschitz constant at a known convergence rate (the tightness is only necessary for the lower bounds). Therefore, any $K$ that upper bounds the second derivatives could be used.
> One can imagine a setting where access to loose bounds on both the Lipschitz constant and on the second derivatives ($K$) is given but a more precise bound on the Lipschitz constant is desired. Then, applying the LCLS algorithm with the given $K$ provides a sequence of increasingly accurate (and feasible) Lipschitz constant estimates that converges to the best Lipschitz constant as the number of samples increases (see Corollary 3.12 and ensuing discussion). We will make this point clearer in the paper.
>
> 20. Optimisation with "s" instead "z" is the British english spelling of the word which is consistent with the rest of the paper (e.g. "characterise" instead of "characterize"). If this is a concern, we are happy to make the change for all relevant grammatical differences.
>
> 25. The lower bound is purely illustrative in the plot in order to provide an intuition on the convergence rate (i.e. an alternate constant $C$ to the ones derived in the theoretical results is used). The error plot of the LCLS in function (b) of Figure 3 actually follows from the changing shape of the partition of the input space. This is discussed in the section: "3.4.2 Discussion" of the paper.
>
> 27. Apologies for the confusion, this was not intended as a general statement but simply for the LCLS algorithm.
>
> 29. Unfortunately, the algorithms of Beliakov and Calliess cannot be applied in the unbounded noise setting and these plots are simply to showcase the fact that there is no performance loss by the proposed LCLS algorithm in the unbounded noise setting vs the bounded noise setting.

---

> > ### Comment · Reviewer_79hU · 2023-08-28
> > **Nice Responses!**
> >
> > Sorry for my late reply!
> >
> > I enjoyed reading the authors' responses, and they are all completely cool with me.
> >
> > I am curious where the gap between a rate of $(MK/\varepsilon)^d$ and $(MK/\varepsilon)^{2d}$ comes from, if Gaussian noise does not achieve it. If the final paper pivots to only use the $d+2$ rate instead of the $2d$ rate, then it doesn't necessarily matter for this paper, but it does seem curious to me to see if some nasty noise distribution can make the convergence rate exponentially worse.
> >
> > Either way, thanks for the response (sorry for the late reply again), and I hope y'all get the paper accepted!

---

> > > ### Author Response · Authors · 2023-08-30
> > > **Thanks and Answer to Question.**
> > >
> > > Hi,
> > >
> > > No worries, happy to discuss any comments or questions!
> > >
> > > (Short answer) This gap is essentially due to the fact that in the previous version of the paper, the sample complexity lower bounds in the subsection on noisy sampling considered a “sup” over all distributions with fixed variance $\sigma^2$ and assumed uncorrelated (instead of independent) noise variables which led to extreme noise distributions. (The noise variables were also not assumed to be identically distributed).
> > >
> > > (Long answer) More precisely, the gap in the rates comes from the following technical argument which no longer holds under independent Gaussian noise.
> > > Consider the average of the noise variables: $\Gamma:= \frac{1}{N}\sum_{i=1}^N \tilde \gamma_{x^{\hat L}_i} $. Then, under the distributional assumptions described above, one can show that the Chebyshev inequality is tight for  $\Gamma$. Note: This is not trivial as we are considering the average of random variables (if $\Gamma$ were any random variable, then the tightness of the Chebyshev bound is an elementary result) and follows from the same arguments as the ones given in [1] (see discussion on pages 2 and 3 on the tightness of equation (5)) which rely on the projective properties of distribution families (see Theorem 1 of [2]). The uncorrelatedness assumption is essential here.
> > >
> > > Since the Chebyshev inequality is tight (for some extreme noise distributions) and a "sup" over all noise distributions (of the class described above) is considered, we can utilise a "Chebyshev equality" in the computation of the lower bound on the sample complexity.
> > >
> > > In contrast, under independent Gaussian noise assumptions, this Chebyshev inequality is far from tight and therefore cannot be utilised in deriving the lower bound.
> > >
> > > Hope this was useful. Note: As you remarked, the above results/arguments are no longer in the paper as we consider the standard independent Gaussian noise case in this subsection instead.
> > >
> > > Thank you again for the in-depth review, it really helped improve the paper.
> > >
> > >
> > > References:
> > >
> > > [1] Rujeerapaiboon, N., Kuhn, D., & Wiesemann, W. (2018). Chebyshev inequalities for products of random variables. Mathematics of Operations Research, 43(3), 887-918.
> > >
> > > [2] Yu, Y. L., Li, Y., Schuurmans, D., & Szepesvári, C. (2009). A general projection property for distribution families. Advances in Neural Information Processing Systems, 22.

---

### Review · Reviewer_cH4u · 2023-07-08

**Summary Of Contributions:**

This study focuses on the problem of estimating the Lipschitz constant, given prior knowledge of a tight upper bound on the second-order partial derivatives. The author introduces novel algorithms known as LCLS (Lipschitz Constant Estimation by Least Squares Regression) and establishes a finite sample complexity guarantee for both noiseless and noisy scenarios. Additionally, the author proposes a theoretical lower bound for the sample complexity, demonstrating that the LCLS algorithm achieves optimal (or near-optimal) sample complexity in the absence (or presence) of noise. Lastly, the author applies the LCLS algorithm within an existing framework based on Lipschitz constants.

**Audience:**

Yes

**Broader Impact Concerns:**

There is no concern about the broader impact.

**Claims And Evidence:**

Yes

**Requested Changes:**

1. In Assumption 4(1), it seems more appropriate to state that the probability is related to $\delta$. Perhaps it should be modified to "probability $P \leq \delta," which would clarify the intended meaning.

2. In the experiments, the chosen dimension $d$ is relatively small ($d=3$). As previously mentioned in Weakness 2, both the LCLS algorithm and non-Strongin based estimators exhibit exponential running times with respect to the dimension $d$. To obtain a more comprehensive understanding of the time complexity, it would be beneficial to test the algorithm's performance with larger dimension sizes.

**Strengths And Weaknesses:**

Strengths:

1. The paper is well-written and easily comprehensible.

2. The proposed algorithm achieves optimal sample complexity in the noiseless case and provides a near-optimal guarantee in the presence of noise.

3. Compared to non-Strongin based estimators, the proposed algorithm exhibits linear time complexity with respect to the sample size, making it significantly more efficient than the baselines.

4. The experimental results validate the efficiency of the proposed algorithms.

Weaknesses:

1. Assumption 2 requires prior knowledge of a tight upper bound on the second-order partial derivatives, which may be considered unreasonable. When dealing with an unknown target function, obtaining the upper bound on the second-order partial derivatives is no easier than estimating the Lipschitz constant itself.

2. Although the proposed algorithm achieves a near-optimal guarantee, the sample complexity still exponentially depends on the dimensionality $d$, which is not acceptable. In this scenario, both the LCLS algorithm and non-Strongin based estimators exhibit exponential running times. It becomes meaningless to compare time complexities when both are unacceptable. A more desirable outcome would be a polynomial sample complexity with stricter assumptions.

3. Assumption 4 raises the question of why the significantly corrupted noise ($Q(\delta)$) does not depend on the sample size $n$. For zero-mean noise, the average corruption typically scales as $1/\sqrt{n}$.

---

> ### Author Response · Authors · 2023-07-24
> **Response to Reviewer cH4u**
>
> Dear Reviewer cH4u,
>
> Thank you for your comments and suggestions. We will incorporate your feedback in the revised version of the manuscript (which will be uploaded shortly).
>
> $\underline{\text{Response to Comments on Assumption 4:}}$
>
> Based on your and other reviewers' comments on Assumption 4, we have modified the paper by providing alternative lower bounds under the more usual Gaussian noise assumption, rather than the general noise assumption initially used.
>
>
> With this Gaussian constraint, we can completely remove Assumption 4 of Subsection 2.3. More precisely:
>
> $\textbf{(1)}$ We modify Subsection 2.3 to consider noise that is assumed to be Gaussian and a slightly different formulation of the Lipschitz learning problem (which is equivalent in practice for the majority of existing algorithms). We obtain a lower bound on the sample complexity rate of the order $\Omega \left(\frac{M^d K^{d+2}\log(\frac{M K}{\epsilon})}{ \epsilon^{d+4}}\right)$.
>
> $\textbf{(2)}$ We propose an additional result on the sample complexity of the LCLS algorithm under the Gaussian noise assumption. The obtained sample complexity rate matches the lower bound of $\textbf{(1)}$ exactly implying that the derived sample complexity rates of the Lipschitz learning problem with Gaussian sampling noise are optimal.
>
>
> $\underline{\text{Response to "Weaknesses":} }$
>
> 1. We note that Assumption 2 only requires a tight upper bound in order to derive the lower bounds on the sample complexity of the general Lipschitz learning problem (in which case the use of a tight bound is relatively standard). For all results pertaining to the proposed LCLS algorithm, any upper bound $K$ on the second order partial derivatives can be utilised. In this case, the LCLS algorithm would simply ensure convergence for a larger class of functions then needed at a slower rate. In practice, one can imagine a setting where access to loose bounds on both the Lipschitz constant and on the second derivatives ($K$) is given but a more precise bound on the Lipschitz constant is desired. Then, applying the LCLS algorithm with the given $K$ provides a sequence of increasingly accurate (and feasible) Lipschitz constant estimates which converges to the best Lipschitz constant as the number of samples increases (see Corollary 3.12 and ensuing discussion). Thank you for this comment, we will make this point clearer in the paper.
>
> 2. This point was the motivation for the lower bounds on the sample complexity derived in the first part of the paper which show that $\textbf{all}$ accurate Lipschitz learning algorithms must suffer from the curse of dimensionality. While obtaining polynomial sample complexity under stricter assumptions would be very useful, we believe that there is also value in developing an understanding of the general problem even given the exponential dependence on the dimensionality. In this case, the proposed LCLS algorithm has been shown - both theoretically and experimentally - to be significantly computationally faster than other existing theoretically-tractable Lipschitz learning algorithms. Future work may look at improving on the results of this paper by considering stricter assumptions, as you suggest.
>
> 3. See "Response to Comments on Assumption 4" above.
>
> $\underline{\text{Response to "Requested Changes":}}$
>
> 1. See "Response to Comments on Assumption 4" above.
>
> 2. As noted in the paper (see subsections "3.4.1 Experimental Setup" and "3.4.2 Discussion"), we do not consider higher dimensional functions (d>3) as the Strongin-based benchmark methods already struggle to converge for (d=3) in the noisy sampling setting (e.g. to obtain an estimation error of $<0.5$ on Function (d), the LCLS estimator needs $8.5$ seconds, while the Strongin-based approach needs $\sim 4000$ seconds). As the point of these illustrations (in particular Figure 4) was to illustrate the difference in computational speed between the LCLS algorithm and existing benchmarks and not to provide a comprehensive understanding (which as you have pointed out and as discussed in the paper will be exponential in the number of sample points) of the LCLS estimator, we believe that these figures are sufficient. We note that the theoretical computational complexity of the LCLS algorithm was stated in Proposition 3.3.

---

### Review · Reviewer_orNs · 2023-07-21

**Summary Of Contributions:**

### Motivation ###

The precise estimation of the Lipschitz constant is an important subroutine with applications in multiple problem domains, e.g., continuous optimization, where this constant is often assumed to be known to set step sizes. Prior work on this (rather fundamental) task typically makes assumptions about the setup, such as knowledge of the problem structure and the oracle access to the function being completely noise-free. Further, the majority of algorithms for this task lacked associated theoretical guarantees of correctness and sample complexity.


### Contributions ###

This paper's contributions are multi-fold: First, it provides lower bounds for the task of Lipschitz estimation under both noise-free and noisy zeroth-order oracle access. These bounds are also agnostic to the structure of the function in question. Second, the paper gives an estimation algorithm, which in the noise-free setting, matches the lower bound given by the paper, thereby establishing its optimality. Finally, the paper provides experimental evidence outperforming (in terms of runtime) the classical Strongin estimation algorithm in many common settings.

**Audience:**

Yes

**Claims And Evidence:**

Yes

**Requested Changes:**

I would be happy to accept the work as it is: a small typo that can be corrected is on page 3, the definition of Lipschitz function has the inequality flipped.

**Strengths And Weaknesses:**

### Strengths ###

The paper studies a very well-motivated problem, provides lower bounds that were previously unknown.

In my opinion, while the upper bound can be thought of as "natural" (the intuition being that a grid search over the $d$-dimensional space for an $\epsilon$-accurate estimate would naturally require $O{(1/\epsilon)}^d$ samples), the fact that this is also the lower bound is quite interesting.

These lower bounds are studied in the noisy setting too, which was one of the justifications to study this problem in the first place.

I also found the paper to be written in a very clear and engaging style.

---

> ### Author Response · Authors · 2023-07-24
> **Response to Reviewer orNs**
>
> Dear Reviewer orNs,
>
> Thank you for your comments and positive overall assessment. We will incorporate your requested change in the revised version of the manuscript (which will be uploaded shortly).
>
> $\underline{\text{Note on Assumption 4:}} $
>
> Based on other reviewers' comments on Assumption 4, we have modified the paper by providing alternative lower bounds under the more usual Gaussian noise assumption, rather than the general noise assumption initially used. With this Gaussian constraint, we can completely remove Assumption 4 of Subsection 2.3. More precisely:
>
> $\textbf{(1)}$ We modify Subsection 2.3 to consider noise that is assumed to be Gaussian and a slightly different formulation of the Lipschitz learning problem (which is equivalent in practice for the majority of existing algorithms). We obtain a lower bound on the sample complexity rate of the order $\Omega \left(\frac{M^d K^{d+2}\log(\frac{M K}{\epsilon})}{ \epsilon^{d+4}}\right)$.
>
> $\textbf{(2)}$ We propose an additional result on the sample complexity of the LCLS algorithm under the Gaussian noise assumption. The obtained sample complexity rate matches the lower bound of $\textbf{(1)}$ exactly implying that the derived sample complexity rates of the Lipschitz learning problem with Gaussian sampling noise are optimal.

---

### Author Response · Authors · 2023-07-25
**Post-review version of paper has been uploaded**

Thank you again for your reviews. We have uploaded the post-review version of the paper.

---

### Author Response · Authors · 2023-09-16
**Thank you!**

Dear Reviewers and Action Editor,

Thank you for your in-depth reviews and suggestions. The review process on TMLR has been a great experience and has improved both the quality and clarity of our paper.

We have uploaded the camera-ready version now with some minor edits:
- changed part of the title ("Lipschitz learning" to "Lipschitz constant estimation") to make it more widely understandable.
- made minor edits for typos and sentence structure.
- added some explanations in the appendix and improved the formatting of equations.

Thank you again,

Best wishes,

the authors

---

### Decision · Action_Editors · 2023-08-29

**Recommendation:** Accept with minor revision

**Comment:**

This paper tackles an important problem and produces a strong set of theoretical results.  Various reviewer comments were made about the assumptions, curse of dimensionality, and writing, but the reviewers were ultimately satisfied with the responses and unanimously agreed that this is a strong paper, particularly due to achieving minimax optimality in various cases of interest.

In addition, two reviewers indicated 'Clear Accept' and one of those two indicated 'Possibly Featured Certification but unsure due to somewhat limited scope'.  I judge that the scope is not a major limitation (i.e., it seems to be roughly at the level one would expect for a paper of this nature), so in view of the unanimous positive comments, I believe that this paper can be considered for Featured Certification.

I will enter 'Accept with minor revision' in order to encourage the authors to revisit the detailed reviewer comments and make any final edits.

**Audience:**

This paper is a strong match to TMLR.  Lipschitz assumptions are extremely prevalent in extensive topics, particularly optimization, and progress in estimating the Lipschitz constant will be appreciated by those working in such topics.

**Claims And Evidence:**

The main claims of this paper are theoretical, establishing upper and lower bounds on the sample complexity of learning the Lipschitz constant of a function under suitable assumptions.  In certain cases, such as noiseless and Gaussian noise, the minimax optimal rate is derived.  The "evidence" for these claims is more precisely in the form of proofs, which the reviewers didn't indicate any major concerns for (and were satisfied with following the response).  The LCLS algorithm attaining the upper bound is also seen to be effective experimentally, at least for some low-dimensional functions considered.  (The lower bounds indicate that *any* algorithm will suffer considerably in high dimensions.)